# Hagfish genome elucidates vertebrate whole-genome duplication events and their evolutionary consequences

Polyploidy or whole-genome duplication (WGD) is a major event that drastically reshapes genome architecture and is often assumed to be causally associated with organismal innovations and radiations. The 2R hypothesis suggests that two WGD events (1R and 2R) occurred during early vertebrate evolution. However, the timing of the 2R event relative to the divergence of gnathostomes (jawed vertebrates) and cyclostomes (jawless hagfishes and lampreys) is unresolved and whether these WGD events underlie vertebrate phenotypic diversification remains elusive. Here we present the genome of the inshore hagfish, *Eptatretus burgeri*. Through comparative analysis with lamprey and gnathostome genomes, we reconstruct the early events in cyclostome genome evolution, leveraging insights into the ancestral vertebrate genome. Genome-wide synteny and phylogenetic analyses support a scenario in which 1R occurred in the vertebrate stem-lineage during the early Cambrian, and 2R occurred in the gnathostome stem-lineage, maximally in the late Cambrian–earliest Ordovician, after its divergence from cyclostomes. We find that the genome of stem-cyclostomes experienced an additional independent genome triplication. Functional genomic and morphospace analyses demonstrate that WGD events generally contribute to developmental evolution with similar changes in the regulatory genome of both vertebrate groups. However, appreciable morphological diversification occurred only in the gnathostome but not in the cyclostome lineage, calling into question the general expectation that WGDs lead to leaps of bodyplan complexity.

Polyploidy or whole-genome duplication (WGD) is a dramatic genomic event commonly invoked causally in organismal evolution[1]. The generally accepted '2R hypothesis'[2,3] suggests that two rounds of WGD occurred during early vertebrate evolution (referred to as 1R and 2R); however, their timing and macroevolutionary consequences remain unclear[4–6]. Most studies agree that 1R occurred before the divergence of living vertebrates, but debate centres on whether 2R predated[7,8] or postdated[9–12] the divergence between cyclostomes and gnathostomes (Fig. 1c). Reconstruction of the ancestral vertebrate karyotype is fundamental to unravel the timing of 2R[8,12–15], but this goal has been stymied by a dearth of cyclostome genomes. The recently described genome of the sea lamprey (*Petromyzon marinus*) has been interpreted to support 2R occurring before[8] or after[12] the gnathostome–cyclostome split, or not at all (with the karyotype diversity explained as the result of large-scale segmental duplications[16,17]). Analysis of the Arctic lamprey (*Lethenteron camtschaticum*) genome has suggested that 2R occurred in the gnathostome lineage while independent WGD event(s) might have occurred in the lamprey lineage[11,18], perhaps shared with

✉e-mail: fergal@ebi.ac.uk; wwang@mail.kiz.ac.cn; Phil.Donoghue@bristol.ac.uk; zhangyong@ioz.ac.cn; jpascualanaya@gmail.com

the hagfish[11,19] (Fig. 1c). However, the lack of a hagfish genome assembly, the only major vertebrate group without a reference genome, has challenged attempts to constrain the number and phylogenetic timing of ploidy events in early vertebrate evolution. Here we describe the outcome of sequencing and comparative analysis of the genome of the inshore hagfish, *Eptatretus burgeri* (Fig. 1a,b).

## Chromosome-scale assembly and genome annotation

Similar to the lamprey[20], the hagfish genome undergoes somatic programmed DNA rearrangement in the way of chromosome elimination[21], making it crucial to obtain a reference assembly from a germline source. We sequenced DNA extracted from the testis of a single sexually mature male of *E. burgeri* and generated a preliminary draft assembly using ~240X of short-read Illumina data assisted by a Chicago in vitro proximity ligation assay at Dovetail Genomics[22] (Supplementary Table 5). We estimated the genome of *E. burgeri* at 3.12 Gb on the basis of *k*-mer frequency distribution (Extended Data Fig. 1a and Supplementary Table 4) in line with other hagfish species (~2.2–4.5 Gb)[23]. Chromosome conformation capture (Hi-C) data obtained from the testis DNA of a second individual were used to further scaffold the genome into a final assembly (v.4.0) containing 19 contact clusters, which we consider as chromosomes for subsequent analyses (Fig. 1d), and 9,295 unplaced scaffolds and contigs (Methods and Supplementary Tables 6–10). The genome was annotated following the Ensembl annotation pipeline[24], assisted by RNA-seq from 9 different adult tissues, and previous embryonic and juvenile transcriptomics data[19] (Supplementary Table 11 and Methods). We generated a final gene dataset of 16,513 protein-coding genes (with 27,960 transcripts), 446 long intergenic non-coding (linc)RNAs and a minority of other classes of non-coding RNA genes (Extended Data Fig. 1b). A total of 180 microRNA (miRNA) genes were found in the *E. burgeri* genome conserved with the hagfish *Myxine glutinosa*[25] belonging to 77 miRNA families and catalogued at MirGeneDB.org[26]. The germline haploid number of *E. burgeri* is 26. However, chromosome elimination occurs in somatic tissues of the hagfish, by which, in the case of *E. burgeri*, 8 pairs of microchromosomes are eliminated during development[21] (somatic *n* = 18). Cluster 19 and unplaced contigs/scaffolds probably correspond to these difficult-to-assemble microchromosomes, which presumably consist mainly of highly repetitive sequences and contain almost no protein-coding genes[21]. Consistently, 98.3% (16,240/16,513) of annotated genes are located in clusters 1–18.

BUSCO analyses show high levels of completeness of the hagfish genome (96.0 and 94.2% of single orthologues are present in the assembly and annotation, respectively; Fig. 1e and Extended Data Fig. 1c). GC-content distribution pattern analysis of the hagfish and other chordate genomes shows that the *E. burgeri* genome represents an intermediate condition between the lamprey and other chordates (Extended Data Fig. 1d), although having an overall GC content similar to that of the lamprey (46.7% and 48.1% for the hagfish and lamprey, respectively). While lamprey protein-coding gene sequences have been demonstrated to pose difficult challenges for comparative analyses due to their high GC content[27] (64.0%), the lower content in hagfish coding sequences (50.4%) is within the typical range of most gnathostomes and non-vertebrate chordates (42.5%–53.4%; Extended Data Fig. 1e,f and Supplementary Table 12). Lamprey represents an outlier in terms of both codon usage bias and amino acid composition, while the hagfish is more similar to other vertebrates (Fig. 1f,g). The hagfish genome contains, on average, significantly longer introns and intergenic regions than other vertebrates ($P < 2.2 \times 10^{-16}$, two-sided Wilcoxon rank-sum test), while the average length of coding sequences is similar to that of other chordates (Extended Data Fig. 1g, Supplementary Fig. 4 and Supplementary Tables 18–22). This might explain why hagfish genomes are larger than lamprey genomes[23]. Altogether, the hagfish genome provides essential, complementary information to lamprey

genomes, especially in analyses such as gene tree reconstruction and comparative genomics.

## Hagfish phylogenomics and gene family evolution

Whether hagfish form a clade with lampreys (Cyclostomata) or represent the sister group to all other vertebrates (including lampreys) has depended on whether molecular or morphological evidence are considered (Fig. 1c). Morphological studies historically supported cyclostome paraphyly but more recent analyses have recovered cyclostome monophyly (reviewed in ref. 28). Phylogenies inferred from molecular evidence have almost exclusively recovered cyclostome monophyly (reviewed in ref. 25). We used Bayesian inference to reconstruct the phylogeny of vertebrates from an alignment of 190 single-copy genes in all taxa analysed (84,017 sites), strongly supporting a monophyletic Cyclostomata (Fig. 2 and Extended Data Fig. 2a). We calculated the likelihood of gene duplication and loss patterns under the competing phylogenetic hypotheses[29] (see Methods), finding that patterns of gene gains and losses better fit cyclostome monophyly. To further compare the two alternative hypotheses of hagfish relationships, an approximately unbiased (AU) test[30] was performed, which strongly rejected cyclostome paraphyly (log likelihood difference = 7,947.7, AU = 0.004, multiscale bootstrap probability < 0.001). These results corroborate previous molecular analyses and recent morphological studies[28], supporting the view that cyclostomes are monophyletic.

To better understand the genomic changes accompanying major transitions in chordate evolution, we used a phylogeny-aware comparative genomic approach[31–33] to infer ancestral gene complements and gene family gains and losses across the vertebrate tree (Fig. 2, Extended Data Fig. 3 and Supplementary Tables 23–26). We observed two peaks of gene novelty in both the vertebrate and gnathostome stem-lineages (novel genes: +560 and +771, respectively), also characterized by a very low amount of gene losses (−341 and −382, respectively) when compared with other deuterostome and chordate nodes (Fig. 2 and Extended Data Fig. 3a). Furthermore, the fraction of highly retained novel gene families (also known as novel core genes, that is, genes that are not lost in descendant lineages and by convention indicated by ++) is the highest in the last common ancestors of vertebrates, gnathostomes and cyclostomes (novel core genes: ++81, ++86 and ++98, respectively; Fig. 2 and Extended Data Fig. 3a). These are notably larger than those observed in other major evolutionary episodes in metazoan evolution[31,33], but generally similar to a recent study using more chondrichthyan but only two invertebrate genomes[34], suggesting that the emergence of new gene families played important roles in the origin and diversification of early vertebrates. Gene Ontology (GO) enrichment analyses demonstrate that the origin of vertebrates was characterized by the appearance of genes involved in signalling pathways, cell communication and transcriptional regulation (Supplementary Tables 24 and 25), while novel core genes involved in immunity played an important role in the origin of gnathostomes (Supplementary Tables 24 and 26). Consistently, gnathostomes and cyclostomes convergently evolved independent adaptive immune systems, based on immunoglobulins in the former, and in variable lymphocyte receptors in cyclostomes[35] (Supplementary Fig. 6 and Supplementary Table 27). The largest fraction of gene losses occurred in the ancestral cyclostome lineage (Fig. 2), suggesting that a strong asymmetric reduction of gene complements accompanied the early evolution of the group. For instance, the hagfish genome lacks several vision and circadian rhythm-related genes, probably associated with its vestigial eyes (Supplementary Fig. 7 and Supplementary Table 28). Inferred rates of gene duplication (irrespective of the duplication mechanism) across Metazoa identify widespread duplications associated with the vertebrate and teleost stem-lineages (Extended Data Fig. 2b,c), probably reflecting the 1R, 2R and teleost 3R WGD events[36]. We also inferred high duplication rates in each of the lineages leading

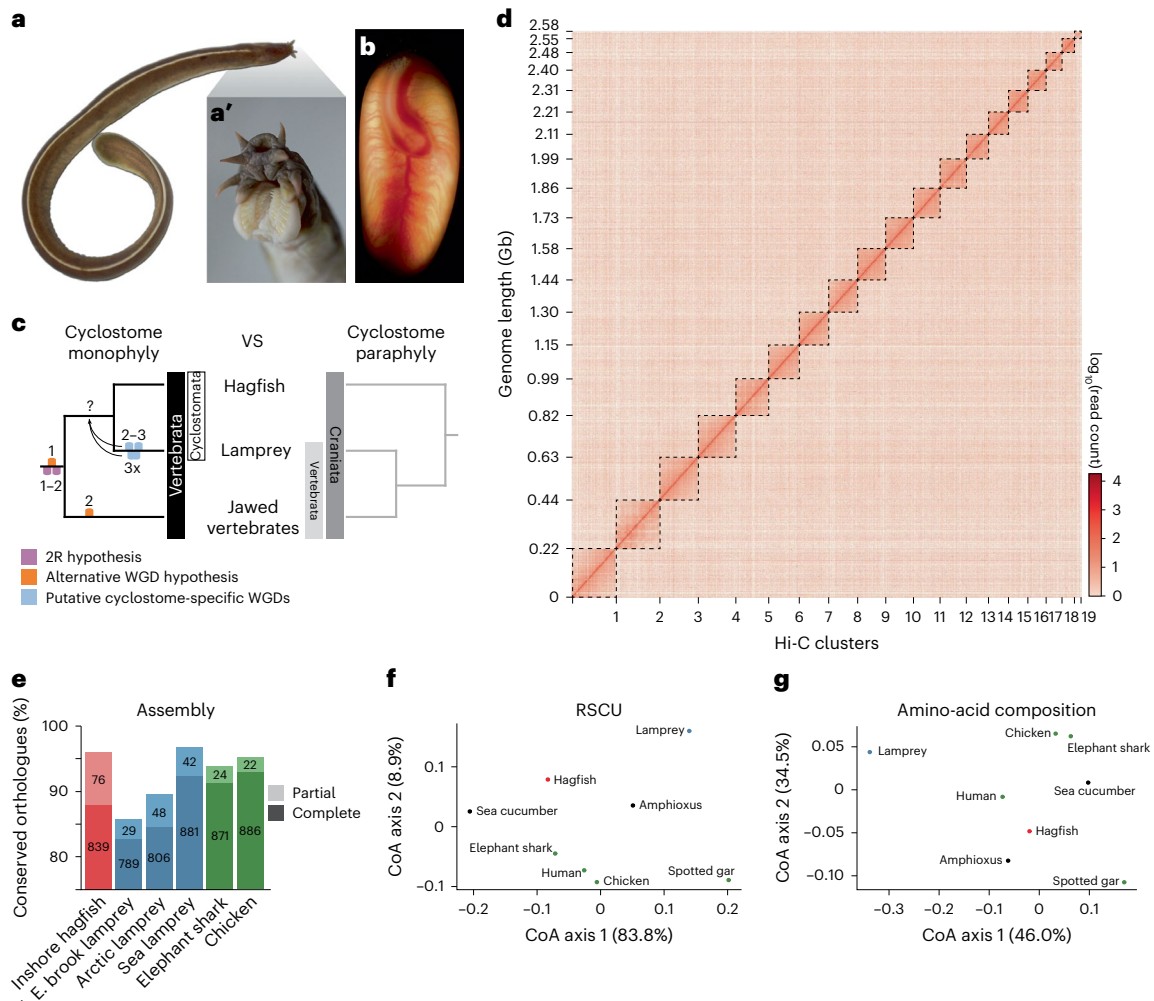

**Fig. 1 | Genome of the inshore hagfish, *E. burgeri*. a**, Dorsal view of a young adult of the inshore hagfish *E. burgeri*, with the head to the top right. The teeth apparatus (and not a jaw) can be observed in a magnification of the head region of a fixed adult individual (**a'**). **b**, Fertilized egg of *E. burgeri* with a developing embryo at stage Dean 53 (ref. 48). Blood vessels can be observed from the exterior. **c**, Two competing hypotheses of vertebrate phylogeny. WGD events corresponding to the 2R hypothesis (lilac), to an alternative vertebrate 2R hypothesis (orange) and to those recently proposed in the lamprey lineage (light blue) are marked. Whether the lamprey-specific events actually occurred in a stem cyclostome remains elusive. **d**, Hi-C contact heatmap of the corrected hagfish genome assembly ordered by cluster (chromosome) length. Dashed boxes indicate the cluster boundaries. **e**, Completeness assessment of the genome assembly of the inshore hagfish *E. burgeri* genome (red), three lamprey species (blue) and two jawed vertebrates (green). Number of conserved metazoan orthologues (metazoa_odb10 dataset, containing 954 BUSCOs) is indicated for each case. F. E., Far Eastern. **f**, Correspondence analysis (CoA) on RSCU values was performed using the nucleotide sequences of all predicted genes concatenated for individual species. The percentage of variance is indicated for each axis. **g**, CoA of amino acid composition, with the percentage of variance indicated for each axis. In **f** and **g**: red, hagfish; blue, lamprey; green, jawed vertebrates; black, invertebrates.

towards crown-gnathostomes and lampreys (Extended Data Fig. 2b,c) which might suggest large-scale duplications associated with these groups, consistent with the WGD events proposed recently[12,18]. This type of analysis, however, cannot discriminate between WGD and other large-scale gene duplication mechanisms.

## Conserved *Hox* cluster evolution in cyclostomes

The number of *Hox* clusters and ancestral WGD events are usually correlated; hence, the former has been used as a genomic marker of the latter. The presence of 6 *Hox* clusters in lamprey genomes[17,18] has been interpreted to indicate the possibility that more than two WGD events occurred in this lineage[18]. We have extended previous observations in *E. burgeri*[19] to confirm the presence of 40 *Hox* genes arranged in 6 complete *Hox* clusters (Fig. 3). Two of the hagfish clusters are located in the same chromosome (cluster 3), separated by >80 Mb, probably the result of chromosomal shuffling due to the intense

reorganization of the hagfish genome from ancestral chromosomes (ACs; see below). Phylogenetic analyses of *Hox* coding sequences have long proven inconclusive to determine the orthology relationship between lamprey and hagfish *Hox* counterparts[19]. We thus applied a microsynteny conservation approach using extended *Hox* loci which, together with phylogenetic analyses of selected non-*Hox* syntenic genes, allowed us to establish clear one-to-one orthologous correspondences between hagfish and lamprey *Hox* clusters, named α to ζ after the lamprey clusters[18] (Fig. 3, Extended Data Fig. 4a–d, Supplementary Fig. 8 and Supplementary Tables 29 and 30). This suggests that the crown-cyclostome already possessed 6 *Hox* clusters, distinct from the ancestral crown-gnathostome, which possessed 4 clusters (Supplementary Fig. 9). This observation provides further evidence of cyclostome monophyly, by suggesting that lampreys and hagfish share a genome history exclusive of gnathostomes. This implies that the events suggested from the different analyses of the Arctic lamprey

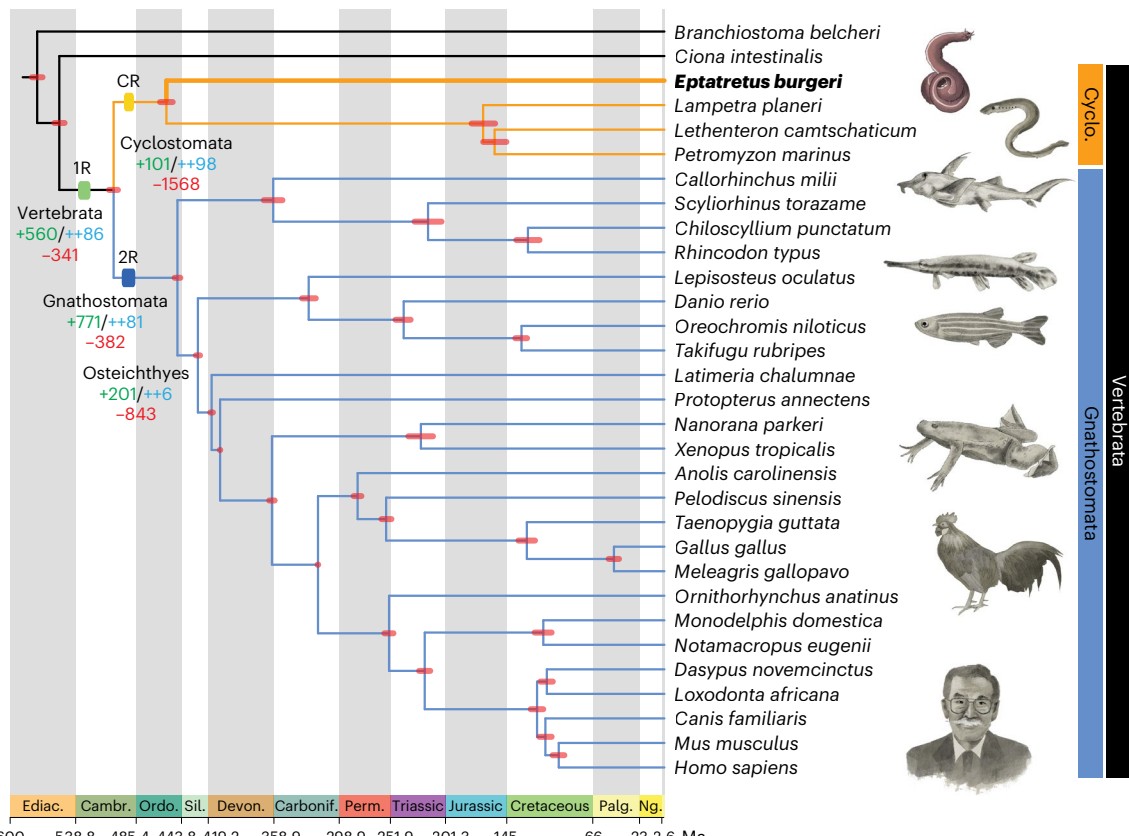

**Fig. 2 | Calibrated and dated vertebrate evolution.** Time-calibrated rooted phylogeny of vertebrates and two non-vertebrate species with 95% credibility intervals for clade divergence times indicated by red bars on nodes. The phylogenetic tree was obtained with Bayesian inference (Extended Data Fig. 2a) and all nodes were recovered with a posterior probability of 1. Numbers of gene family gains (green, novel homology group (HG); blue, novel core HG) and losses (red) are indicated in selected nodes (see text). Dated WGD events, including 1R, 2R and cyclostome-specific event (CR) described in this study, are indicated with coloured rectangles. The hagfish position is highlighted with a thickened line and bold font. Geological periods are colour-coded at the bottom: Ediac., Ediacaran; Cambr., Cambrian; Ordo., Ordovician; Sil., Silurian; Devon., Devonian; Carbonif., Carboniferous; Perm., Permian; Palg., Paleogene; Ng., Neogene. Animal illustrations kindly provided by Tamara de Dios Fernández; human, zebrafish, lamprey and hagfish illustrations reproduced with permission from ref. 133.

genome, two extra WGDs[18] or a triplication[11], might have occurred in early cyclostome evolution, probably before the lamprey and hagfish divergence[11,19].

## Ancestral vertebrate karyotype

The reconstruction of the pre-WGD vertebrate proto-karyotype by means of macrosynteny analysis stands as the most robust approach to test the 2R event and its phylogenetic position[37]. Earlier attempts at reconstructing the ancestral vertebrate karyotype have yielded widely disparate outcomes, indicating 10–13 (refs. 14,16) or 17–18 (refs. 8,11–13,38) ancestral pre-duplicative chromosomes. These reconstructions have also unveiled a perplexing scenario where lampreys' divergence from gnathostomes occurred either before or following the 2R event[8,11,12,18]. To shed new light on early vertebrate genome evolution, we performed a macrosynteny conservation analysis between gnathostomes, cyclostomes and selected invertebrate deuterostomes. First, to minimize noise from lineage-specific fusion and fission events, we reconstructed ancestral chicken[39] and spotted gar[40] genomes using elephant shark[41] as an outgroup, obtaining an almost perfect one-to-one chromosome orthology (Supplementary Table 31). Next, to infer the ancestral vertebrate karyotype, we elaborated a map of homology relationships between the genes of these slow-evolving gnathostome genomes with the chromosome-level genome assembly of the sea cucumber *Apostichopus japonicus* (echinoderm) as a pre-duplicative outgroup species[42]. With this, we inferred a proto-vertebrate karyotype of 17 ACs (Supplementary Fig. 15 and Supplementary Table 32).

We mapped genes from the Belcher's lancelet (*Branchiostoma belcheri*) genome[43] to each AC (Methods and Supplementary Fig. 21) using very stringent criteria, requiring homology relationships of an amphioxus gene with both a sea cucumber gene and several chicken and/or spotted gar genes, and all anchored to the same AC. In total, we mapped 5,065 Belcher's lancelet genes to AC1–17 (ranging from 115 to 534 genes in AC1–AC16; AC17 consisted of only 20 genes and was thus excluded from several subsequent analyses; Supplementary Table 34). With these in hand, we corroborated our ancestral vertebrate karyotype reconstruction through comparisons with the chromosome-scale genome of the amphioxus *Branchiostoma floridae*[12] (Supplementary Tables 35 and 36).

Our inference of an ancestral karyotype with 17 ACs matches a previous study[8], has minor differences with other 17-chromosome inferences[12,13] and depicts one less chromosome than more recent studies[11,38] (Supplementary Tables 37 and 38). In our model, 4 of the 17 ACs (AC1, AC2, AC3 and AC6) each correspond to 2 or 3 linkage groups (putative chromosomes) of the sea cucumber genome (Supplementary Fig. 15 and Supplementary Table 32) as well as to distinct homologous chromosomes in *B. floridae* (Supplementary Table 36), suggesting that these 4 ACs probably originated via fusions of ancestral chordate chromosomes in the vertebrate lineage before 1R[38]. The difference from previous 18-chromosome models is that while we consider that the vertebrate AC3 is a single chromosome resulting from a pre-1R fusion event of two ancestral chordate chromosomes (Supplementary Fig. 23a), others[11,12,38] consider that these two chromosomes

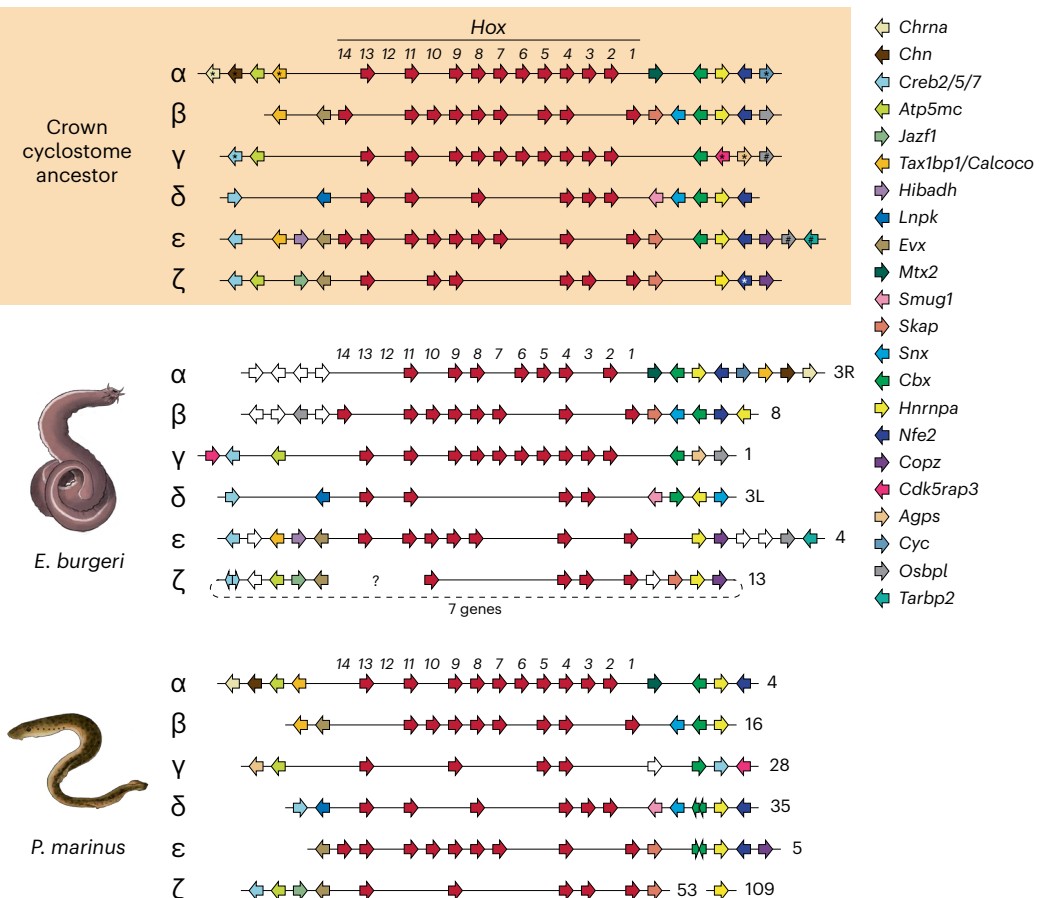

**Fig. 3 | Reconstruction of the *Hox* complement of an ancestral cyclostome.** Schematic representations of *Hox* clusters and syntenic genes of the inshore hagfish (*E. burgeri*, middle), the sea lamprey (*P. marinus*, bottom) and a reconstruction of the complement of the last common ancestor of hagfishes and lampreys (top). Genes are represented by colour-coded arrows whose direction marks the sense of transcription: *Hox* genes in red, non-*Hox* genes coloured by homology (legend at right). The block between *Evx-ζ* and *Creb2/5/7-ζ* is assembled downstream of the *Hox-ζ* cluster, separated by 7 genes. This might be a misassembly, and their 'natural' upstream position is marked by a dashed line. The sea lamprey genome scaffolds and hagfish Hi-C clusters in which *Hox* clusters are located are indicated to the right of each cluster, with cluster 3 separated into 3L (0–107.78 Mb) and 3R (107.78–194 Mb). Black asterisks mark genes placed at opposite sides of a cluster in the hagfish and the lamprey, but placed at one side of the ancestrally reconstructed cluster on the basis of comparisons with gnathostomes (Supplementary Fig. 9); hashes denote genes present in lampreys in the same chromosome but at a long distance (see Supplementary Fig. 8); white asterisk denotes *Nfe2-ζ* inferred due to its presence in the Arctic lamprey. Animal illustrations kindly provided by Tamara de Dios Fernández; lamprey and hagfish illustrations reproduced with permission from ref. 133.

remained separate through 1R (Nakatani's Pvc8 and 9, or Simakov's CLGQ and CLGI, respectively). While Pvc8/CLGQ and Pvc9/CLGI are consistently co-located in gnathostome chromosomes, they remain separate in invertebrate karyotypes[11,12,38]. We did not find any signals of linkage between Pvc8/CLGQ and Pvc9/CLGI in the lamprey and the hagfish genomes (Supplementary Figs. 25 and 26). Therefore, there exist two alternative scenarios: (1) the 18-chromosome model implies that two independent pairwise fusions occurred after 1R in a stem gnathostome, mimicking a single pre-1R fusion event (Supplementary Fig. 23a); and (2) our 17-chromosome model requires symmetric fissions of two AC3-derived post-1R chromosomes occurring in an ancestral cyclostome (Supplementary Fig. 23b). Although in silico simulations show that a scenario of pairwise post-1R fusions would not be extremely rare (30% of cases expected by chance; Supplementary Table 40 and Methods), we believe the pairwise fissions to be more plausible given the higher level of reorganization found in cyclostome karyotypes (see next section). Altogether, while we propose a scenario involving 17 ancestral vertebrate chromosomes, a scenario with 18 chromosomes[11,38] is also possible.

Importantly, all ACs correspond to sets of four (11/17) or three (6/17) paralogous chromosomes in the gnathostomes chicken and gar,

a strong genome-wide pattern consistent with 2R[2,11–13]. We stringently selected 701 sets of orthologous genes (Methods and Supplementary Table 39) between the sea cucumber, chicken, spotted gar and an AC gene (from *B. belcheri*), and built robust chromosome-level phylogenies with a median of 38 concatenated gene sets across each of the ACs (Extended Data Fig. 5a and Supplementary Table 39). The highly supported, clear-cut topologies further support the existence of 2R in gnathostomes and depict the exact evolutionary trajectory from each AC to their modern chicken and spotted gar descendants (Extended Data Fig. 5a,b). Our reconstruction of gnathostome karyotype evolution involved 8 fusion events that took place after 1R but before 2R (Extended Data Fig. 5b), similarly to what has been previously found by others[8,11,12] (Supplementary Table 41). Furthermore, we found a significant gene retention asymmetry after 2R, with a median of 1:2.28 genes per ohnologous (duplicates that originate through WGD, after ref. 3) chromosome pair, but not after 1R (median 1.16; $P = 3.4 \times 10^{-7}$, Wilcoxon rank-sum test; Extended Data Fig. 5b). This pattern is consistent with previous studies suggesting that 1R was an autotetraploidization event and 2R an allotetraploidization event[11,12] (but see ref. 44 on asymmetric gene retention after teleost 3R autotetraploidy).

## Gnathostomes and cyclostomes share 1R but not 2R

We next tested hypotheses of WGD timing relative to cyclostome divergence. We assessed the phylogenetic signal of hagfish and lamprey genes anchored to 661 orthologous gene sets (Supplementary Table 42), including elephant shark orthologues as a control for the 2R signal and amphioxus genes as outgroups. Approximately 73.2%, 79.1% and 75.7% of trees including hagfish, lamprey or both hagfish and lamprey orthologous genes, respectively, are compatible with shared 1R (Fig. 4a). However, while 99.5% of elephant shark gene tree topologies are 2R-compatible, only 19.1% of hagfish, 10.6% of lamprey and 8.2% of cyclostome (including both lamprey and hagfish) gene trees are compatible with a 2R history (Fig. 4a and Supplementary Files 5–8). Thus, we find strong support only for 1R as shared among cyclostomes and gnathostomes.

To further confirm the timing of the 1R event, we investigated whether signals of the inferred four pre-1R and eight post-1R fusion events are present in cyclostomes. When assessing how hagfish and lamprey chromosomes descended from the 17 ACs, we found that the hagfish genome displays a large amount of rearrangement (at least 52 fusions detected), making any signal of hypothetically shared events unreliable (Extended Data Fig. 6b). However, most lamprey chromosomes are descendants of single ACs[12] (Extended Data Fig. 6a), making the lamprey a better model to investigate these rare genomic changes. We found that the sea lamprey genome[17,45] bears signals of three (AC1, AC2 and AC6) and the hagfish genome of two (AC1 and AC2) of the four pre-1R fusions (Supplementary Figs. 25 and 26). On the other hand, similar to previous studies[11,12] we did not find any reliable signal of the eight post-1R fusions detected in gnathostomes, suggesting that the lamprey and hagfish diverged after the 1R but before all eight post-1R/pre-2R fusions[11,12]. Taken together, our comprehensive phylogenetic analysis and the constraints given by pre- and post-1R chromosomal fusions provide strong evidence in favour of a pan-vertebrate 1R event, but constrains 2R to the gnathostome lineage as recently suggested in similar analyses[11,12].

## Cyclostome-specific whole-genome triplication

It has been suggested that the lamprey genome has been shaped by either three duplicative events[18] or a hexaploidization[11]. The presence of six orthologous *Hox* clusters in both the lamprey and the hagfish (Fig. 3) implies that this is the ancestral condition for cyclostomes and supports the triplication event[11,19]. Although we find that multiple chromosomes and large chromosomal sections are descendant copies of each AC in both cyclostome groups, the extensive rearrangements observed in the hagfish and the large haploid number in the lamprey impede chromosome-level macrosynteny conservation analysis to distinguish intraspecific ohnologous and interspecific orthologous relationships. To confidently infer karyotype evolution in cyclostomes, we developed a new metric, the 'overlapping ratio' (OR), to measure the similarity of gene retention profiles of any two chromosomes hypothetically descending from a common AC (Fig. 4b and Supplementary Fig. 29; gene-poor AC17 was excluded from this analysis, which required at least 20 genes retained in each descendant chromosome). A retention profile is defined by a vector listing the presence or absence of genes on a modern vertebrate chromosome from their corresponding AC. Therefore, we expect the OR of chromosomes deriving from a duplication event to be significantly higher than that of chromosomes deriving from an ancestral fission followed by gene translocations. As proof of concept, we applied this metric to gnathostomes: knowing that their genomes have been shaped by the 2R event, we found that the median OR of ohnologous chromosome pairs in chicken or spotted gar was 0.49 (interquartile range, IQR: 0.44–0.56) and 0.54 (IQR: 0.47–0.65), respectively (Fig. 4c), while OR value for simulated fission-derived chromosome pairs was never larger than 0.15, indicating that ohnologous chromosomes indeed share more retained genes (Fig. 4c and Methods).

We then applied the OR metric to the sea lamprey (after correcting misassemblies, assisted by a meiotic map of the Pacific lamprey *Entosphenus tridentatus*[17] and confirmed by the recent chromosome-level genome assembly[45]) (Supplementary Table 50 and Supplementary Fig. 24) and the hagfish, defining ohnologous chromosome pairs as those with OR > 0.15. We found that the median OR between putative ohnologous chromosomes was 0.30 (IQR: 0.23–0.36) and 0.29 (IQR: 0.23–0.37) for the lamprey and hagfish, respectively (Fig. 4d, Extended Data Fig. 7a,b and Supplementary File 9). Using this, we found that most ACs analysed (12/16 or 75%) have descended into three or more mutually ohnologous chromosomes in both the lamprey and hagfish (Fig. 4f), suggesting that at least a second WGD might have occurred in cyclostomes. In both genomes, at least five chromosomal regions are direct descendants of each of the same five ACs (1, 2, 6, 10 and 14), with 3 ACs contributing to 6 chromosomes each in the lamprey (Fig. 4f). We do not find more than 6 descendant copies from any AC, supporting a whole-genome triplication in the cyclostome lineage as previously proposed in the analysis of the lamprey genome[11]. The distribution of multiplicity across ACs is highly correlated across the two species (Spearman $\rho$ = 0.91), suggesting that this triplication event is conserved between the lamprey and hagfish and thus occurred in an ancestral cyclostome. It is expected that OR will decrease with each WGD (Supplementary Fig. 29), hence the lower value in cyclostomes is consistent with the occurrence of this larger polyploidy event. While our data do not definitively rule out the possibility of two cyclostome-specific WGD events followed by extensive chromosome losses, this scenario is less plausible than a single triplication event, particularly given the absence of instances with eight copies of any chromosomal region in the lamprey or the hagfish.

To further confirm that this proposed triplication event is shared by hagfish and lamprey, we extended the use of the OR to detect putative orthologous chromosomes. During the process of diploidization after a polyploidy, two descendant chromosomes diverge and fix their mutations independently, hence it is expected that interspecific orthologous chromosomes will have more similar gene retention profiles than intraspecific ohnologous chromosomes (as long as rediploidization precedes speciation[46]). Accordingly, orthologous chromosomes of chicken and spotted gar have a median OR = 0.96 (IQR 0.95–0.98; Fig. 4e and Supplementary Table 44) and clustering-based analysis based on gene retention profiles places chicken and gar orthologous chromosomes closer to each other, completely reflecting the phylogenetic signal (Extended Data Fig. 7c and Supplementary File 10). When we applied this approach to cyclostome genomes, we found the median OR = 0.84 (IQR: 0.74–0.91; Fig. 4e and Supplementary Table 45) for 52 (~87%) chromosome pairings between lamprey and hagfish that putatively represent 1:1 orthologues (higher than that of ohnologous chromosomes; Supplementary Table 45) and only 8 (~13%) one-to-two or two-to-one ambiguous relationships, probably due to secondary independent chromosome losses in either group. Clustering analysis of retention profiles recovers orthologous relationships between lamprey and hagfish (Fig. 4g and Supplementary File 11). Overall, intra- and interspecific gene retention profile analyses indicate that a triplication event took place in the cyclostome stem-lineage; we refer to this as CR, to avoid confusion with the gnathostome-specific 2R event.

## Increase of developmental regulatory complexity

To investigate the immediate consequences of the independent CR event on cyclostome genome evolution, we first asked whether retained duplicates (ohnologues) are especially associated with developmental functions in the hagfish as in their gnathostome counterparts[13,47]. GO enrichment analysis shows that hagfish gene ohnologues are also significantly enriched for functions associated with developmental processes (Extended Data Fig. 8a,b). Gnathostomes have increased their regulatory complexity (higher number of regulatory regions per gene), particularly of developmental ohnologues[47]. We identified accessible

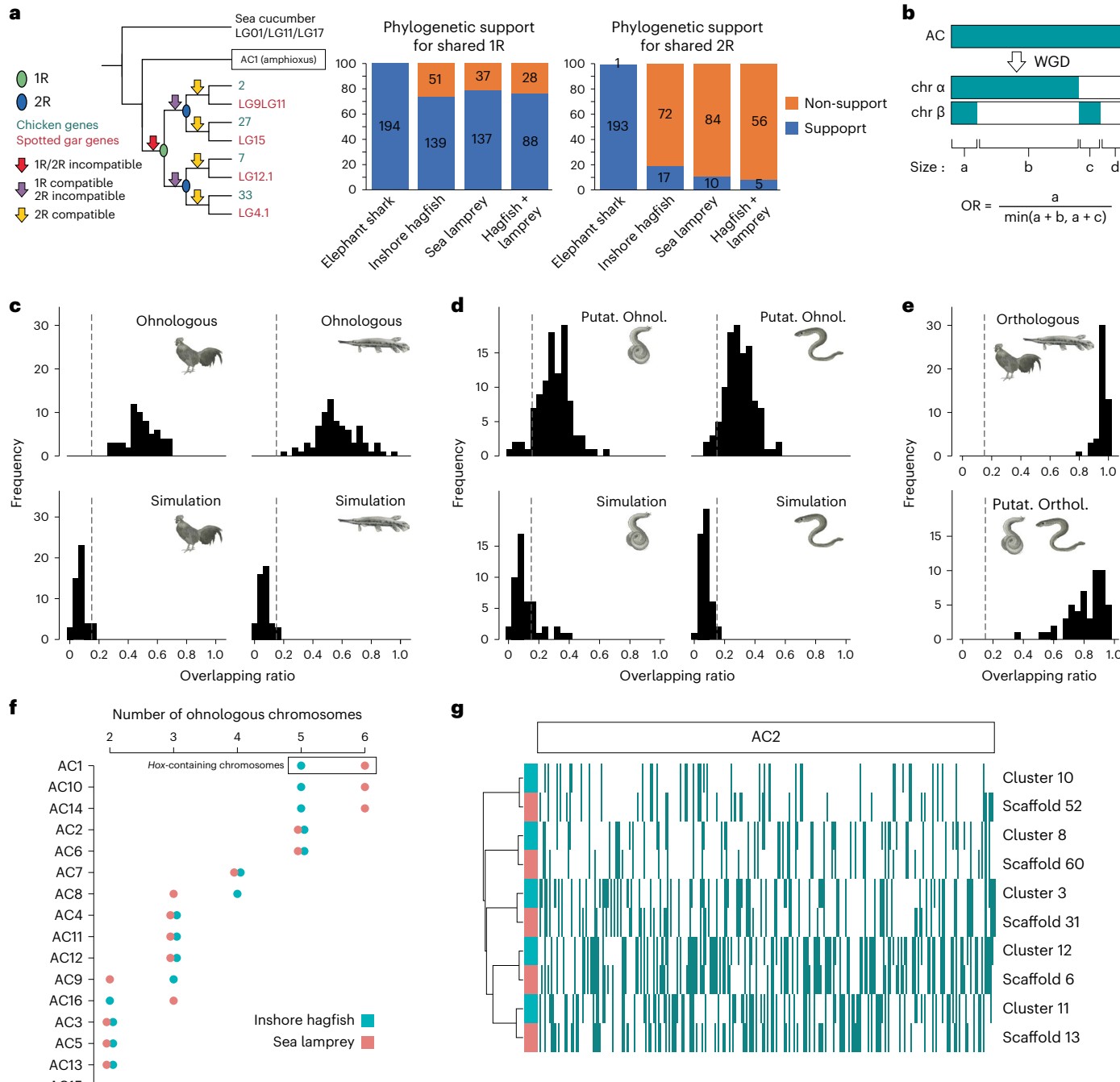

**Fig. 4 | Hagfish and lamprey share a whole-genome triplication.**
**a**, Phylogenetic support of gnathostome and cyclostome genes for 1R and 2R. Elephant shark, hagfish, lamprey or both cyclostomes' genes (both hagfish and lamprey genes included) were analysed as test genes in the context of spotted gar and chicken gene phylogenies by each AC (using amphioxus genes) and orthologous sea cucumber genes (outgroup). Left: possible positions where test genes can branch, supporting or not 1R or/and 2R (see legend). Middle and right: statistics of supporting (blue) or not supporting (orange) gene phylogenies from each species' tested genes. All phylogenetic trees are available in Supplementary Files 5–8. **b**, Formula to calculate the OR between two chromosomes. Dark cyan denotes genes from the AC, retained in modern chromosomes; white indicates gene loss. **c**, OR values distribution between WGD-generated paralogous (ohnologous) chromosomes in chicken (top left) and spotted gar (top right), and the artificially split chromosomes in chicken (bottom left) and spotted gar

(bottom right). Dashed lines mark OR = 0.15. **d**, OR values distribution between putative ohnologous chromosomes in hagfish (top left) and lamprey (top right), and the artificially split chromosomes in hagfish (bottom left) and lamprey (bottom right). **e**, OR values distribution between chicken and spotted gar (top) and between hagfish and lamprey (bottom) orthologous chromosomes. **f**, Numbers of mutually ohnologous chromosomes in cyclostome genomes that correspond to each one of the 16 reconstructed ACs. **g**, Retention profile clustering analysis of cyclostome chromosomes deriving from AC2. Retained genes are denoted by dark cyan lines. Five putative orthologous chromosome pairs are defined. Note that AC17 was excluded from the analyses depicted in **c**–**f** because of the low number of genes we recovered (20 genes). Animal illustrations kindly provided by Tamara de Dios Fernández; chicken, spotted gar, lamprey and hagfish illustrations reproduced with permission from ref. 133.

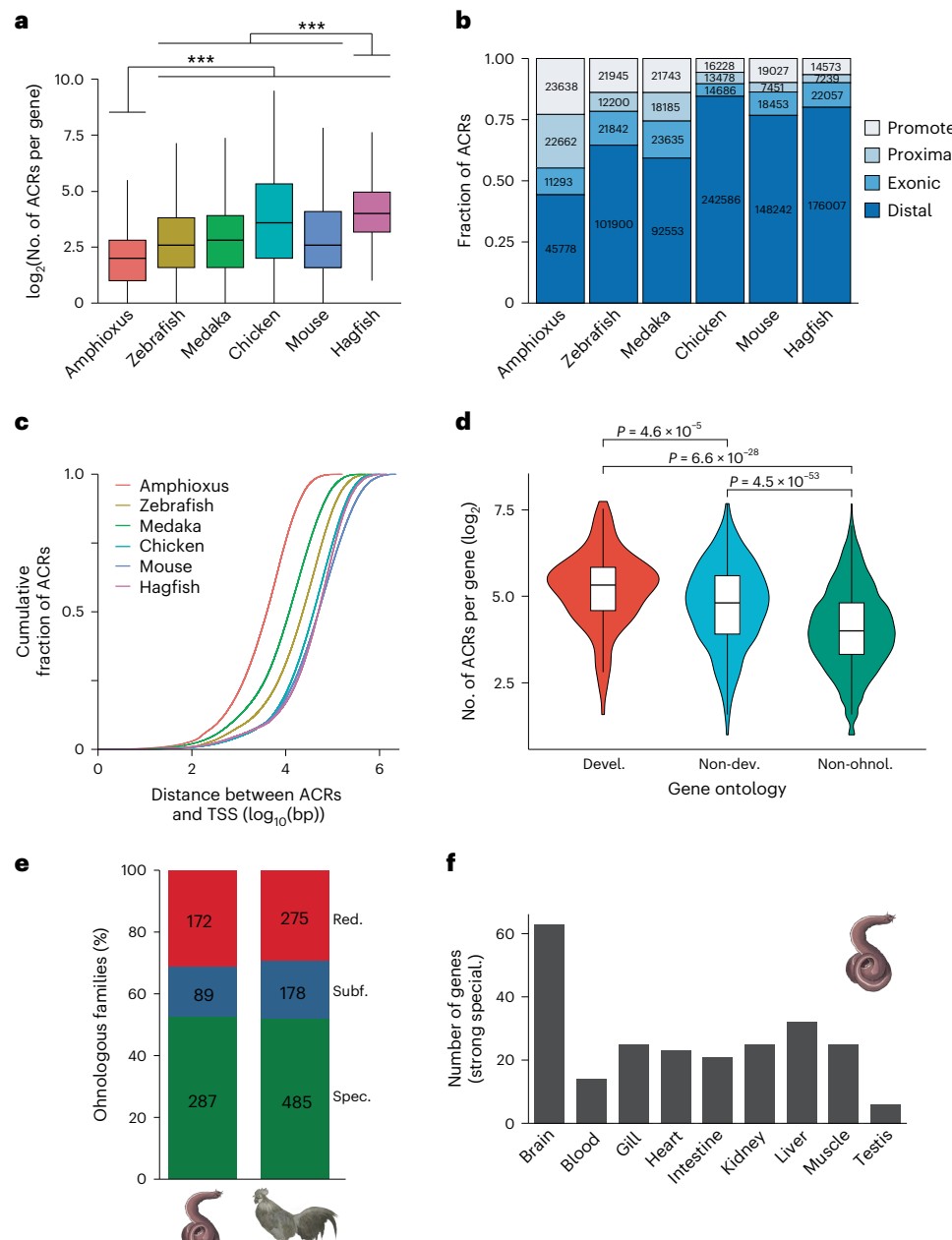

**Fig. 5 | Impact of WGD events on the regulatory genome. a**, Distributions of the ACR numbers within the *cis*-regulatory regions of each gene (see Methods). $n$ = 28,497 (amphioxus), $n$ = 23,183 (zebrafish), $n$ = 22,184 (medaka), $n$ = 15,213 (chicken), $n$ = 23,256 (mouse) and $n$ = 16,951 (hagfish) genes. ***$P < 2.2 \times 10^{-16}$, Bonferroni-adjusted, two-sided Wilcoxon rank-sum tests. **b**, Numbers and fractions of ACRs with respect to genomic annotations in each species. Promoters, between 1 kb upstream and 0.5 kb downstream of annotated transcription start sites (TSSs); proximal, within 5 kb upstream and 1 kb downstream of annotated TSSs, but not overlapping promoters; exonic, within exons of protein-coding genes but not overlapping proximal regions; distal, not in aforementioned locations. **c**, Cumulative proportion of the distance of ACRs from the closest TSSs in each species. For the result with scaling based on the average length of intergenic regions of each species genome, see Extended Data Fig. 8c. **d**, The distribution of ACR numbers across different classes of

genes, according to PANTHER Gene Ontology database (devel., developmental ohnologues; non-dev., non-developmental ohnologues; non-ohnol., singletons). $n$ = 143 (devel. ohnol.), $n$ = 816 (non-devel. ohnol.) and $n$ = 7,303 (non-ohnol.) genes. $P$ values from Bonferroni-adjusted two-sided Wilcoxon rank-sum tests are indicated. **e**, Distribution of fates of ohnologous families after WGD. Red., potential redundancy; Subf., potential subfunctionalization; Spec., potential specialization. **f**, Number of ohnologues with strong specialization expressed in hagfish tissues. In **a** and **d**, boxes correspond to the median (centre line) and the first and third quartiles. Whiskers extend to the last point no further than 1.5× the interquartile range from the first and third quartiles. For **a**–**d**, see Supplementary Tables 52–57 for detailed statistical information, including $P$ value for each pairwise comparison. Animal illustrations kindly provided by Tamara de Dios Fernández; chicken and hagfish illustrations reproduced with permission from ref. 133.

chromatin regions (ACRs) as putative non-coding regulatory elements in the hagfish genome with an assay for transposase-accessible chromatin coupled to sequencing (ATAC-seq), using a total of two embryos of *E. burgeri*, each at a different stage[48] (45 and 53; Supplementary Fig. 32). We found a significantly higher number of ACRs per gene

than in the cephalochordate amphioxus, similar to what has been observed in gnathostomes[47] (Fig. 5a), particularly in distal regions from transcriptional start sites (Fig. 5b,c and Extended Data Fig. 8c). This pattern is especially evident in developmental genes (Fig. 5d and Extended Data Fig. 8d–f), implying that their higher retention after a

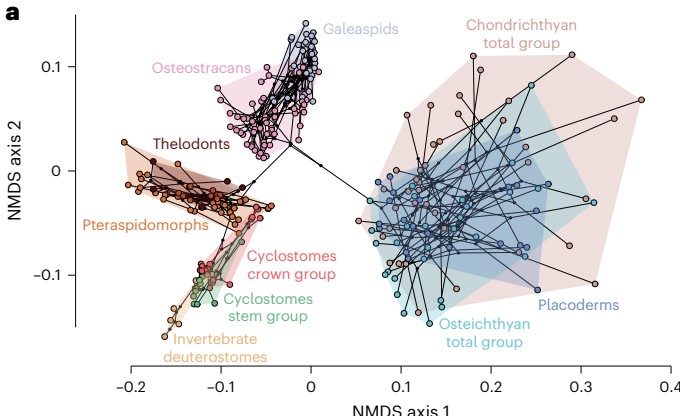

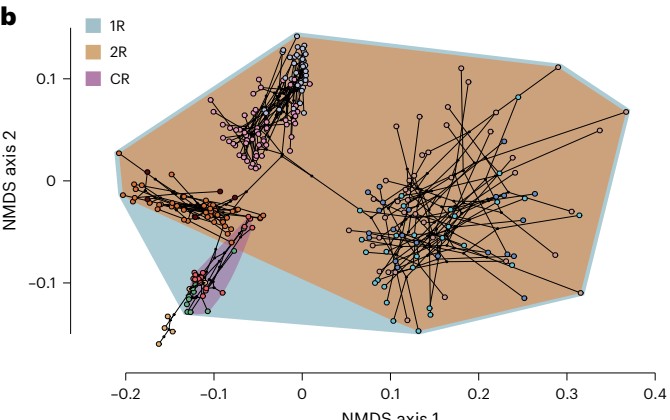

**Fig. 6 | Morphological evolution of vertebrates. a,b,** Morphological disparity across vertebrates. Non-metric ordinations are presented, highlighting the morphological variance among (**a**) taxonomic lineages of extant and extinct vertebrates and (**b**) the descendants of 3 whole-genome duplication events. Convex hulls have been fitted around groups. The underlying tree was derived from a consensus of relationships from the literature.

cyclostome CR event is underlain by a more complex regulatory landscape of developmental genes, as in gnathostomes[47].

In gnathostomes, retained duplicates can evolve via expressional specialization (reduction of expression domains of one of the ohnologues)[47], probably coupled to neofunctionalization rather than subfunctionalization (differential erosion of enhancers)[49]. Taking advantage of adult transcriptome data across nine organs (see Methods), we next analysed the putative fates of hagfish ohnologues after CR (Supplementary Fig. 31). Hagfish duplicates also tend to reduce their expressional domains: over 68% and 71% of gene families subfunctionalized or specialized in the hagfish and chicken, respectively (Fig. 5e). Hagfish ohnologues that have potentially restricted their expression domains (subfunctionalization or specialization) are associated with a larger amount of regulatory elements and a higher sequence evolutionary rate than those that have maintained the ancestral patterns (Extended Data Fig. 8g,h), similar to gnathostomes (Extended Data Fig. 8i)[47]. Furthermore, the largest portion of ohnologues with strong specialization (one or two ancestral expression domains) are expressed in the brain (Fig. 5f), mirroring the pattern observed in gnathostomes[47] (Extended Data Fig. 8j). In summary, our results indicate that cyclostomes and gnathostomes followed parallel evolutionary pathways after their independent WGD events. Genes gained a larger regulatory complexity, mostly on distal regions and especially in duplicates with developmental functions, which tend to be retained more often. Furthermore, specialization is a common fate of ohnologues associated with faster sequence evolution and the acquisition of novel regulatory

elements that drive their tissue-specific expression. Alternatively, the possibility that a decrease in the number of regulatory elements took place in the amphioxus cannot be confidently ruled out.

## Impact of WGD events on vertebrate morphological diversity

Hypotheses on the role of WGD events in the origin and elaboration of the vertebrate bodyplan range from deterministic to permissive[4]. There can be no doubt that many vertebrate and gnathostome novelties are contingent on gene paralogues that are the product of the 1R and 2R events, although whether WGD played a causal role remains unclear. We employed two tests of a causal relationship: (1) absolute timing of the WGD events and the clades with which they are causally associated and (2) contrast in morphological phenotypic diversity before and after the WGD events. Using a dataset of 177 genes and 33 fossil calibrations, we provide estimated times at which duplicated subgenomes diverged. We adopted a sequential Bayesian approach in which the posterior clade age estimates from our species timescale were used as prior on the speciation nodes in our concatenated gene tree; to achieve statistical consistency, this requires that the same molecular loci are not used in the two analyses[50]. A parallel analysis in which the concatenated gene tree was calibrated using the original fossil calibrations yielded results that are less precise but otherwise not materially different. Interpretation of the results depends on the nature of the ploidy event; in the case of autopolyploidy, we estimate the minimum timing of rediploidization, when the two subgenomes derived from WGD stopped homologously recombining, a process that can be asynchronous and span dozens of millions of years[44,46,51–54]. In the case of allopolyploidy, the age estimate represents the speciation event that isolated each of the 2 subgenomes that later came together to form an allopolyploid genome[55]. Our relaxed molecular clock analyses estimate the 1R event to have occurred 535.3–524.8 Ma (early Cambrian), 14.3–29.2 Myr before the divergence of crown-vertebrates (510.5–506.1 Ma; middle Cambrian) (Fig. 2); the CR event is dated to 500–492 Ma (late Cambrian), 23.5–36.5 Myr before the divergence of crown-cyclostomes (468.5–463.5 Ma; Middle Ordovician) (Fig. 2); and the 2R event is dated to 498.4–485.2 Ma (late Cambrian–earliest Ordovician), 35.1–53.3 Myr before the divergence of crown-gnathostomes (450.1–445.1 Ma; Late Ordovician) (Fig. 2).

To characterize morphological disparity across WGD events, we compiled a phenotype matrix composed of 577 traits for 278 living and fossil chordates encompassing all aspects of morphology, which we subjected to pairwise distance analysis followed by ordination using non-metric multidimensional scaling (NMDS) (Fig. 6a). This multivariate approach groups organisms with similar suites of characteristics while separating organisms with dissimilar traits, providing us with a relative measure of bodyplan diversity. The results of this analysis show that each genome duplication is followed by an increase in morphological disparity through occupation of novel regions of morphospace (Fig. 6b), but the majority of chordate disparity (88–97% of the morphospace encompassed by a vertebrate convex hull) emerged subsequent to the 2R event (Fig. 6b). Thus, while 2R and CR are of comparable antiquity, there is a stark contrast in terms of bodyplan evolution and species diversity between the descendants of 2R and the other WGD events.

## Discussion

The sequencing of a hagfish genome has enabled us to better understand the evolution of early vertebrates. First, our analysis of vertebrate genomes, including the hagfish, now establishes a robust and accurate history of WGD events in early vertebrates, corroborating the idea that cyclostomes diverged from gnathostomes after the 1R but before the 2R event. This is consistent with early[9,10,13] and recent studies on the matter that included the lamprey[11,12]. We think that debate over the timing of 2R[5] can now be concluded. Our hagfish genome also confirms an additional genome-wide duplicative event in stem-cyclostomes,

CR, which most probably was a triplication, as previously suggested[11]. Thus, key vertebrate innovations (for example, elaborate tripartite brain, neural crest cell-derived tissues among other novelties[56]) originated in a stem-vertebrate. However, at this point we cannot reliably establish whether these innovations pre- or postdate the 1R event. This basic vertebrate bodyplan was further elaborated independently in cyclostomes and gnathostomes as a result of their lineage-specific genome duplications, for instance, facilitating the evolution of different adaptive immune systems (immunoglobulin-based in jawed vertebrates, variable lymphocyte receptor-based in cyclostomes[35]), or the appearance of key morphological innovations, such as the jaw and paired appendages in gnathostomes. Interestingly, these independent WGD events shaped their ancestral genomes in similar ways by permitting an increase in regulatory complexity, especially of genes with roles in development. Duplicates of developmental genes are indeed more likely to be retained in both lineages, highlighting the crucial role of development in evolution of novel complex traits.

The contrasting morphological phenotypic consequences of 2R versus the other WGD events might suggest that there is no direct causal relationship or that there should be no general expectation of macroevolutionary consequences from WGD events despite their clear impact in increasing the regulatory potential of the genome. Another possibility is that the 2R event is different in nature from the 1R and CR events. Indeed, a number of recent studies together with our own analyses (Extended Data Fig. 5b) have suggested that while 1R was probably an autopolyploidy event, 2R was an allopolyploidy[11,12,57]. This is significant since it impacts our interpretation of the absolute timing of 2R, with the age estimate reflecting the divergence of the two lineages that later hybridized, not the allopolyploidy event itself[55]. Thus, the event occurred later than 498.4–485.2 Ma (late Cambrian), potentially coinciding with or even postdating the acquisition of gnathostome novelties that accrued among ostracoderms[4,58] before the divergence of crown-gnathostomes (which definitively postdated 2R) at 450.1–445.1 Ma (Late Ordovician). By the same token, the macroevolutionary consequences of allopolyploidy are expected to be more immediate than those of autopolyploidy, resulting in chromosomal rearrangements, changes in chromatin structure, DNA methylation, gene expression and the activation of transposable elements[59–61], extensive and immediate changes that promote species, and ecological diversification[62,63] as well as evolutionary novelty[64–67]. This may go some way to explain why the evolutionary consequences of the 2R WGD are so much greater, leading to the profound diversification of gnathostome bodyplans that have dominated vertebrate communities since the early Palaeozoic.

## Methods

No statistical methods were used to predetermine sample size. The experiments were not randomized and investigators were not blinded to allocation during experiments and outcome assessment.

### Animal sampling and experimentation

Adult inshore hagfish animals were captured off the coast of Shimane, Japan, as previously described[68]. Hagfish embryos (staged according to ref. 48) used for ATAC-seq were obtained as previously described[19,68]. The sampling and experiments were conducted according to institutional and national guidelines for animal ethics, approved by the RIKEN Animal Experiments Committee (approvals H14-25-23 and H14-25-25).

### Genome sequencing and assembly

We sequenced a mix of short-insert paired-end and long-insert mate pair libraries prepared from DNA extracted from the testis of a single, sexually mature male individual of the inshore hagfish, *E. burgeri*, resulting in ~240X of Illumina clean data (Supplementary Tables 2 and 3). Hagfish species have large genome sizes, ranging between ~2.2 and 4.5 Gb[23]. We estimated the genome of *E. burgeri* at 3.12 Gb on the basis

of *k*-mer frequency distribution (Extended Data Fig. 1a and Supplementary Table 4), in line with other hagfish species. We assembled the genome of *E. burgeri* following gradual steps using different strategies. First, we obtained a primary assembly using just the Illumina short-read data (v.2.0). To improve contiguity, this primary assembly was super-scaffolded using Chicago in vitro proximity ligation at Dovetail Genomics[22], significantly increasing the scaffold N50 (scaffolds equal to or longer than this value contain 50% of the assembly) from 0.44 to 2.69 Mb. This assembly was polished with all short-insert sequencing data using Pilon[69] v.1.22, and the resulting version (3.2 in our pipeline) was made publicly available in both NCBI (GenBank accession no. GCA_900186335.2) and Ensembl[70] (release 93; https://www.ensembl.org/Eptatretus_burgeri/). We further sequenced over 2200X of raw Hi-C short-read data from a second adult male individual and obtained ~350X valid Hi-C contact data to improve scaffolding. Hi-C contacts were also used to correct 280 likely misjoined scaffolds (Supplementary Table 7). After a process of parameter optimization, we used LACHESIS[71] to assemble 1,573 scaffolds into 19 Hi-C contact clusters.

### RNA sequencing

Adult tissues were dissected from two adult male individuals of *E. burgeri* (brain, gills, liver, intestine, heart, skeletal muscle, kidney and testis from animal #20150825; blood from animal #20150917). Total RNA was extracted using an RNeasy Plus Universal mini kit (QIAGEN) for the brain, heart, skeletal muscle, kidney and testis samples, and with ISOGEN (Nippon Gene), a guanidinium thiocyanate-phenol-chloroform-based extraction protocol, for the intestine, liver and gill samples. In all cases, DNA was removed including a DNaseI step. RNA-seq libraries were prepared with the TruSeq Stranded RNA Lib Prep kit (Illumina) and quantified by qPCR using the KAPA Library Quantification kit for Illumina Libraries (KapaBiosystems) for all samples. Library profiles were assessed with an Agilent 2100 Bioanalyzer. All libraries were sequenced at RIKEN BDR in an Illumina HiSeq 1500 platform, obtaining a total of ~650 M 127-bp paired-end strand-specific reads, with an average of ~54.5 M reads per tissue.

### Genome annotation

Annotation of the hagfish genome assembly v.3.2 was created via the Ensembl gene annotation system[24], assisted by RNA-seq data from 9 adult tissues (this study) and by developmental RNA-seq data from three embryos (Dean stages 35, 40 and 45) generated in a previous study[19]. Coordinates of annotated features were later converted to the final Hi-C assembly, v.4.0. Detailed methodology and annotation results can be found in Ensembl (http://www.ensembl.org/info/genome/genebuild/2018_06_eptatretus_burgeri_genebuild.pdf) and in Supplementary Information (section 1.2). In addition to the Ensembl pipeline, miRNA genes were further annotated using MirMachine[72] (v.0.1.2) and MirMiner[73] (v.1.0). Before performing phylogenetic analyses corresponding to Fig. 4a, 1,957 gene models were manually corrected, with the numbers ranging between 120 (amphioxus) and 704 (sea lamprey).

### GC content, codon usage and amino acid composition

Overall GC-content percentage was analysed for whole genomes of the inshore hagfish and 9 other chordate genomes (human, *Homo sapiens*; chicken, *Gallus gallus*; tropical clawed frog, *Xenopus tropicalis*; zebrafish, *Danio rerio*; spotted gar, *Lepisosteus oculatus*; elephant shark, *Callorhinchus milii*; sea lamprey, *Petromyzon marinus*; sea squirt, *Ciona robusta*; and the Floridian lancelet, *Branchiostoma floridae*) (Supplementary Table 12). GC-content distribution (Extended Data Fig. 1d) was calculated from non-overlapping sliding 10-kb windows. To calculate codon type frequency, we categorized each codon into GC-0/1/2/3 on the basis of the number of G or C bases in a codon. The summed frequency of usage for each category is the sum of the normalized frequency of codon usage for all codons included in each category.

To plot the distribution of GC content per codon position, the GC percentage of each codon position for each protein-coding gene (with only the longest coding sequence per gene) was calculated, as well as the GC content for each whole coding sequence (equivalent to the GC content of all three codon positions). RSCU calculates the relative synonymous codon usage on degenerative sites of third codon positions, which is independent from the amino acid usage. Therefore, RSCU was used as a robust measurement of GC bias in codons. Correspondence analysis of RSCU values was performed with codonW according to ref. 74.

### Completeness evaluation of genome and annotation
We used BUSCO[75] v.5.2.2 to assess the completeness of genomes at both assembly and annotation levels of hagfish (*E. burgeri*), three lamprey species (Far Eastern brook lamprey, *Lethenteron reissneri*[76]; sea lamprey, *P. marinus*[45]; and Arctic lamprey, *L. camtschaticum*[11]) and two jawed vertebrates (elephant shark, *C. milii*[11]; and chicken, *G. gallus* v.7.0, downloaded from Ensembl 109). The programme was run in both 'genome' and 'protein' modes, with gene predictor 'metaeuk' against the core metazoan database embedded in BUSCO (metazoa_odb10 dataset, built on 17 February 2021, with 954 BUSCOs).

### Species tree inference
Orthogroups of protein-coding genes previously used in the analysis of the spotted gar genome[40] were extended using HaMStR[77] (v.13.2.6). The spotted gar genes from ref. 40 were used as bait sequences in HaMStR, which sequentially added the best matching protein sequence for each species, provided the bait sequence was in turn the best match in the spotted gar proteome (reciprocity was fulfilled). HaMStR uses hidden Markov model profiles to assign similarity scores. Of the 242 alignments used in the spotted gar study, 190 remained single copy in all the taxa used here. These were used to reconstruct the topology of vertebrates. The 190 protein families were individually aligned using MAFFT[78] (v.7.402) with default settings, concatenated to form an alignment of 310,527 sites and trimmed with automatic method selection in trimAl[79] v.1.2 (-automated1). This concatenated alignment containing 84,017 sites is available in Supplementary File 13. This was used to infer a phylogeny (also provided in Supplementary File 13). We used PhyloBayes[80,81] v.4.1 with the CAT[82] GTR[83] model with 4 discrete gamma categories for site rates[84]. The analysis can be repeated in PhyloBayes with: phylobayes – pb -d alignment -cat -gtr -dgam 4. Convergence was analysed visually in Tracer[85] (v1.7.1) and using bpcomp and tracecomp in the PhyloBayes suite. Six chains were run for between 12,991 and 13,836 cycles. After a burn-in of 1,500 cycles, bpcomp revealed that all bipartitions were present in exactly the same frequencies (maxdiff and meandiff = 0). Tracecomp revealed effective sample sizes of parameters ranging from 522 to 11,491, with relative differences of 0.018 to 0.218. We deemed that, at least for topology construction, these chains had converged sufficiently, therefore, recovered topologies reflected the true posterior distribution.

### Dating species divergences
For molecular clock analysis, we expanded the dataset to include several non-vertebrate outgroups because many of the calibrations have similar maximum bounds, meaning the effective time prior would be older than intended if we did not include the outgroups. HaMStR was then used to extend the orthogroups to include the new taxa. Of the original 190 orthogroups, 172 were retained in single copy in all taxa; they were aligned and trimmed as before. This alignment (provided in Supplementary File 14) was used as input to MCMCtree[84] (v.4.9j) using approximate likelihood estimation[86]. The analysis was run on each gene under the simplest possible model. The temporary control files were then used as input to CODEML (v.4.9j) for each gene with the following modifications. The substitution model was changed to the one that was preferred by ProtTest[87] (v.3.4.2) from a subset of LG[88], WAG[89], JTT[90], Dayhoff[91] and BLOSUM62 (ref. 92). Fix_alpha was set to 0, alpha

was set to 0.5 and the number of gamma categories was set to 5. The Hessian matrices generated were concatenated to form the .BV file, which was used for the approximate likelihood estimation in the full analysis. The time prior was constructed by applying a uniform prior distribution with a hard minimum bound and a soft maximum bound (with 2.5% probability greater than the maximum) to nodes. We used the autocorrelated rates clock model with a gamma prior distribution with shape = 2 and scale = 4.53. This was constructed by dividing a typical distance between two tips whose most recent common ancestor was at the root of the tree under LG + F + G4 (inferred with IQ-TREE[93] v.1.6.3) by the expected time for the tree based on the root prior. This was multiplied by the shape parameter of 2 (leading to a fairly flat gamma distribution, corresponding to a relatively uninformative prior). The variance prior (sigma2) had shape = 1 and scale = 1, meaning that variation in rates is not highly penalized in the posterior distribution. Rates across sites were modelled by a gamma distribution with shape = 1 and scale = 1 with 5 discrete categories. After a burn-in period of 10,000 generations, parameter values were saved every 20th generation until 20,000 cycles were saved (400,000 generations in total). Convergence was investigated in Tracer[85], revealing convergence had been reached in the six chains run (the lowest effective sample size was 194 and posterior distributions in all 6 chains looked almost identical). The alignments, control files and tree are available in Supplementary File 14.

### Estimation of gene duplication rates
Orthogroups were predicted using OrthoFinder[94,95] v.2.3.5; output from this analysis is available as Supplementary File 15. OrthoFinder includes a gene duplication prediction step as part of its pipeline. Gene duplication events presented here had >50% support. The species tree was fixed to the topology inferred in this study.

### Rooting the vertebrate phylogeny
Orthogroups were predicted using OrthoFinder[94] v.2.3.5 for only the vertebrate taxa (hagfish, lampreys and gnathostomes). For each gene family, sequences were aligned using MAFFT[78] (v.7.402) with default settings, then trimmed using trimAl[79] with heuristic choice of trimming parameters. IQ-TREE[93] was then used to generate 1,000 bootstrapped trees in a maximum likelihood framework, with the model selected using ModelFinder[96] (as part of IQ-TREE v.1.6.3). These bootstrapped trees were used as the input to ALEobserve (v.1.0) to create ALE objects. Two species trees were used as hypotheses; one with hagfish as sister to all other vertebrates and one with monophyletic cyclostomes. ALEml_undated[29] (v.1.0) was used with each of these species tree hypotheses with default settings, except for tau (the transfer rate) which was set to 0, meaning that transfers could not be inferred. This estimates the pattern of gene duplication and loss for each gene family under the different species tree hypotheses, as well as a likelihood under the species tree. An approximately unbiased test[30] was then performed on the likelihoods of each gene family under the two competing hypotheses using the programme CONSEL[97] (v.0.2.0).

### Ancestral gene family complements
A total of 45 animal genomes (Supplementary Table 1 and Extended Data Fig. 3) were compared using a pipeline described previously[31–33]. Briefly, the proteomes were compared using a reciprocal blastp of all-vs-all sequences with DIAMOND[98] (v.0.9.30.131; *e*-value threshold of $1 \times 10^{-5}$). Markov cluster algorithm[99] (v.1:14-137+ds-4) was used to infer homology groups (HGs) from the BLAST output with default inflation parameter (I = 2). GOs were assigned to the different HGs by analysing the human protein sequences in each HG with PANTHER GO[100] (v.15.0).

### Orthology relationships of *Hox* gene clusters
Hagfish *Hox* sequences were obtained from a previous study[19] and used as queries in TBLASTN (v.2.10.1+) to find the location in the Hi-C assembly and Ensembl annotation. Information about *Hox* syntenic genes

in lamprey, human and elephant shark, with the European amphioxus as outgroup, were obtained from previous studies[17–19,101], downloaded from Ensembl or NCBI GenBank and used as queries to find their presence in the hagfish Hi-C assembly and Ensembl annotation in TBLASTN. Location of *Hox* and their syntenic genes, as well as their Ensembl Gene IDs is provided in Supplementary Table 29. For phylogenetic analysis of Hnrnpa, Cbx, Gbx and Agap, amino acid sequences were aligned using MUSCLE[102] as implemented in MEGAX[103] v.10.2.4. The alignment was trimmed by trimAl[79] v.1.2rev59 using the '-automated1' option and then formatted into a nexus file using readAl (bundled with the trimAl package). The Bayesian inference tree was constructed using MrBayes[104] v.3.2.6, under the assumption of an LG + I + G evolutionary model, with two independent runs and four chains. The tree was considered to have reached convergence when the standard deviation stabilized under a value of <0.01. A burn-in of 25% of the trees was performed to generate consensus trees. Multisequence alignments with MrBayes parameters and number of generations for each tree are provided in Supplementary Files 16–19.

### Dating genome duplications in vertebrates
OrthoFinder-inferred gene families were selected that showed a clear signal of both the 1R and 2R duplication events and were broadly congruent with current phylogenetic hypotheses. This resulted in 35 gene families in which each gnathostome was represented up to four times and each cyclostome twice. Gene families containing a signal of both 1R, 2R and the cyclostome duplication event (CR) were rare, hence, to date the CR event, an additional dataset was assembled consisting of 27 gene families in which each cyclostome species was represented by at least two gene copies.

For each analysis, taxon sampling towards the root of the tree was improved by including additional outgroup taxa *Nematostella vectensis* (Cnidaria), *Trichoplax adhaerens* (Placozoa), *Mnemiopsis leidyi* (Ctenophora) and *Hofstenia miamia* (Xenacoelomorpha); this served to remove the nodes of interest from the root of the tree and include additional relative and absolute calibration information for more universal clades. Individual gene families were aligned using MUSCLE[102] (v.5) and trimmed using the '-automated1' option in trimAl[79]. The best-fitting model for each gene family was determined using IQ-TREE[93] (v.2.1.3) and all gene families were concatenated into a single alignment.

The node age time priors were based on the posterior estimates from the associated species divergence times analysis (see 'Dating species divergences' above), using the span of the 95% highest density credibility intervals of node ages from that analysis to inform uniform time priors on the same species nodes in gene tree analysis, with a 1% probability tail that the maximum age could be exceeded. Calibrations within lineages that have undergone WGD were repeated across the duplicated clades with identical probability distributions. Molecular clock analyses were performed using the normal approximation method in MCMCtree[105] (v.4.9j), with each gene treated as a separate partition. Four independent Markov chain Monte Carlo (MCMC) chains were run for 2 million generations each, with the first 20% discarded as burn-in. Convergence was determined using Tracer[85] and by comparing congruence among all four runs. The alignments, MCMCtree control files and calibrations used are available in Supplementary File 20; the dates of species divergence are presented in Fig. 2; and dating of WGD events is shown in Fig. 2 and Supplementary Fig. 34.

### Reconstruction of vertebrate ancestral chromosomes
On the basis of reciprocal best BLASTP v.2.6.0+ (*e*-value threshold of $1 \times 10^{-6}$) search and a chi-squared test (multiple test correction with false discovery rate, *q* value threshold of 0.05), we identified homologous chromosomes within either the chicken or spotted gar genome that possessed significantly more between-chromosome homologous genes. Homologous chromosomes between either the

chicken or spotted gar genome and sea cucumber chromosomes were also inferred, except that the best BLASTP search was unidirectional wherein the sea cucumber genes were the reference. From inferred within-species and between species homologies, all chicken and spotted gar chromosomes were grouped into 17 groups representing the 17 predicted ACs that contribute to extant gnathostome karyotypes. The gene content of these ACs was reconstructed with Belcher's lancelet (*B. belcheri*) genes[43]. A Belcher's lancelet gene was distributed to one vertebrate ancestral chromosome if either (1) the scaffold this gene is located on is a homologous scaffold to the specific sea cucumber linkage group, and this gene is homologous to the corresponding chicken and spotted gar genes; or (2) this gene is homologous to at least five different chicken and spotted gar genes. As a result, 5,065 lancelet genes were anchored to the 17 inferred ACs and used as their gene content, ranging from 115 to 534 genes in ACs 1–16, and only 20 genes in AC17 (Supplementary Table 34).

### Phylogenetic support around 1R/2R
A homologous gene set is a group of genes that share the same best BLASTP hit AC gene. Multiple sequence alignments were obtained with PRANK[106] v.150803. ModelFinder[96] (embedded in IQ-TREE[93] v.1.6.12) with BIC criteria and '-mtree' parameter was used to find the best-fitting model. RAxML-ng v.0.9.0 and IQ-TREE were repeatedly run for 10 times with different seed numbers. For the 20 obtained maximum likelihood trees, we used RAxML-ng (v.0.9.0) to re-evaluate their likelihoods and chose the best tree as the final tree for each homologous gene set (Supplementary Files 5–8).

### Definition and calculation of overlapping ratio
For a reference chromosome and all genes on it, the existence or absence of a homologue on a query chromosome is denoted as binary mode 1 or 0. We defined it as the gene retention profile. Mathematically, it is a vector with values of either 1 or 0 and with fixed length that corresponds to the number of genes on the query chromosome. One notable property of the gene retention profile is that the gene order within the query chromosome does not alter the gene retention profile itself. The OR was calculated between two gene retention profiles that correspond to one same reference chromosome. It is equal to the number of shared homologues divided by the smaller one of two total numbers of homologues and has a value range between 0 and 1. Notably, the OR is insensitive to the size difference between two query chromosomes.

### Hierarchical clustering based on the gene retention profile
For multiple chromosomes homologous to one same vertebrate ancestral chromosome, we inferred their gene retention profiles and calculated all pairwise ORs. We used 1 − OR as a measure of pairwise distance and performed hierarchical clustering with the 'Ward.D' method provided in the R platform.

### GO enrichment analysis of ohnologues
We mapped hagfish and chicken ohnologues to human genes and performed GO enrichment analysis with human orthologues. Functional enrichment was examined with the Metascape[107] online tool. We used the 959 human orthologues of hagfish ohnologues and randomly sampled 2,999 genes (as Metascape has a limit of 3,000) from a total of 3,595 chicken orthologues. GO (biological process) enrichment analysis was performed against all genes of the two species. Genes were annotated as either developmental ohnologues, non-developmental ohnologuess or non-ohnologous genes according to their GO terms annotated by PANTHER[108] v.17.0.

### Chromatin accessibility profiling
ATAC-seq experiments on two hagfish embryos at stages Dean 45 (collected in 2018) and 53 (collected in 2017) (Supplementary Fig. 31) were performed following previous descriptions[101,109] with slight

variations (details are provided in Supplementary Information, section 5). Embryos were divided and processed into two halves to gain positional information to be used in a future project. Approximately 50,000 nuclei per replicate (~200,000 nuclei per embryo) were processed for tagmentation using Tn5 from the Illumina Nextera DNA Library Prep kit. Libraries were multiplexed and sequenced at the Beijing Genomic Institute in 4 lanes (2 per embryo) in an Illumina HiSeq 4000 platform.

To identify ACRs as putative gene *cis*-regulatory regions, we collected ATAC-seq data of hagfish and other chordate embryos (amphioxus, zebrafish, medaka, chicken and mouse; GSE106428 (ref. [47]) and DRA006971 (ref. [110])) with two replicates each (Supplementary Table 51). For each data, ATAC-seq paired-end reads were aligned to the reference genome using Bowtie2 (ref. [111]) (v.2.4.2). After extracting nucleosome-free read pairs (the insert shorter than 120 bp), we performed peak-calling by using MACS2 (ref. [112]) (v.2.2.7.1). Finally, on the basis of the replicate information, reproducible peaks were identified as ACRs using the IDR framework[113].

### Fate of ohnologues after WGD

After quantile normalization, transcripts per million >5 was used as a threshold to consider a gene to have either an 'expressed' or 'not expressed' state (Supplementary Fig. 31). Only ohnologue pairs in which both genes are expressed in at least one tissue were analysed. Fates of ohnologues were classified according to their expressional patterns in the tissues assayed in this study for the hagfish, or from a previous study in the case of chicken[114]. Fates were defined following a different strategy from that in ref. [47] due to the lack of information from homologous tissues of the amphioxus (outgroup). After WGD, ohnologues can follow one of the following fates: (1) potential redundancy, if the two ohnologues are expressed in the same set of tissues; (2) potential subfunctionalization, if both ohnologues are each expressed in a tissue not shared with the other. In other words, each of them has tissue-specific expression domains; (3) potential specialization, if one ohnologue has a reduced set of expression domains contained in a larger set of tissues in which the other ohnologue is expressed. Gene families within 'specialization' can be further defined as having either 'potential strong specialization' when the ohnologue with the narrower expression pattern is transcribed in <40% of the domains than the ohnologue with the broader expression pattern; or 'potential mild specialization' when the ohnologue with the narrower expression pattern is transcribed in ≥40% of domains than the ohnologue with the broader expression pattern.

### Phenotypic disparity analyses

A character matrix of 578 characters and 278 taxa was assembled as follows. Characters were collected from direct observations and multiple literature sources[115–126]. Previous literature sources were modified to ensure that duplicated character states were removed and that overlapping characters from different sources were combined into single characters or subdivided into multiple characters to encompass all variation across vertebrates. We ensured that all characters were coded for as many taxa as possible. Missing data are coded as '?'; inapplicable characters are coded as '-'. Character state observations were coded using primary observations and through the literature. Characters were coded using hierarchical contingencies[127–129]. The character matrix and descriptions are available in Supplementary File 21 (Vertebrate_disparity_matrix.nex).

The phenotype character matrix was transformed before disparity analyses such that characters coded 'not applicable' were scored as '0' and each subsequent character state was increased by 1. Ancestral character states were estimated along a tree representative of current phylogenetic hypotheses using stochastic character mapping[130], with 1,000 simulations per character; the tree is available in Supplementary File 21 (Disparity.tre). Distances between taxa and reconstructed internal nodes were estimated using Gower's dissimilarity metric[131], and these distances were ordinated using NMDS, a method that seeks to reduce dimensionality while preserving distances between taxa. A pre-ordination phylomorphospace was plotted using the inferred ancestral states, NMDS scores and the representative phylogeny. Convex hulls were fitted around taxonomic lineages and groups that have undergone successive rounds of WGD. All stem gnathostomes were adjudged to have undergone the 2R WGD because they postdate the timing of 2R inferred from the gene tree-based molecular clock analysis (see 'Dating genome duplications in vertebrates' above). Disparity metrics were estimated using dispRity (v.1.7.0) in R (v.2.6-4) with 1,000 bootstrap replicates[132].

### Reporting summary

Further information on research design is available in the Nature Portfolio Reporting Summary linked to this article.

## Data availability

The *Eptatretus burgeri* (inshore hagfish) v.4.0 genome is available in NCBI GenBank under accession number GCA_900186335.3. Raw genome sequencing data together with adult RNA-seq data have been deposited in the European Nucleotide Archive (ENA) at EMBL-EBI under accession number PRJEB21290. ATAC-seq data have been deposited in Gene Expression Omnibus (GEO) under accession number GSE247552. Supplementary files are available at FigShare (https://figshare.com/projects/Hagfish_Genome_Project/163186). Gene annotation used in this study is available at https://www.ensembl.org/Eptatretus_burgeri. A mirror of the UCSC Genome Browser containing hagfish assembly and annotations is available at http://ucsc.crg.eu/.

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

## Acknowledgements

This research was supported by grants from the Ministry of Science and Innovation of Spain (PID2021-125078NA-I00) and a KAKENHI Grant-in-aid for Scientific Research C from the Japan Society for the Promotion of Science (19K06798) to J.P.-A.; the National Key R&D Program of China (2019YFA0802600), the Chinese Academy of Sciences (ZDBS-LY-SM005), the National Natural Science Foundation of China (31970565) and the Open Research Program of the Chinese Institute for Brain Research (Beijing) to Y.E.Z.; the Strategic Priority Research Program of the Chinese Academy of Sciences (XDB13000000) to Y.E.Z and W. Wang; the Wellcome Trust (to F.J.M., Grant number 108749/Z/15/Z); the Leverhulme Trust (RF-2022-167 to P.C.J.D); the Biotechnology and Biological Sciences Research Council (BB/T012773/1 to P.C.J.D); the Natural Environment Research Council (NERC) grant (NE/P013678/1 to P.C.J.D. and D.P.), part of the Biosphere Evolution, Transitions and Resilience (BETR) programme, which is co-funded by the Natural Science Foundation of China (NSFC); the John Templeton Foundation (Grant 62220 to P.C.J.D. and D.P.; the opinions expressed in this publication are those of the author(s) and do not necessarily reflect the views of the John Templeton Foundation); Wellcome Trust Seed Award (210101/Z/18/Z to J.P.); the Tromsø Research Foundation (Tromsø Forskningsstiftelse, TFS) to B.F. (20_SG_BF 'MIRevolution'); a National Science Foundation (NSF) grant (1755418) to M.H.; and National Institutes of Health (NIH) grants (R01AI072435 and R35GM122591) to M.D.C. Computing was jointly supported by the HPC Platform of BIG and that of the Scientific Information Centre of IOZ. We thank O. Kakitani and the members of the Fishery Association in Gotsu City, Shimane Prefecture, Japan, for assistance in collecting hagfish; the technical support staff of the Laboratory for Phyloinformatics in RIKEN Kobe for sequencing data production, and RIKEN aquarium staff; T. de Dios Fernández for animal illustrations. For the purpose of open access, the authors have applied for a CC BY public copyright licence to any author-accepted manuscript version arising from this submission.

## Author contributions

D.Y., Y.R., J.S., Y.L., W. Wan, W. Wang, Y.E.Z. and J.P.-A sequenced, assembled and evaluated the hagfish genome. F.J.M. generated and coordinated the annotation process at Ensembl. M.M., C.C., M.P. and W.A. generated Ensembl Compara data. A.d.M., D.Y., A.J.S.B. and J.W.C. generated orthogroups. R.F., J.F.F., F.S., S.D'A., C.B., C.A., G.S., B.F., K.J.P., S.D., M.H., J.P.R. and M.D.C. contributed to gene family curation. A.J.S.B., J.W.C., D.P. and P.C.J.D. performed phylogenomics and dating analyses. J.P. performed analysis of gene family evolution. D.Y., J.S. and Y.E.Z performed macrosynteny analysis. I.M. and M.I. advised on macrosynteny and ohnologue evolution analyses and discussed the data. J.P.-A., I.S., F.S. and S.K. obtained hagfish embryonic and adult material. J.P.-A performed ATAC-seq and RNA-seq experiments and ensured sequencing project management with BGI and RIKEN. M.U. and J.S. carried out regulatory genome profiling and ohnologue fate evolution analyses. J.N.K., E.M.C., R.P.D., S.G., E.R., R.S.S. and P.C.J.D. obtained fossil and calibration data, and performed morphological disparity analysis. W. Wang, S.K., N.I. and J.P.-A. conceived the project. S.K., N.I., P.C.J.D., W. Wang, Y.E.Z and J.P.-A contributed to project design. J.P.-A. coordinated the project. Y.E.Z., P.C.J.D and J.P.-A. wrote the paper, with inputs from all authors. All authors revised and approved the paper.

## Competing interests

M.D.C. is a cofounder and shareholder of NovAb, Inc., which produces lamprey antibodies for biomedical purposes. J.P.R. is a consultant for NovAb. The other authors declare no competing interests.

## Additional information

**Extended data** is available for this paper at https://doi.org/10.1038/s41559-023-02299-z.

**Correspondence and requests for materials** should be addressed to Fergal J. Martin, Wen Wang, Philip C. J. Donoghue, Yong E. Zhang or Juan Pascual-Anaya.

Daqi Yu[1,2,41], Yandong Ren[3,4,41], Masahiro Uesaka[5,6,41], Alan J. S. Beavan[7,36,41], Matthieu Muffato[8,37,41], Jieyu Shen[1,2], Yongxin Li[3,4], Iori Sato[5,38], Wenting Wan[3,4], James W. Clark[7,39], Joseph N. Keating[9], Emily M. Carlisle[9], Richard P. Dearden[10,40], Sam Giles[10], Emma Randle[11], Robert S. Sansom[11], Roberto Feuda[12], James F. Fleming[13,14], Fumiaki Sugahara[15,16], Carla Cummins[8], Mateus Patricio[8], Wasiu Akanni[8], Salvatore D'Aniello[17], Cristiano Bertolucci[17,18], Naoki Irie[19,20], Cantas Alev[21], Guojun Sheng[22], Alex de Mendoza[23], Ignacio Maeso[24,25], Manuel Irimia[26,27,28], Bastian Fromm[29], Kevin J. Peterson[30], Sabyasachi Das[31,32], Masayuki Hirano[31,32], Jonathan P. Rast[31,32], Max D. Cooper[31,32], Jordi Paps[9], Davide Pisani[7,9], Shigeru Kuratani[5,16], Fergal J. Martin[8 ✉], Wen Wang[3,4,33 ✉], Philip C. J. Donoghue[9 ✉], Yong E. Zhang[1,2,33 ✉] & Juan Pascual-Anaya[16,34,35 ✉]

[1]Key Laboratory of Zoological Systematics and Evolution and State Key Laboratory of Integrated Management of Pest Insects and Rodents, Institute of Zoology, Chinese Academy of Sciences, Beijing, China. [2]University of Chinese Academy of Sciences, Beijing, China. [3]State Key Laboratory of Genetic Resources and Evolution, Kunming Institute of Zoology, Chinese Academy of Sciences, Kunming, China. [4]School of Ecology and Environment, Northwestern Polytechnical University, Xi'an, China. [5]Laboratory for Evolutionary Morphology, RIKEN Center for Biosystems Dynamics Research (BDR), Kobe, Japan. [6]Department of Ecological Developmental Adaptability Life Sciences, Graduate School of Life Sciences, Tohoku University, Sendai, Japan. [7]Bristol Palaeobiology Group, School of Biological Sciences, University of Bristol, Bristol, UK. [8]European Molecular Biology Laboratory, European Bioinformatics Institute, Wellcome Genome Campus, Hinxton, UK. [9]Bristol Palaeobiology Group, School of Earth Sciences, University of Bristol, Bristol, UK. [10]School of Geography, Earth and Environmental Sciences, University of Birmingham, Edgbaston, Birmingham, UK. [11]Department of Earth and Environmental Sciences, University of Manchester, Manchester, UK. [12]Department of Genetics and Genome Biology, University of Leicester, Leicester, UK. [13]Keio University Institute for Advanced Biosciences, Tsuruoka, Japan. [14]Natural History Museum, University of Oslo, Oslo, Norway. [15]Division of Biology, Hyogo Medical University, Nishinomiya, Japan. [16]Evolutionary Morphology Laboratory, RIKEN Cluster for Pioneering Research (CPR), Kobe, Japan. [17]Biology and Evolution of Marine Organisms, Stazione Zoologica Anton Dohrn Napoli, Villa Comunale, Napoli, Italy. [18]Department of Life Sciences and Biotechnology, University of Ferrara, Ferrara, Italy. [19]Research Center for Integrative Evolutionary Science, The Graduate University for Advanced Studies, SOKENDAI, Hayama, Japan. [20]Department of Biological Sciences, Graduate School of Science, The University of Tokyo, Tokyo, Japan. [21]Institute for the Advanced Study of Human Biology (ASHBi), Kyoto University, Kyoto, Japan. [22]International Research Center for Medical Sciences (IRCMS), Kumamoto University, Kumamoto, Japan. [23]School of Biological and Behavioural Sciences, Queen Mary University of London, London, UK. [24]Department of Genetics, Microbiology and Statistics, Faculty of Biology, University of Barcelona (UB), Barcelona, Spain. [25]Institut de Recerca de la Biodiversitat (IRBio), Universitat de Barcelona (UB), Barcelona, Spain. [26]Centre for Genomic Regulation (CRG), Barcelona Institute of Science and Technology (BIST), Barcelona, Spain. [27]Universitat Pompeu Fabra (UPF), Barcelona, Spain. [28]ICREA, Barcelona, Spain. [29]The Arctic University Museum of Norway, UiT - The Arctic University of Norway, Tromsø, Norway. [30]Department of Biological Sciences, Dartmouth College, Hanover, NH, USA. [31]Department of Pathology and Laboratory Medicine, Emory University, Atlanta, GA, USA. [32]Emory Vaccine Center, Emory University, Atlanta, GA, USA. [33]CAS Center for Excellence in Animal Evolution and Genetics, Chinese Academy of Sciences, Kunming, China. [34]Department of Animal Biology, Faculty of Science, University of Málaga (UMA), Málaga, Spain. [35]Edificio de Bioinnovación, Universidad de Málaga, Málaga, Spain. [36]Present address: School of Life Sciences, University of Nottingham, Nottingham, UK. [37]Present address: Tree of Life, Wellcome Sanger Institute, Hinxton, UK. [38]Present address: iPS Cell Advanced Characterization and Development Team, RIKEN BioResource Research Center, Tsukuba, Japan. [39]Present address: Milner Centre for Evolution, University of Bath, Claverton Down, Bath, UK. [40]Present address: Naturalis Biodiversity Center, Leiden, the Netherlands. [41]These authors contributed equally: Daqi Yu, Yandong Ren, Masahiro Uesaka, Alan J. S. Beavan, Matthieu Muffato. ✉e-mail: fergal@ebi.ac.uk; wwang@mail.kiz.ac.cn; Phil.Donoghue@bristol.ac.uk; zhangyong@ioz.ac.cn; jpascualanaya@gmail.com

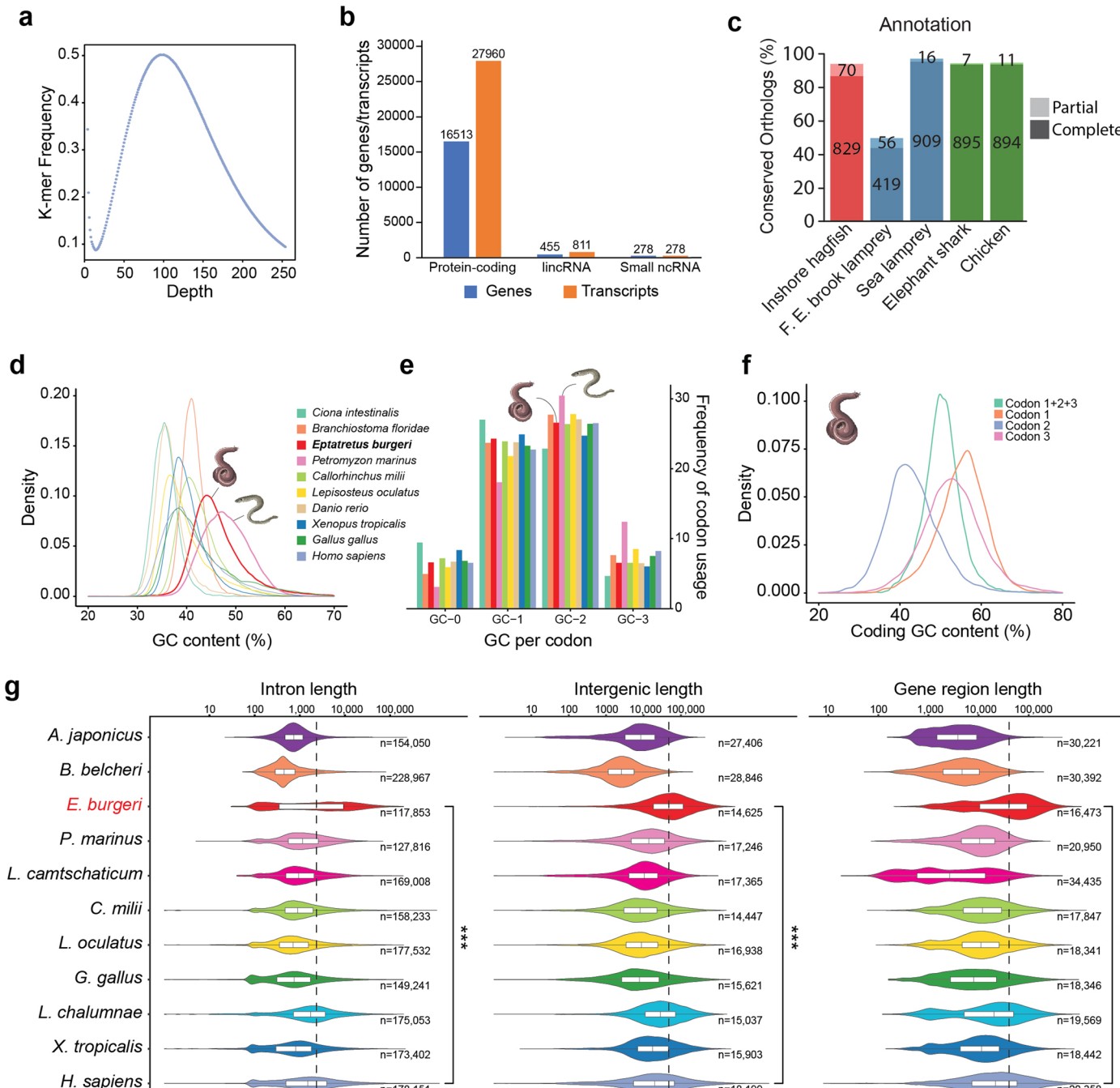

**Extended Data Fig. 1 | Features of the hagfish genome. a**, 17-mer distribution for inshore hagfish genome size estimation using all raw reads from short insert-size libraries. **b**, Counts for major classes of genes and transcripts from Ensembl annotation. **c**, Completeness assessment of the annotation of the inshore hagfish *E. burgeri* genome (red), three lamprey species (blue) and two jawed vertebrates (green). Numbers of conserved metazoan orthologs (metazoa_odb10 dataset, n = 954) are indicated for each case. F. E., Far Eastern **d**, GC-content distribution of the hagfish genome and other chordates calculated from 10-kb non-overlapping windows. **e**, All codon type frequency in given chordate genomes according to GC-content. GC-0/1/2/3 indicates the number of G or C bases in a codon. **f**, Distribution of GC content at each codon position or at all codon positions (Codon 1 + 2 + 3). For each protein coding gene, we only kept the longest coding sequence. For each coding sequence, we calculated the GC content at separate codon positions. We also calculated the GC content for each coding sequence, which is equal to the GC content of all three codon positions. **g**, Violin plots of size distribution of intron (left), intergenic (middle) and gene body (right) lengths over a logarithmic scale of the hagfish (*E. burgeri*), two lamprey species (sea lamprey, *P. marinus*; Arctic lamprey, *L. camtschaticum*), six gnathostome vertebrates (human, *H. sapiens*; frog, *X. tropicalis*; coelacanth, *L. chalumnae*; chicken, *G. gallus*; spotted gar, *L. oculatus*; and the elephant shark, *C. milii*) and two invertebrate deuterostomes (sea cucumber, *A. japonicus*; amphioxus, *B. belcheri*). For each genomic feature and each species, the median (centre line) and IQR (interquartile range) length statistics are indicated with a white rectangle; whiskers extend to the last point no further than 1.5 times the interquartile range from the first and third quartiles Dashed vertical line indicates median size of *E. burgeri* features; *** $P < 2.2 \times 10^{-16}$, two-sided Wilcoxon rank sum test. Animal illustrations kindly provided by Tamara de Dios Fernández and reproduced with permission from REF.[133].

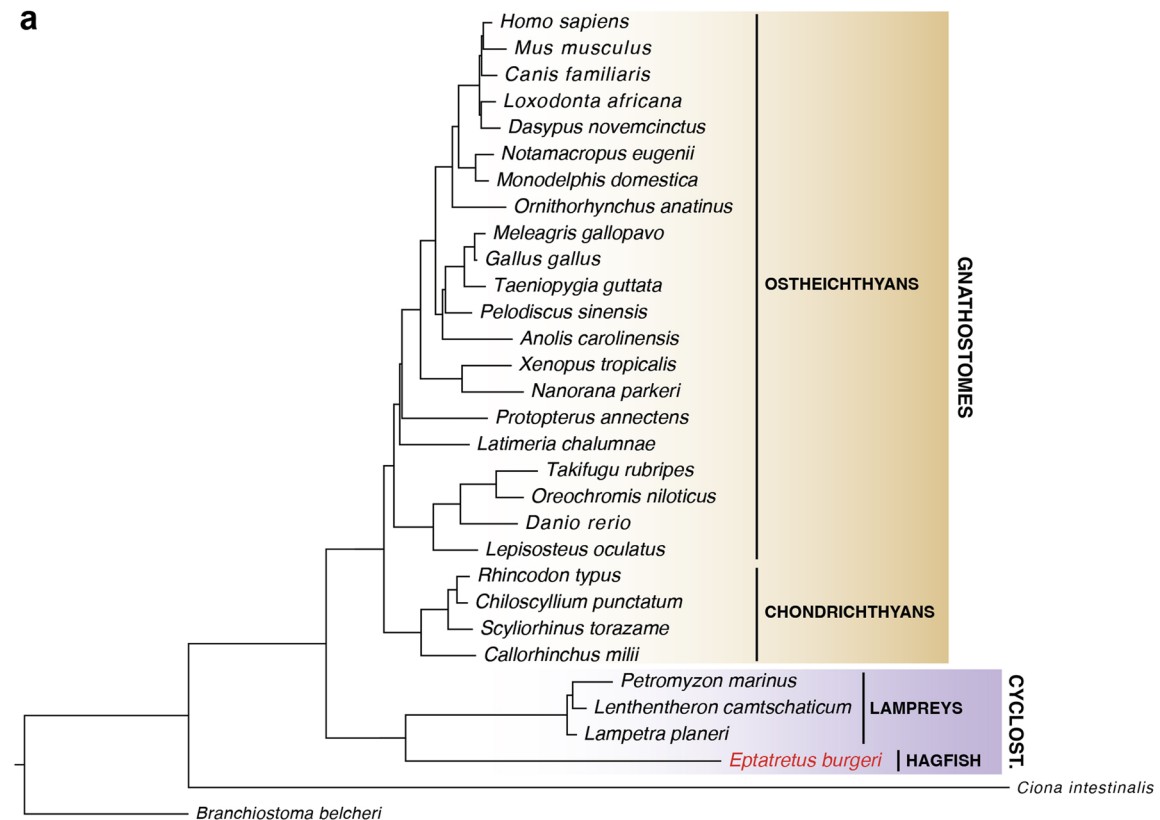

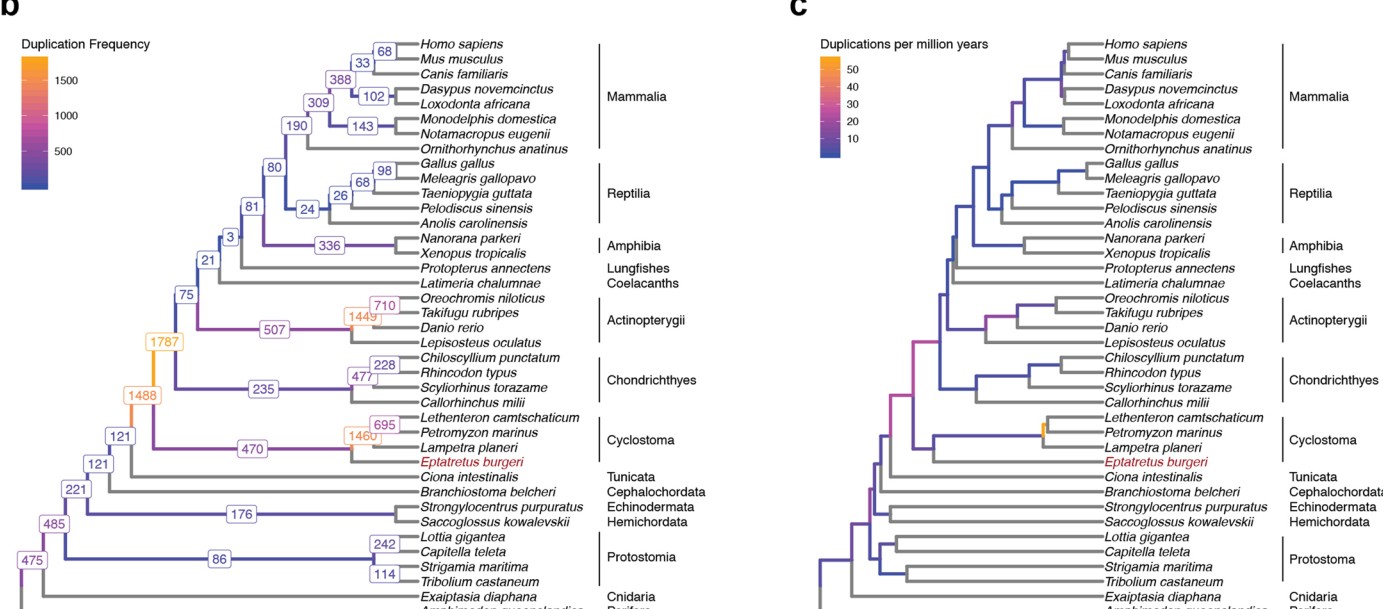

**Extended Data Fig. 2 | Phylogenomic analysis of chordates and gene duplication rates across metazoan evolution. a**, Bayesian Inference tree of 31 chordate species was built using a protein alignment (see Methods). All nodes were recovered with a posterior probability of 1. Cyclostome monophyly was unequivocally supported. Scale bar indicates 0.1 substitutions per site.

**b, c**, The exact number of gene duplication events inferred using OrthoFinder2 with greater than 50% support (**b**) and the number of events per million years per branch (**c**) are shown. In each case, the colour of the branch represents the value according to the key on the upper-left of each pane. Hagfish is highlighted in red font in all panels.

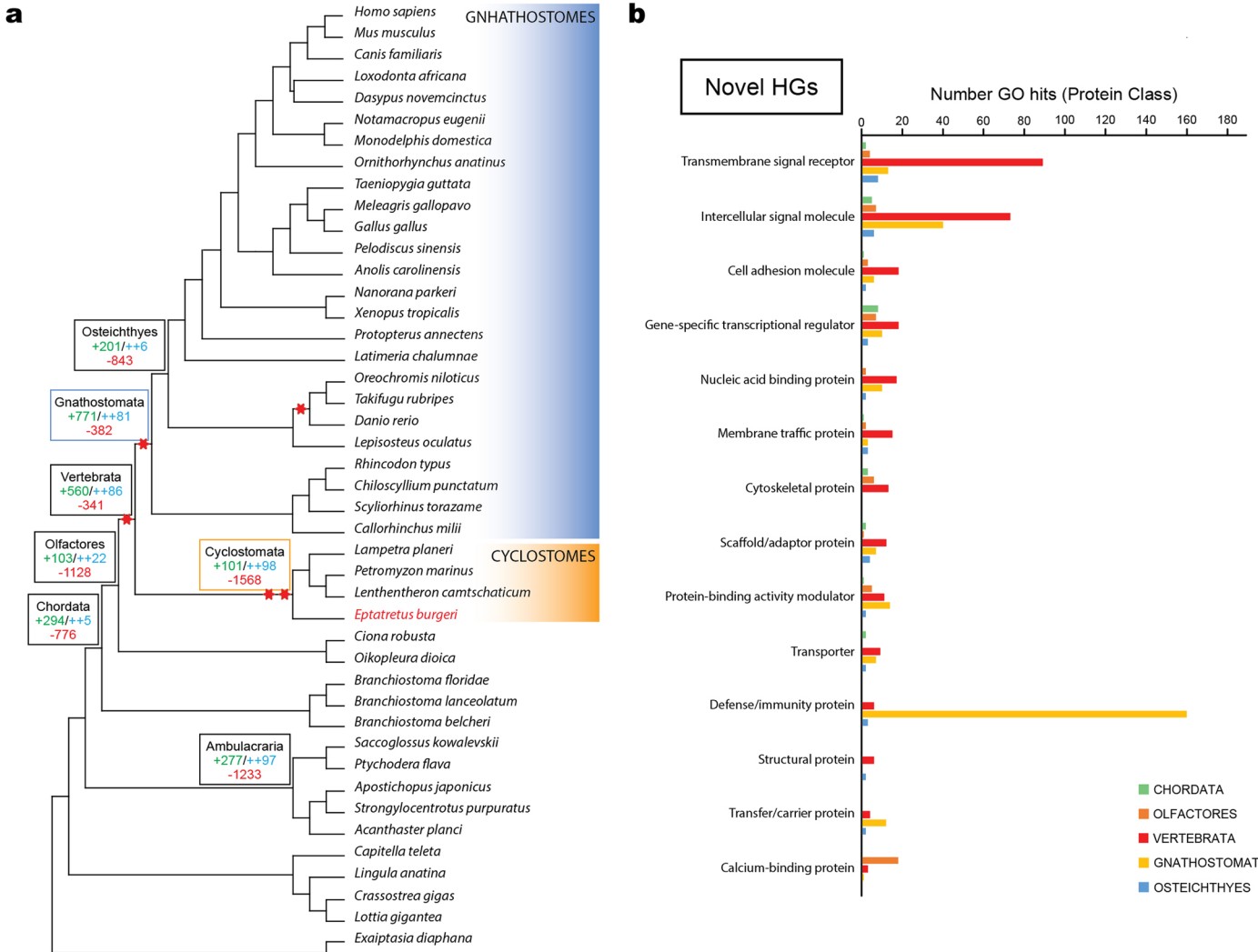

**Extended Data Fig. 3 | Reconstruction of ancestral gene content.**
**a**, Cladogram showing the phylogenetic relationships of 45 species representatives of all major eumetazoan taxa with species of gnathostomes and cyclostomes highlighted in blue and orange, respectively. Gene family gains and losses are indicated in selected nodes: green, novel homology groups (HG); blue, novel core HGs; red, lost HGs. **b**, Top 14 Protein Class GO hits for novel homology groups (HG) gained across different nodes of chordates, color coded by taxa (legend at the bottom right) and sorted by the Vertebrata node. The largest GO enriched terms are 'transmembrante signal receptor' and 'intercellular signal molecule' in vertebrates, and 'defense/immunity protein' in gnathostomes.

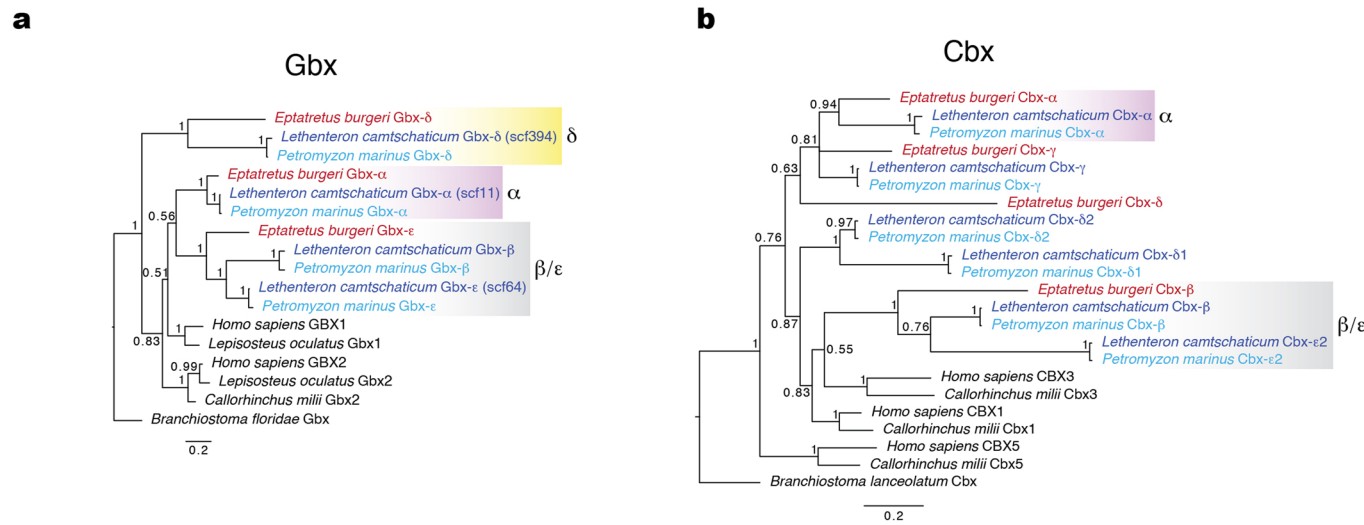

Extended Data Fig. 4 | See next page for caption.

**Extended Data Fig. 4 | Phylogenetic and retention profile clustering analyses of Hox syntenic regions. a-d**, Bayesian inference phylogenetic trees of amino acid sequences of 4 non-*Hox* syntenic genes to *Hox* clusters, Gbx (**a**), Cbx (**b**), Hnrnpa (**c**) and Agap (**d**), of the inshore hagfish (in red), the sea lamprey (in light blue), the Arctic lamprey (in dark blue) and selected gnathostomes. Orthologs from the European amphioxus *Branchiostoma lanceolatum* were used as outgroup to root the trees. Posterior probability is indicated in each node. Scales indicate number of substitutions per site. Phylogenetic analyses of *Hox* genes generally fail to determine orthology due their high conservation and short alignments. The phylogenetic trees of these non-*Hox* linked genes clearly support the orthology of *Hox-α* (Gbx, Cbx and Hnrnpa), *Hox-δ* (Gbx, Hnrnpa and Agap), and *Hox-ζ* (Hnrnpa) clusters, while *β* and *ε* genes always group together, as previously observed for the lamprey[17]. The alignments used to build the trees, together with the MrBayes parameters and number of generations used to build each tree are provided as Supplementary Files 16–19. **e**, clustering analysis of retention profiles (see main text) resolved the orthology relationships of *Hox-β*, *Hox-ε*, *Hox-γ*, as well as *Hox-ζ* clusters. Supported orthologies in each analysis are marked with color-coded rectangles. The location of each cluster is indicated in parenthesis in e (ssc, super scaffold; HiC cl, Hi-C contact cluster, or chromosome). For the clustering analysis of AC1-derived chromosomes in the lamprey and hagfish, we split Hi-C cluster 3 into two halves, each containing one Hox cluster: 3L (coordinates 0–107.78 Mb), 3R (107.78-194 Mb).

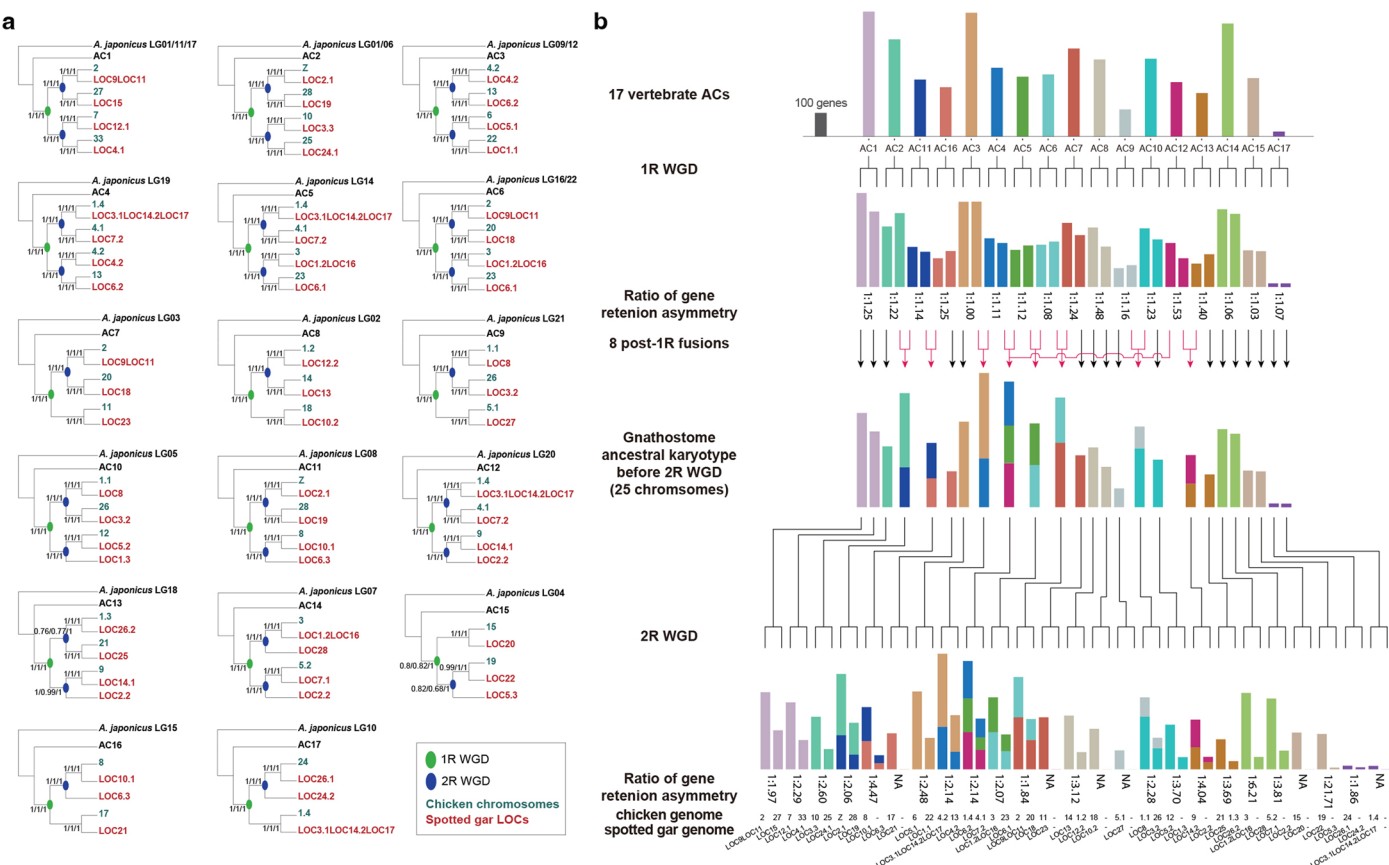

**Extended Data Fig. 5 | Emergence of gnathostome karyotypes via 2R.**
**a**, Chromosomal level phylogenetic trees demonstrate the occurrence of 2R
WGD. The two rounds of WGD are color-coded. Cyan and red denote genes
encoded by corresponding chicken chromosomes and gar LOCs, respectively.
Bootstrap support and posterior probability values of three methods are marked
on branches: bootstrap value from RAxML-ng/bootstrap value from IQ-Tree/

posterior probability from Astral-III. **b**, Evolution of gnathostome karyotype
through 1R and 2R, with post-1R/pre-2R chromosomal fusions indicated with red
lines, and ratios of gene retention asymmetries shown for directly ohnologous
chromosomes after 1R and 2R. The size of each ancestral chromosome is
proportional to the number of retained genes from ACs.

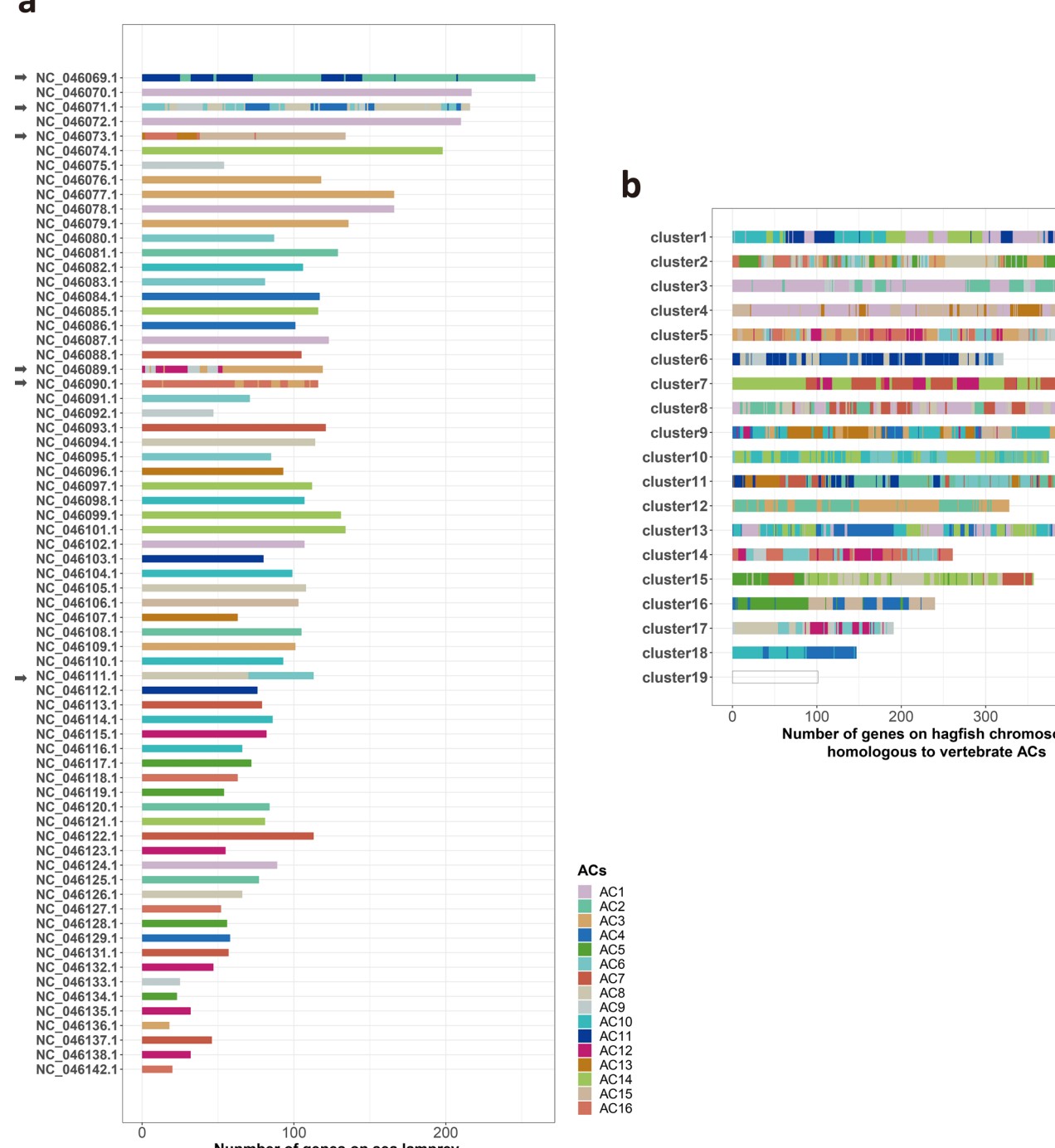

**Extended Data Fig. 6 | Contributions of vertebrate ACs to the genomes of hagfish and sea lamprey. a**, Sea lamprey super-scaffolds[17] (pseudo-chromosomes) are generally homologous to one single AC except five scaffolds labeled with an arrow. Sea lamprey scaffolds scaf_00001, scaf_00002, scaf_00008, scaf_00010, scaf_00011 and scaf_00023, confounded by missassembly, are not presented here. **b**, The distribution of the descendant copies of AC genes for the 19 hagfish Hi-C clusters (putative chromosomes). Only significant homologous relationships between hagfish clusters and ACs are shown. Because hagfish cluster 19 is not homologous to any AC, it is presented as a blank block. Genes are color-coded according to its homologous AC.

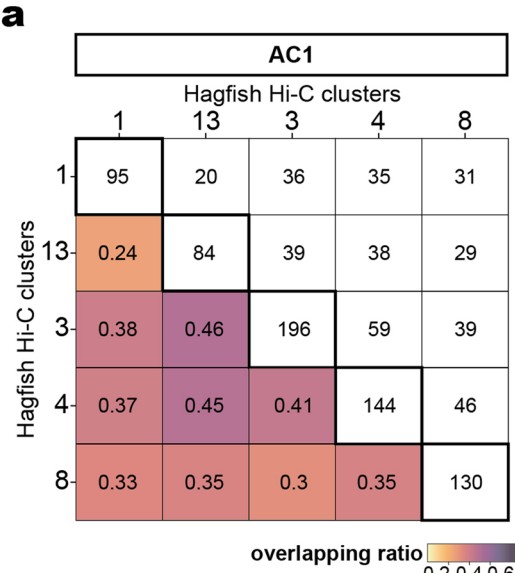

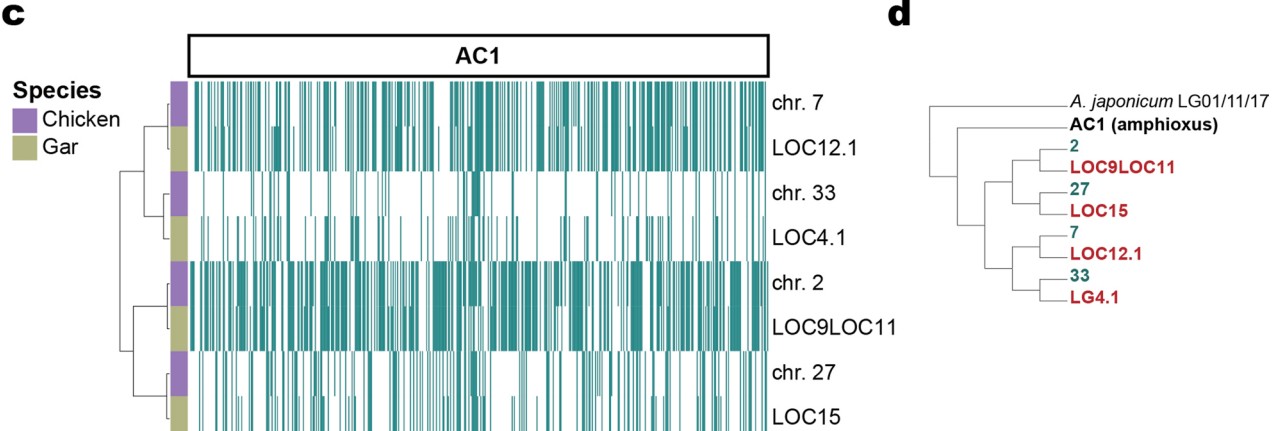

**Extended Data Fig. 7 | Overlapping ratio and clustering analysis of ohnologous and orthologous chromosomes. a**, **b**, AC1 corresponds to five and six mutually paralogous chromosomes in hagfish (**a**) and sea lamprey (**b**) genomes, respectively. Numbers in colour-coded cells (bottom left triangle) indicate the OR between two chromosomes. Numbers in white cells (top right triangle) indicate the number of shared retained genes between two chromosomes. Numbers on the diagonal line from top left to bottom right (thick-lined cells) indicate the total number of retained genes of a chromosome.

Overlapping rations corresponding to all hagfish chromosomes and lamprey scaffolds are provided in Supplementary File 9. **c**, Retention profile clustering analysis of gnathostome orthologous chromosomes deriving from AC1. Retained genes are denoted by dark cyan lines. Four orthologous chromosome pairs are defined. **d**, The clustering found in (**c**) is the same as that found in the phylogenetic analysis (Extended Data Fig. 5a), demonstrating the reliability of the OR approach.

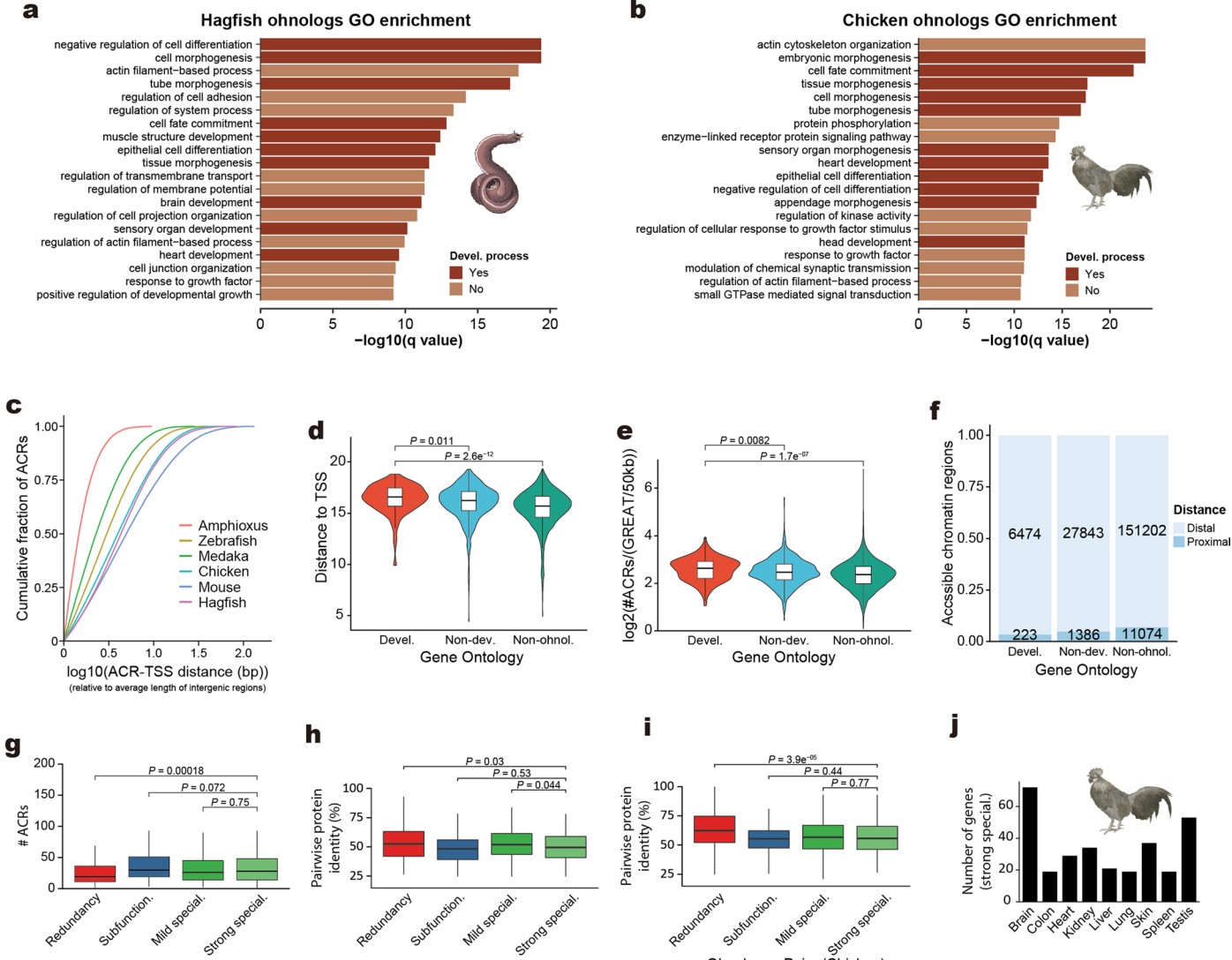

**Extended Data Fig. 8 | Fate and *cis*-regulatory evolution of ohnologs after WGD. a**, **b**, Gene Ontology enrichment analysis of ohnologs in the hagfish (**a**) and the chicken (**b**). Top 20 terms are shown, majority of which are related with development. **c**, Cumulative distribution of distance of ACRs from the closest TSSs normalized by the average length of intergenic regions in each genome. **d**, Distribution of the distance from ACRs to closest TSS of developmental ohnologs (Devel.), non-developmental ohnologs (Non-dev.) and non-ohnologous (Non-ohnol.) genes. **e**, Distribution of the number of ACRs per gene, normalized by the GREAT region length, for developmental ohnologs (Devel.), non-developmental ohnologs (Non-dev.) and non-ohnologous (Non-ohnol.) genes. Sample size for groups in (d) and (e) are identical. n = 143 (Devel.), n = 816 (Non-dev.), n = 7303 (Non-ohnol.). **f**, Proportion of distal ACRs across different gene functional categories. Within a GREAT-defined region, proximal regulatory sequences were defined as those from 5 kb upstream to 1 kb downstream of a TSS, and the rest of the region was treated as distal. **g**, Distribution of the number of ACRs of hagfish ohnologs for each category (special., specialization). n = 344

(Redundancy), n = 178 (Subfunction.), n = 334 (Mild special.), n = 240 (Strong special.). **h**, Distribution of pairwise protein identity for hagfish ohnologous pairs for each category. n = 170 (Redundancy), n = 88 (Subfunction.), n = 165 (Mild special.), n = 120 (Strong special.). **i**, Distribution of pairwise protein identity for chicken ohnologous pairs for each category. n = 275 (Redundancy), n = 178 (Subfunction.), n = 318 (Mild special.), n = 167 (Strong special.). **j**, Number of ohnologues with strong specialization in chicken expressed in each tissue. Only the gene in a pair with narrower expression breadth is analyzed. In panels **d**, **e**, **g-i** *P* values correspond to two-sided Wilcoxon rank sum test between the indicated groups; boxes correspond to the median (centre line) and the first and third quartiles; whiskers extend to the last point no further than 1.5 times the interquartile range from the first and third quartiles. All statistical information for panels **d**, **e**, **g-i** is provided in Supplementary Tables 58–62. Animal illustrations kindly provided by Tamara de Dios Fernández and reproduced with permission from REF. [133].

# Reporting Summary

## Statistics

For all statistical analyses, confirm that the following items are present in the figure legend, table legend, main text, or Methods section.

| n/a | Confirmed | |
|---|---|---|
| ☐ | ☒ | The exact sample size (*n*) for each experimental group/condition, given as a discrete number and unit of measurement |
| ☒ | ☐ | A statement on whether measurements were taken from distinct samples or whether the same sample was measured repeatedly |
| ☐ | ☒ | The statistical test(s) used AND whether they are one- or two-sided *Only common tests should be described solely by name; describe more complex techniques in the Methods section.* |
| ☒ | ☐ | A description of all covariates tested |
| ☒ | ☐ | A description of any assumptions or corrections, such as tests of normality and adjustment for multiple comparisons |
| ☐ | ☒ | A full description of the statistical parameters including central tendency (e.g. means) or other basic estimates (e.g. regression coefficient) AND variation (e.g. standard deviation) or associated estimates of uncertainty (e.g. confidence intervals) |
| ☐ | ☒ | For null hypothesis testing, the test statistic (e.g. *F*, *t*, *r*) with confidence intervals, effect sizes, degrees of freedom and *P* value noted *Give P values as exact values whenever suitable.* |
| ☐ | ☒ | For Bayesian analysis, information on the choice of priors and Markov chain Monte Carlo settings |
| ☒ | ☐ | For hierarchical and complex designs, identification of the appropriate level for tests and full reporting of outcomes |
| ☐ | ☒ | Estimates of effect sizes (e.g. Cohen's *d*, Pearson's *r*), indicating how they were calculated |

*Our web collection on statistics for biologists contains articles on many of the points above.*

## Software and code

Policy information about availability of computer code

| Data collection | No software was used for data collection |
|---|---|
| Data analysis | The following software was used for data anlysis:<br><br>- SOAPec (v2.03)<br>- SOAPdenovo2 (v2.04-r24154)<br>- ABySS (v1.9.055)<br>- GapCloser (v1.12-r6)<br>- Pilon (v1.22)<br>- LACHESIS (compiled Apr. 19, 2019)<br>- R (v2.6-4 and v3.6.0)<br>- ggplot2 (v3.3.5)<br>- dispRity (v1.7.0)<br>- pheatmap (v1.0.12)<br>- LastZ (v1.04)<br>- BWA (v0.7.2-r351)<br>- bedtools (v2.25.0 and v2.29.2)<br>- MirMachine (v0.1.2)<br>- MirMiner (v1.0)<br>- BUSCO (v5.2.2)<br>- HaMSTR (1.3.2.6) |

- MAFFT (v7.402)
- trimAl (v1.2rev59)
- PhyloBayes (v4.1)
- MCMCtree (v4.9j)
- CODEML (v4.9j)
- ProtTest (v3.4.2)
- IQ-TREE (v1.6.3, v1.6.12 and v2.1.3)
- OrthoFinder (v2.3.5)
- ModelFinder (part of IQ-TREE package v1.6.3)
- Tracer (1.7.1)
- ALE (v1.0)
- CONSEL (v0.2.0)
- DIAMOND (v0.9.30.131)
- MCL (v1:14-137+ds-4)
- PANTHER GO (v15.0 and v17.0)
- BLAST package (v2.6.0+ and v2.10.1+)
- MEGA7 (v7.0.18)
- MEGAX (v10.2.4)
- Gblocks (v0.91b)
- Astral-III (v5.6.3)
- MUSCLE (v5; and that bundled with MEGAX v10.2.4)
- MrBayes (v3.2.6)
- PRANK (v150803)
- RAxML-ng (v0.9.0)
- ETE Toolkit (v3.1.3)
- bowtie (v2.4.2)
- NGmerge (v0.3)
- Picard (v2.23.8)
- SAMtools (v1.10)
- MACS (v2.2.7.1)
- RSEM (v1.3.1)
- STAR (v2.6.1d)

For manuscripts utilizing custom algorithms or software that are central to the research but not yet described in published literature, software must be made available to editors and reviewers. We strongly encourage code deposition in a community repository (e.g. GitHub). See the Nature Portfolio guidelines for submitting code & software for further information.

## Data

Policy information about availability of data

All manuscripts must include a data availability statement. This statement should provide the following information, where applicable:
- Accession codes, unique identifiers, or web links for publicly available datasets
- A description of any restrictions on data availability
- For clinical datasets or third party data, please ensure that the statement adheres to our policy

The Eptatretus burgeri (inshore hagfish) v4.0 genome is available in NCBI GenBank under accession number GCA_900186335.3. Raw genome sequencing data together with adult RNA-seq data have been deposited in the European Nucleotide Archive (ENA) at EMBL-EBI under accession number PRJEB21290. ATAC-seq sequencing data have been deposited in Gene Expression Omnibus (GEO) under the following accession numbers: GSE247552. Supplementary files are available at FigShare (https://figshare.com/projects/Hagfish_Genome_Project/163186). Gene annotation used in this study is available at https://www.ensembl.org/Eptatretus_burgeri. A mirror of the UCSC Genome Browser containing hagfish assembly and annotations is available at http://ucsc.crg.eu/.

## Research involving human participants, their data, or biological material

Policy information about studies with human participants or human data. See also policy information about sex, gender (identity/presentation), and sexual orientation and race, ethnicity and racism.

| | |
|---|---|
| Reporting on sex and gender | N/A |
| Reporting on race, ethnicity, or other socially relevant groupings | N/A |
| Population characteristics | N/A |
| Recruitment | N/A |
| Ethics oversight | N/A |

Note that full information on the approval of the study protocol must also be provided in the manuscript.

# Field-specific reporting

Please select the one below that is the best fit for your research. If you are not sure, read the appropriate sections before making your selection.

☒ Life sciences ☐ Behavioural & social sciences ☐ Ecological, evolutionary & environmental sciences

For a reference copy of the document with all sections, see nature.com/documents/nr-reporting-summary-flat.pdf

# Life sciences study design

All studies must disclose on these points even when the disclosure is negative.

| | |
|---|---|
| Sample size | No statistical method was used to predetermine sample size. Our statistical tests involved comparison among gene groups. We have included all analyzable genes in each group. In each group, the sample size was always over 80 genes, providing enough statistical power for conventional Student's t test and two-sided Wilcoxon rank sum test. |
| Data exclusions | No data was excluded from the analyses. |
| Replication | Nearly all the findings reported in this study correspond to computational analyses of next generation sequencing data, and we thoroughly describe the methods and provide relevant data and code when necessary to reproduce our findings, including raw seq data (genome sequencing, RNA-seq and ATAC-seq; see Data Availability). ATAC-seq data was obtained from two single embryos (one at each stage) due to the scarcity and difficulty in obtaining hagfish embryonic material (few embryos per year, obtained from a single laboratory in the world - Shigeru Kuratani laboratory at RIKEN, Japan-); in this case, we divided the nuclei into two pools per sample in order to at least provide technical replicates. Regulatory profiling analyses have been performed in two different laboratories (RIKEN, Japan, and Institute of Zoology, China) to confirm reproducibility. |
| Randomization | This study did not involved experimental groups, thus experiments were not randomized. |
| Blinding | Investigators were not blinded to allocation during experiments and outcome assessment because this study did not involve comparisons between treatment and control groups. Blinding was thus not applicable to this study. |

# Reporting for specific materials, systems and methods

We require information from authors about some types of materials, experimental systems and methods used in many studies. Here, indicate whether each material, system or method listed is relevant to your study. If you are not sure if a list item applies to your research, read the appropriate section before selecting a response.

### Materials & experimental systems

| n/a | Involved in the study |
|---|---|
| ☒ | ☐ Antibodies |
| ☒ | ☐ Eukaryotic cell lines |
| ☒ | ☐ Palaeontology and archaeology |
| ☐ | ☒ Animals and other organisms |
| ☒ | ☐ Clinical data |
| ☒ | ☐ Dual use research of concern |
| ☒ | ☐ Plants |

### Methods

| n/a | Involved in the study |
|---|---|
| ☒ | ☐ ChIP-seq |
| ☒ | ☐ Flow cytometry |
| ☒ | ☐ MRI-based neuroimaging |

## Animals and other research organisms

Policy information about studies involving animals; ARRIVE guidelines recommended for reporting animal research, and Sex and Gender in Research

| | |
|---|---|
| Laboratory animals | No laboratory animals were used in this study. |
| Wild animals | Sexually mature adults (unkown age) of the inshore hagfish, Eptatretus burgeri, were captured from the Japan Sea, off the Shimane coast in Japan, on field trips done on different years (stated in the manuscript). Animals were captured with tradicional hagfish traps in determined spots and brought to the lab at RIKEN, Kobe, Japan, where they were kept. A ratio of 3:1 females to males were kept in cages in the sea in their natural environment and eggs were retrieved from the cages a few months later (animals caputred in August, eggs collected in November). Eggs were brought back to the lab where they were assayed at appropiate stages. Adults in the lab were euthanized with an overdose of MS-222 (Tricaine) with the approval of RIKEN ethical committee since they needed to be dissected for the sampling of tissues of interest for the study (brain, testis, liver, skeletal muscle, gills, heart, intestine, kidney). |
| Reporting on sex | Genome data was obtained from the testis of two male individuals. This was confirmed dissecting the animal and extracting the testis. For embryonic material, sex was undetermined. |

| Field-collected samples | Adult hagfishes and eggs are kept in tanks of artificial sea water at 16 C, covered to keep a 24-hour dark cycle. |
| --- | --- |
| Ethics oversight | The sampling and experiments were conducted according to the institutional and national (Japan) guidelines for animal ethics, approved by the RIKEN Animal Experiments Committee (approvals H14-25-23 and H14-25-25). |

Note that full information on the approval of the study protocol must also be provided in the manuscript.

