## [Peer Review File · Nature Ecology & Evolution]

Peer Review Information

Journal: Nature Ecology & Evolution

Manuscript Title: Hagfish genome elucidates vertebrate whole genome duplication events and their evolutionary consequences.

Corresponding author name(s): Juan Pascual-Anaya

Editorial Notes:

Reviewer Comments & Decisions:

Decision Letter, initial version:
--

23rd May 2023

Dear Juan,

Your manuscript entitled "Hagfish genome illuminates vertebrate whole genome duplications and their evolutionary consequences" has now been seen by three reviewers, whose comments are attached. The reviewers have raised a number of concerns which will need to be addressed before we can offer publication in Nature Ecology & Evolution. We will therefore need to see your responses to the criticisms raised and to some editorial concerns, along with a revised manuscript, before we can reach a final decision regarding publication.

We therefore invite you to revise your manuscript taking into account all reviewer and editor comments. Please highlight all changes in the manuscript text file in Microsoft Word format.

* If you have not done so already please begin to revise your manuscript so that it conforms to our Article format instructions at <http://www.nature.com/natecolevol/info/final-submission>. Refer also to any guidelines provided in this letter.

[REDACTED]

Nature Ecology & Evolution is committed to improving transparency in authorship. As part of our efforts in this direction, we are now requesting that all authors identified as 'corresponding author' on published papers create and link their Open Researcher and Contributor Identifier (ORCID) with their account on the Manuscript Tracking System (MTS), prior to acceptance. ORCID helps the scientific community achieve unambiguous attribution of all scholarly contributions. You can create and link your ORCID from the home page of the MTS by clicking on 'Modify my Springer Nature account'. For more information please visit www.springernature.com/orcid.

Yours sincerely,

[REDACTED]

Reviewer expertise:

Reviewer #1: genome sequencing, evolution of vertebrate genomes including WGD

Reviewer #2: evolution of vertebrate genomes, including WGD

2Reviewer #3: fish evolutionary genomics, genome sequencing

Reviewers' comments:

Reviewer #1 (Remarks to the Author):

Summary of the manuscript:

In this manuscript, the authors sequenced the inshore hagfish genome, performed comparative genome analyses, and discussed several long-standing debates regarding early vertebrate evolution. First, the authors studied the timing of the two WGD events (called 1R and 2R) during early vertebrate/gnathostome evolution and concluded that the cyclostome lineage diverged from the gnathostome lineage between 1R and 2R, which is consistent with recent studies (Simakov et al. *Nat Ecol Evol* 2020; Nakatani et al. *Nat Commun* 2021). Second, the authors studied duplication events in the cyclostome lineage, and concluded that the cyclostome genomes were shaped by two cyclostome-specific WGD events (called CR1 and CR2) in addition to 1R. The inferred duplication scenario is completely different from previous studies: i.e. chromosome-scale duplications (Smith et al. *Nat Genet* 2018) and genome triplication (Nakatani et al. *Nat Commun* 2021). Finally, the authors studied how developmental complexity and morphological diversity increased in cyclostomes and gnathostomes. They performed ATAC-seq from hagfish embryos and suggested that both cyclostomes and gnathostomes likely increased regulatory complexity independently after WGD events. Next, they studied morphological diversity and found that only the 2R event likely had a large impact on morphological evolution in vertebrates.

The main contribution of this paper is the discovery of two cyclostome-specific WGD events (CR1 and CR2). The authors also discussed potential roles of WGD on the morphological and developmental evolution of gnathostomes: It is possible that WGD events are not causally related to morphological complexity; alternatively, the nature of duplication events (2R is allotetraploidy but 1R/CR are autotetraploidy) might have been important.

General Comments:

Whole-genome duplication events during evolution of early vertebrate lineages (i.e. cyclostomes and gnathostomes) have been hotly disputed but remained unresolved for a long time. Many researchers anticipated that cyclostome genomes (i.e. lampreys and hagfishes) are the key. However, sequencing projects of cyclostome genomes have reported quite different duplication scenarios: The sea lamprey genome analysis suggested that the lamprey genome was shaped by accumulation of segmental or chromosome-scale duplications (refs 4 and 17); the Arctic lamprey genome analysis suggested that the proto-cyclostome genome was shaped by genome triplication (ref 6); and now the inshore hagfish genome analysis proposes two cyclostome-specific WGD events (CR1 and CR2) in this manuscript. It is not surprising that some discrepancies arise from these genome analyses performed by independent research groups, considering the extremely old timing of 1R/2R/CR1/CR2. Therefore, it is important for the current manuscript to clarify similarities and discrepancies in methods and results of relevant studies, as described in more detail in Major Comments.

As the authors discussed in the manuscript, the inshore hagfish genome is crucially important for understanding the biology and evolution of early vertebrates, and has a large impact on related fields.

3Thus, I hope the manuscript will be eventually published after addressing the review comments written below.

Major Comments:

1. The authors studied the pre-1R ancestral genome, timing of 2R, and existence of CR1/CR2 events by comparative genome analyses. Similar analyses have been performed by Smith et al. (ref 17), Sacerdot et al. (ref 11), Simakov et al. (ref5), and Nakatani et al. (ref 6). In fact, the authors have already compared their analyses with these relevant previous studies in the manuscript (e.g. in Lines 249-253 and in Supplementary Table 34); however, sometimes citations and discussions seem to be insufficient in the main text (please see minor comments written below). In particular Smith et al. proposed that the lamprey genomes were shaped by segmental and chromosome-scale duplications, and Nakatani et al. proposed that a genome triplication occurred in the early cyclostome lineage. The authors' analysis (Fig 4f) seems to be compatible with these previous models, and therefore it is important to discuss if the hagfish genome analysis can reject the previously proposed models.

2. If the hagfish genome analysis fails to reject the possibility of the previous models, the authors can discuss the three alternative duplication scenarios, underlying assumptions, methods, biological significance, etc. in Discussion or in Supplementary Information.

3. In the analysis for detecting fusions between 1R and 2R, the authors identified three or four paralogous chromosomes in chicken or gar (Line 217 in Main Text). In previous studies three paralogous chromosomes were regarded as evidence of chromosome duplications in refs 4 and 17. On the other hand, Sacerdot et al. (ref 11) increased the number of analyzed genomes to address this issue and reconstructed 54 post-2R chromosomes. Besides, the number of fusions between 1R and 2R is different (i.e. seven in ref 11 and eight in the authors analysis). Thus, it might be necessary to check consistency by comparing Fig 3D in ref 11 and Supplementary Figure 15.

4. In the relaxed molecular clock analysis, the authors inferred the age of 2R and other WGD events based on the duplication nodes of the gene trees. However, if 2R is allotetraploidy as suggested in refs 5 and 6, a "duplication" node in a gene tree might actually be species divergence of two post-1R/pre-2R ancestors that eventually underwent allotetraploidy. In addition, if 1R/CR1/CR2 are autotetraploidy, the estimated time may point to the timing of diploidization after WGD. Therefore, there might be additional uncertainty when we compare the inferred time of WGD with e.g. 'ostracoderm' fossil record.

Minor Comments:

1. Page 3, Line 81: "debate centres on whether 2R predated (refs 10 and 11)"

I think ref 60 also argued that 2R predated gnathostome-cyclostome divergence.

Smith, J. J. et al. Sequencing of the sea lamprey (*Petromyzon marinus*) genome provides insights into vertebrate evolution. *Nat Genet* 45, 415–421 (2013).

2. Page 3, Line 85: "support 2R occurring before (ref 11) or after (ref 5)"

I guess that the authors forgot to cite ref 60 here.

Smith, J. J. et al. Sequencing of the sea lamprey (*Petromyzon marinus*) genome provides insights into

4vertebrate evolution. Nat Genet 45, 415–421 (2013).

3. Pages 2-3, Lines 87-89: "Analysis of the Arctic lamprey (*Lethenteron camtschaticum*) genome has suggested that 2R occurred in the gnathostome lineage and that two additional WGD events might have occurred in the lamprey lineage (refs 6, 18)"

I could not find any sentences in refs 6 and 18 that support the authors' sentence. It might be possible that the authors misunderstand those papers. For example, the authors' sentence seems to be slightly different from the following sentence in ref 18: "it is possible that the first two rounds of whole-genome duplication may have occurred independently in the lamprey and gnathostome lineage."

4. Page 4, Lines 91-92: "has impeded assessment of the conservation of these events during early vertebrate evolution"

The meaning of "assessment of the conservation of these events" is not clear to me. Is it possible to clarify what the lack of the hagfish genome has impeded? I guess the authors actually meant "during early cyclostome evolution".

5. Page 5, Line 132: " $P < 2.2 \times 10^{-16}$, two-sided Wilcoxon sum-rank test"

Please correct this typo: 10-16. I had never heard of "sum-rank test". Isn't it rank sum test?

6. Page 5, Line 134: "hagfish genome and than"

Typo?

7. Page 6, Line 161: "++81, ++86 and ++98"

Where can I find the definition of "++"?

8. Page 7, Lines 199-200; "This implies that there were likely more than two WGDs in early vertebrate evolution (refs 6, 18)"

Please clarify that the authors counted hexaploidization proposed in ref 6 as two separate events here.

9. Page 8, Line 205: "with either 10-13 (refs 4, 15)"

The following paper should be included.

Putnam, N. H. et al. Sea anemone genome reveals ancestral eumetazoan gene repertoire and genomic organization. *Science* 317, 86–94 (2007).

10. Page 8, Line 205: "or 17 (refs 5,11,14) ancestral, pre-duplicative chromosomes"

Ref 6 with 18 ancestral chromosomes may be added here.

11. Page 8, Lines 207-209: "The lack of a hagfish genome assembly has precluded confirmation or rejection of the predictions based on lamprey genomes (ref 37)"

I am grateful if the authors could indicate the exact sentences in ref 37 that justify this citation, since I don't have time to confirm before the deadline for submitting this review.

12. Page 8, Lines 215-216: "largely consistent with previous studies (refs 5,11,14), but with minor differences (Supplementary Table 34, Supplementary Information)."

5Supplementary Table 34 compares the reconstructions proposed in refs 5, 11 and 6 but not 14?

13. Page 8, Line 218: "confirming the 2R event (ref 39)"

Similar observation was previously said to be consistent with accumulation of chromosome-scale duplications as discussed in ref 4. Besides, I am not sure if ref 39 states that the presence of four or three paralogous chromosomes in chicken or gar confirms the 2R event.

14. Page 8, Line 235: "are compatible with 1R"

Are compatible with shared 1R?

15. Page 9, Line 237: "8.2% of cyclostome genes are compatible with a 2R history"

It would be helpful if the definition of "cyclostome genes" is described also in the main text (in addition to the description in the figure legend).

16. Page 9, Lines 241-249: "When assessing how hagfish and lamprey chromosomes descended from the 17ACs ... suggesting that the lamprey (and, thus, cyclostomes) diverged after the 1R but before all eight post-1R/pre-2R fusions detected in gnathostomes."

If the authors decide to not use the hagfish genome for this analysis, what is the difference (or improvement) from similar analyses described in previous studies (refs 5 and 6)?

17. Page 9, Line 261: "at least three WGD events (refs 6, 18)"

Three WGD events were suggested in ref 18, but it is not clear how the authors counted three WGD events in ref 6.

18. Page 9, Lines 264-265: "we find that multiple chromosomes are direct descendants of each AC in both cyclostome groups."

The authors wrote that the hagfish chromosomes are highly rearranged. Which hagfish chromosomes (i.e. which Hi-C clusters) are direct descendants of each AC? What is the meaning of "direct" here?

19. Page 10, Lines 286-287: "suggesting that, at least, a second WGD occurred in cyclostomes"

This observation seems to be consistent with accumulation of chromosome-scale duplications proposed in refs 4 and 17.

20. Page 10, Lines 290-291: "suggesting that cyclostome genomes have been shaped by three shared WGD events."

It seems that other models proposed in refs 4, 6 and 17 are also possible. After thinking about this argument, I think it might be necessary to add a clause like "assuming that chromosomes can be duplicated only by WGD."

21. Page 11, Lines 309-310: "Overall, intra- and interspecific gene retention profile analyses indicate that two independent WGD events took place in the cyclostome stem-lineage;"

The result of the authors' analysis seems to be consistent with previously proposed models (refs 4,6 and 17). In order to avoid confusions, I suggest adding "under the assumption that cyclostome chromosomes were duplicated only by WGD."

22. Page 13, Line 370: "all three of WGD events"
Which three? 2R, CR1 and CR2?

23. Page 13, Line 372: "2R occurred very early within the gnathostome stem-lineage"
Since refs 5 and 6 have suggested that 2R was allotetraploidy (i.e. hybridization of related species followed by genome doubling to recover the homologous chromosomes), it is not obvious that the inferred date corresponds to the genome duplication event.

24. Page 13, Lines 379-380: "First, it reveals a robust and accurate history of WGD events in early vertebrates, demonstrating that cyclostomes diverged from gnathostomes after the 1R, but before 2R."

What is the meaning of robust and accurate here? Is it related to Page 54, Lines 1127-1128 in Supplementary Information?

25. Page 13, Lines 381-382: "This is consistent with early studies on the matter (refs 12, 14), ending debate over the timing of 2R"
Maybe ref 13 is missing? Also, refs 5 and 6 can be mentioned around here?

26. Page 13, Lines 382-283: "evidencing two additional WGD events, CR1 and CR2, in stem-cyclostomes"
I think that the previously proposed models (refs 4, 6 and 17) should also be mentioned in Discussion, if they were not rejected by the hagfish genome analysis.

27. Page 14, Lines 397-399: "Another possibility is that the 2R event is different in nature from the 1R and CR events...., while 1R was likely an autopolyploidy event, 2R was an allopolyploidy"
It might be necessary to discuss if the CR events were auto-polyploidy before the argument that the 2R event is different from 1R and CR.

28. Page 20, Lines 574-575:
Please add the common name of *X. tropicalis*, *D.* should be *Danio*, *mili* should be *millii*, and please delete the period after *Petromyzon*.

29. Page 23, Line 655: "x10-5"
Is it correct?

30. Page 27, Lines 785-787: "All stem gnathostomes were adjudged to have undergone ... molecular clock analysis"
The inferred timing of 2R is the date of divergence between the two progenitor species, which may not be close to the timing of allotetraploidy. Can we still safely assume that all stem gnathostomes post-dated 2R?

31. Page 39, Figure 4a
The figure explains the case for 4-way paralogous trees, but not for 3-way paralogous trees. If chicken chromosome 27 and gar LG15 are deleted from the figure, how do we classify the purple and yellow arrows in Fig 4a? Purple and yellow arrows will be 1R compatible and 2R uninformative, and in this

7case the purple and yellow arrows are not counted in "Phylogenetic Support for shared 2R." Is it correct?

Supplementary Information

32. Page 33, Line 665: " $e \leq 10^{-6}$ "

Typo?

33. Page 53, Line 1106: "two rounds of WGDs"

Two rounds of WGD?

34. Page 54, Lines 1143-1144: "manually split these homologous gene into two parts given the 20%, 30%, 40%, 50%, 60%, 70% and 80% quantile rank position"

The authors wrote their method calculates an upper bound of the overlapping ratio (Page 55, Line 1150). If we really need an upper bound or a conservative estimate, it might be easier to understand if fission simulations are performed at all gene boundaries.

35. Page 59, Lines 1235-1245

Is it possible to calculate the theoretical distribution of paralogous chromosome fusions?

36. Page 61, Line 1278: "four gnathostome species"

What is the reason for choosing these four species?

37. Page 65, Line 1359: "Branchiostoma_lanceolatum"

Typo? Branchiostoma lanceolatum

38. Page 91, Line 1966 Supplementary Files

It would be helpful if the URL for downloading these supplementary files is written between Line 1966 and Line 1967.

39. Page 108, Supplementary Figure 9:

HoxC6, C3, C1, D5 and D2 are missing in the Crown Gnathostome Ancestor?

40. Page 117, Line 2256: "circus"

Circos?

41. Supplementary Table 31, "Note, chicken chrZ is mainly orthologous to gar LG4.2 and chicken chr4.2 is mainly othologous to gar LG2.2."

Is it correct? The table looks different. Also, please check the spelling of orthologous.

Reviewer #2 (Remarks to the Author):

"Hagfish genome illuminates vertebrate whole genome duplications and their evolutionary consequences" by Yu et al.

Reviewer: Aoife McLysaght (I'm happy to reveal my name to the authors ... I think they'll guess anyway :-))

This manuscript describes the sequencing and genome analysis of the inshore hagfish, *E. burgeri*.

This is the first Hagfish genome to be published, both in terms of the data availability (the genome data has generously been available through ensembl.org for several years) and in terms of the manuscript release (this appeared on BioRxiv about a week before another hagfish genome paper was published there, though of a different hagfish species). Thus this is an intrinsically important paper as the hagfish lies in an important position in the vertebrate tree, forming a clade with the lampreys. Additionally, as pointed out in this manuscript the GC content of hagfish is not as extreme as lampreys, which should make this genome very useful for comparative genomics of vertebrates. I am confident that this paper will attract a large amount of attention and will be highly cited.

The summary of my view is that this paper should definitely be published in your journal, but first the manuscript requires some important improvements.

In some aspects this manuscript (as distinct from the analyses) appears to be a bit rushed. One consequence of this is that some of the existing literature is not correctly cited or is cited inconsistently (in some places of the manuscript, but not others even though some papers are cited), or not cited at all. The authors need to pay close attention to acknowledging the work of others. This is just one symptom of a not terribly well written paper - the authors can definitely fix this with some care and attention. There are some parts where the manuscript doesn't flow well, where the figure legends are a bit lacking, and where the explanations too sparse. I have pointed out a good number of these below.

Nakatani et al (ref 6) had several findings that are mis-cited here: they showed that only the 1R WGD was shared with cyclostomes, and this finding was definitive due to the non-sharing of several chromosomal fusions that occurred after 1R but before 2R; they also showed that there was additional polyploidy ancestral to the cyclostomes and inferred that this was a hexaploidy/triplication (notably the data presented by Yu et al are consistent with hexaploidy, more on that later) but the manuscript cites this paper in reference to 'additional WGD events' in lamprey, but this is triplication, not duplication. Nakatani was the first to propose hexaploidy in cyclostomes; Nakatani et al were also the first to infer that the photo-vertebrate genome contained 18 chromosomes, an idea which has since been accepted by Simakov as a co-author, but does not originate with Simakov, as implied by the citation here. I feel honour-bound to point out that I am corresponding author on Nakatani et al, and I would never abuse the role of reviewer to request citations unnecessarily, but in this case the paper is very relevant and it is important in my view to cite it accurately.

Gene complement analysis (line 153 onwards) - this is interesting but it would benefit from a comparison with other results, for example those from the whale shark paper

Tan, M. et al. The whale shark genome reveals patterns of vertebrate gene family evolution. *Elife* 10, e65394 (2021).

Inference of the proto-vertebrate chromosomes (PVCs)/ancestral chromosomes (ACs) and analysis of 1R and 2R:

The details of the PVCs are opaque and the text is muddled. The text talks about 17 inferred proto-gnathostome chromosomes, whereas it is clear from supp figure 5 that they are actually referring to proto-vertebrate chromosomes, (and instead there are 24/26 proto-gnathostome chromosomes in their reconstruction ... trying to count in the figure). Additionally, they vary between 16 (e.g. Fig 4) and 17 Acs and there is only a small note about this in the supplementary information (line 955) that I could find. The authors clearly went to a great deal of effort to reconstruct these ACs/PVCs, but it isn't clear to me what they gain over using the available published reconstructions or how their method might be superior. I note that the supp info compares their reconstruction to those of Simakov, Nakatani and Sacerdot (supp table 34) but these are not all cited on line 215 (main text). The comparison is very sparse - just naming equivalent chromosomes in the various reconstructions, without any detailed comparisons, so it is hard to know what is added here with another new construction, and one that has a different number of chromosomes to the others.

Given the proto-gnathosome/proto-vertebrate confusion I then find the rest of the paragraph (line 217 onwards) hard to follow. I am struggling to find the details of the pre-1R fusion in the supplementary material and I am thus unsure of the strength of evidence to infer this.

With regard to the post-1R pre-2R fusions, these have been previously described, but this is only properly acknowledged in the supplement. This paragraph in the main text (line 222 and around) does not acknowledge this. This must be corrected. Similarly, the following paragraphs that use the non-sharing of the fusions to infer the non-sharing of 2R with cyclostomes previously appeared in ref 6 (Nakatani) at the very least.

In the section on cyclostome-specific WGD (line 260 onwards) the data presented are consistent with genome TRIplication (hexaploidy) as first proposed by Nakatani et al, but these authors do not discuss that. In Fig 4f we see that the maximal multiplicity of proto-vertebrate chromosomes to cyclostome chromosomes is 6, which is consistent with hexaploidy. Could they please analyse their data in the light of that published hypothesis? The Hox cluster data are also consistent with hexaploidy. Is there actually any reasonable evidence to support CR1+2 over hexaploidy? If the authors have gene tree topology evidence or other more detailed synteny evidence then they should present it. As it stands, the data provided look to be more supportive of hexaploidy.

Dating the polyploidy: the approach is not invalid, but it is not conservative. It is not clear if the approach of using the posterior results from the species tree based dating is entirely appropriate. I am not convinced it is wrong to do this, but I do think it is the opposite of being conservative about the ability to date the WGD events because these posteriors are likely more constrained in time than the raw, purely fossil-based priors. I would be interested to see 1. Analysis of this type using the priors from the species tree analysis (how does this affect the date range for WGDs?), and 2. Runs under the prior (i.e. the posterior of the original species tree analysis) presented to show that using the posteriors of the main analyses are sufficiently diffuse so as not to inform the WGD dates. In addition

10to this – you should probably consider that if CR2 is not well supported then the CR1 and CR2 dating is inappropriate.

In the Discussion (line 378 onwards) some of the claims are overstated in terms of novelty: It was previously demonstrated that cyclostomes diverged after 1R and before 2R; additional polyploidy in cyclostomes has previously been reported.

With regard to the uncoupling of polyploidy and morphological evolution, note also the studies on paddlefish and sturgeon - sturgeon is one of those with hardly any morphological change yet has experienced polyploidy.

Minor comments:

line 59: add 'often assumed to be' before 'causally'

line 61: it would be a quite eccentric view these days to doubt the veracity of 2R

line 255: Nakatani was the first to infer 18 proto-vertebrate chromosomes.

line 324: appears to be an incomplete sentence or bad syntax. Higher than what?

note on nomenclature: WGD refers to 'duplication' but not every polyploidy is tetraploidy. (duplication) so sometimes WGD is not an appropriate acronym

line 660 : cite the study where the Hox genes were sourced

Fig 4a is confusing. I wonder if there are missing purple and orange arrows? My interpretation is that there should be another purple arrow on the branch directly below the first one, and the same for all the branches below the orange arrow - shouldn't they all have orange arrows? It is also unclear what is meant in the legend when it is written "possible positions were test genes can group" - what is 'grouping' in this context?

In fig 4 there are only 16 ACs presented but this is very confusing because there is no mention of there being only 16 anywhere else in the main text (including the methods of the main text) - this only appears, not well explained (that I can find) in the supplement.

Fig5e - explain 'red', 'subf' 'spec' in the legend

Reviewer #3 (Remarks to the Author):

This article presents the first genome analysis of a hagfish, representing one of the two major lineages of extant agnathan, cyclostome vertebrates and one of last major groups of vertebrates to be genomically explored. By its study subject itself, this work will be of highest interest and impact to the

11vertebrate genome evolution community.

Major parts of the present study use the hagfish genome assembly to investigate and clarify major patterns of genome evolution in early vertebrate lineages, especially with regard to the timing and patterns of whole genome duplications (WGD) within cyclostomes as well as in the jawed vertebrates, the gnathostomes. By the addition of the hagfish genome, the current study confirms cyclostome monophyly and that only the first round of WGD (1R) was shared among all extant vertebrates. According to the present study, the cyclostomes then underwent two additional rounds of WGD (CR1, CR2) before the divergence of hagfish and lampreys, explaining for example their high number of Hox clusters. The authors further find that the 2R event was specific to the gnathostome lineage in agreement with previous studies (Simakov et al. 2020; Nakatani et al. 2021). The study then goes on to investigate patterns of post-WGD evolution in terms of gene gain and retention, gene expression, and regulatory evolution. Most interestingly, phylogenetic analyses integrating data from the fossil record provide evidence that this 2R WGD predates the massive morphological diversification across the lineage leading to extant gnathostomes, suggesting that 2R may have spurred this extensive exploration of morphological space.

Overall, I think that the study will be of highest interest to the broad readership of Nature Ecology and Evolution. However, the manuscript needs to be rewritten for a more general audience, especially for more clarity. In several places, I needed to go back and forth between the main text and the supplements to understand the main conclusions to be gained from the analyses. The authors should strive to let the main text stand for itself.

General points:

1. Genome assembly:

Multiple versions of the hagfish genome have been generated. Publicly available has been an intermediate genome assembly version in Ensembl for quite a while now (version 3.2). The final assembly version used for the analyses in this manuscript (v4.0) does not seem to be publicly available yet and accession numbers for this final version are not available as far as I can tell.

2. Hagfish and cyclostome biology:

Comparatively small parts of the article are devoted to the genomic exploration of hagfish biology itself which may be presented in another study (e.g., genes involved in the formation of the iconic "slime eel" slime, etc.). At some instances, the analyses show that the gene complement of cyclostomes is quite reduced compared to that of other vertebrate lineages (e.g., l. 172 "suggesting that a strong asymmetric reduction of gene complements accompanied the early evolution of the group"). I think it would have been interesting to discuss these results in more detail and connect them to the general understanding that cyclostomes and especially hagfishes represent rather morphologically derived and reduced lineages.

3. Genomic comparisons:

For the comparative genomic analyses, a lot of other vertebrate and non-vertebrate key taxa are included. I appreciate the inclusion of Supplementary Table 1 to list all the resources. However, information should be added whether the included versions of these species are chromosome-level

12genome assemblies or not.

This is particularly relevant for the synteny-based analyses. For example, the elephant shark genome assembly used does not seem to be the more recent, chromosome-level version (Nakatani et al. 2021). An alternative assembly for Chondrichthyes could have been the little skate genome. Likewise, the sea lamprey genome assembly also seems to have been updated since the versions used for the analyses here, and the Arctic lamprey (Nakatani et al. 2021) could have been included as an additional cyclostome genome.

To be clear, I do not necessarily ask the authors to redo or expand their analyses, but some more clarity and discussion on the limitations of the assemblies used in the present study would be helpful.

4. Vertebrate WGD nomenclature:

The cyclostome-specific genome duplications here are termed CR1 and CR2.

1R is the WGD shared among all crown vertebrates, 2R is the WGD in the lineage leading to gnathostomes. Within gnathostomes, later WGDs have been called for example 3R for the teleost fish WGD, in an enumerating manner starting from the vertebrate ancestor.

CR1 and CR2 hence are the 2nd and 3rd WGD in the cyclostome lineage and to me, it would make more sense to call them CR2 and CR3, respectively. Likewise, it would be good to qualify 1R as, for example, V1R (vertebrate 1R) and G2R (gnathostome 2R). I think this study provides an excellent opportunity to rework the nomenclature of vertebrate WGDs.

5. Hox cluster evolution:

The current study clarifies the orthology of the six hox clusters among cyclostomes. However, I could not find a model of how these six clusters were generated through the three cyclostomes WGDs. Following the 1R-CR1-CR2 model, we could expect to see up to 8 hox clusters in cyclostomes, so 2 clusters were lost somewhere along the way to the cyclostome ancestor. Using your map of cyclostome chromosomal relationships to the ancestral chromosomes (ACs), it should be possible to build a model of hox cluster evolution through the three rounds of WGD leading to cyclostomes.

6. Post 2R morphological evolution:

One of the most exciting parts of the current study was the integration of fossil data which led the authors to conclude that basically all non-cyclostome vertebrate lineages (extinct and extant) are derived from the 2R WGD (Fig. 5h). This has important implications for the significance of the 2R event for the morphological diversification of vertebrates. However, I found this part particularly difficult to follow and to understand how its major finding was derived. Please discuss your approach in more detail and make it more clear how you derived this major conclusion. As written, the relevant section in the main text (starting l. 350) does not clearly convey how you came to this far-reaching conclusion. Even after reading the supplementary text multiple times, I am still not fully clear how it was concluded that basically all non-cyclostome vertebrates are 2R-derived. Please rework this part so

13that it becomes digestible for the non-expert.

7. Nature of WGD events:

a) Simakov et al. 2020 concluded that 1R was an auto- and 2R an allotetraploidization event. Nakatani et al. 2021 later came to similar conclusions. The current study, however, does not make an inference for establishing the nature of the 1R and 2R events. I thus wonder if the current analyses are at least consistent with and/or supporting Simakov's model? I find this particularly important since the allotetraploidization of 2R from previous studies is cited here as possible explanation for the impact of 2R on morphological evolution. I therefore would like to see a more in-depth analysis and discussion whether the authors' chromosome evolution data here also support an allotetraploidization for 2R (and an 1R autotetraploidization).

b) Nakatani et al. 2021 conclude that following 1R, the extant cyclostome genome was generated by hexaploidization which appears to be at odds with the CR1-CR2 model derived here. I would like to see some additional discussion why the hexaploidization model has been rejected in the current study.

8. Gene regulatory evolution:

I was missing some clarity on the definition of distal regions among the different species analyzed for accessible chromatin regions. Have they been scaled in some way to genome size? Maybe I missed this information, but it was hard to find.

9. Hagfish genome:

a) Karyotype:

I found it difficult to understand the description of the expected hagfish chromosome number to which the assembly is matched. Please say it more clearly in the text. I understand that the germline genome is $n=26$ so that we could expect the germline assembly here to have 26 scaffolds representing these chromosomes? The somatic genome should then have $n=18$ after elimination of 8 chromosomes. The description in the supplements was particularly confusing. Please revise karyotype descriptions and expectations for the assembled genome.

b) Sex chromosomes:

Please also provide a clearer explanation why an XY sex determination system is suggested from the genome assembly. Also, do you mean an XY system with highly sexually dimorphic chromosomes? While read-depth analyses could potentially identify such a system, what if the sex determination region is small and read-depth is only prevalent in a small portion of the chromosome (instead of showing a strong effect across the entire chromosome/cluster)? I think this part of the genome analysis is too preliminary and should either be expanded (e.g., by a sliding window analysis across clusters to identify potential regions with read-depth differences) or removed from the current study.

Additional Specific Points:

- I. 109/110: 16,500 predicted genes seem a rather small number compared to gnathostomes. It

14would be good to compare here to the predicted total gene numbers in Sea and Arctic lampreys. Is this low number due to more gene "hiding" in hard to assemble microchromosomes? See also 2. above.

- l. 124, better say: "... Although having an overall GC content similar to that of the lamprey..."
- l. 131, say: "The hagfish genome..."
- l. 134: "...larger size of hagfish genome than those of..." (delete 'and')
- l.135/6: I'd avoid this statement to say that hagfish may be a better model than lamprey. Later in the text, it is said that lamprey is more suited for other types of analyses (l.245-46). Rather say something along the lines that hagfish genome provides essential, complementary information to genomes of lampreys.
- l. 144/145: Briefly say what type of data are underlying the Bayesian phylogenetic analysis, even if it is explained in more detail in the supplements or elsewhere. Figure legend 2 also doesn't provide this information.
- l. 158: lowest amount of gene losses among what?
- l. 159/60, for clarity, say for example: "... highly retained novel gene families (also known as novel core genes, i.e., genes that are not lost in descendant lineages, by convention indicated with ++)"
- l. 168, better say: "... gnathostomes and cyclostomes convergently evolved independent adaptive immune systems..."
- l. 174: "Inferred rates of gene duplication..." Say more explicitly here that you do not distinguish types of duplication (at least this is my understanding).
- l. 176, say: "...1R, 2R, and teleost 3R WGD events"
- l. 196/97: "the crown-cyclostome already possessed six Hox clusters, distinct from the crown-gnathostome ancestor" You could make it more explicit that this observation is another evidence for cyclostome monophyly.
- l.199/20: This last sentence is confusing, as it is so vague – what is the point you want to make here? I think it could also just be deleted.
- l. 203: with "pre-duplicative" do you mean "pre-WGD"?
- l. 213: provide some justification for the use of the sea cucumber genome as reference here instead of some other possible non-vertebrate genomes (e.g. Nakatani et al. 2021 and Simakov et al. 2020 used scallop). To be clear, I am not questioning its use but would like to know more why this genome has been chosen here over other options. It does not seem to be a chromosome-level genome

assembly.

- l. 217, for clarity say: "...in the gnathostomes chicken and gar..."
- l. 223-24: Why is the lancelet gene number brought up, this is confusing. Either explain or delete the info.
- l. 226: Is AC = amphioxus? Please explain.
- l. 233-34: Please explain the switch to elephant shark as gnathostome representative here. Also see point 1. above about the elephant shark genome version, which does not seem to be chromosome-level.
- l. 240-41, say: "... of the inferred four pre-1R and..."
- l. 261-263: I understand you mean to say that 2 Hox clusters of a total of potentially 8 Hox clusters after 1R->CR1->CR3 were lost along the way to the LCA of crown cyclostomes. See my comments on Hox clusters above (point 5.).
- l. 293: What would be a comparable OR value for a different vertebrate lineage that is derived from 3 rounds of WGD, e.g. in zebrafish?
- l. 323, better say: "...using one embryos of E. burger at two different stages..."
- l. 324, please explicitly say what your comparison is, higher than amphioxus?
- l. 351-352, please elaborate what you mean with deterministic and permissive.
- l. 383-84: Please explain your reasoning why the key vertebrate innovation would have originated in a post-1R tetraploid stem vertebrate, as opposed to a pre-1R stem vertebrate. What is the evidence that 1R came before these innovations?
- l. 570-71: In which ways were gene models manually curated?
- l. 583-84: Overall, the RSCU analysis is not well explained. Why is it relevant and what is its purpose?
- l. 591: It would be good to also provide statistics for the vertebrata dataset – which will be highly biased towards gnathostomes.
- l. 674-99: It is unclear which figures this part refers to, please include figure numbers.
- l. 711, say: "By definition..."
- l. 754-66: Please make it more explicit that you are only measuring on/off states here (at least this

16is my understanding). It would be helpful to refer to the relevant figure in the supplements that show the definitions (Suppl. Fig. 30).

I am not convinced that you can infer 'subfunctionalization' with data from ingroups only and without evidence for the ancestral, pre-duplication state inferred from an outgroup. As presented, some expression domains could still be novel. Thus, without outgroup information, I'd rather call this 'tissue-specific expression' or 'divergent expression'.

- l. 779: ... the tree is available..."

- Fig. 1. l. 986: lungfish -> hagfish

- Fig. 2: are there also data for gains and losses in Chondrichthyes? What data is this phylogeny based on?

- l. 1052: should be "ohnologous". The term ohnolog does not seem to be defined in the main text. Please do so at first appearance.

- Supplements, l. 145: What do you mean with the sex determination system is "mostly unknown"?

- Supplements, l. 622-631: It would be interesting to know which signaling pathways were differently retained between gnathostomes and cyclostomes?

- Supplements, l. 629, it sounds strange to say "In contrast to vertebrates, gnathostomes retained genes...". Please rephrase.

- Supplements, l.1385-85. The last sentence is unclear, please rephrase.

- Supplements, 5. Regulatory Evolution: This section needs to be reorganized and streamlined as it is difficult to follow how the different subsections connect to each other. For example, is 5.1 the overall discussion of the results?

- Suppl. Fig. 30: Please refer to this figure when presenting definitions of ohnolog fates in the text. I'd also include a scheme to define "mild specialization" as shown in Extended Fig. 8. I couldn't find a definition of this case.

- Suppl. Fig. 31: This figure appears to be essential to understand the dating of WGDs but the figure legend is too minimal to understand the figure. Please explain in detail here and/or in the relevant text section; see also my main point on WGD dating above (6.).

*****END*****Response to Reviewers:

- Reviewers' comments in black;
- Authors' responses in blue.

Line numbers in this document refer to those present in the source .docx files provided, and not to the PDF automatically generated during submission. For the sake of clarity, Word files have been submitted with tracked changes.

Reviewer #1 (Remarks to the Author):

Summary of the manuscript:

In this manuscript, the authors sequenced the inshore hagfish genome, performed comparative genome analyses, and discussed several long-standing debates regarding early vertebrate evolution.

First, the authors studied the timing of the two WGD events (called 1R and 2R) during early vertebrate/gnathostome evolution and concluded that the cyclostome lineage diverged from the gnathostome lineage between 1R and 2R, which is consistent with recent studies (Simakov et al. Nat Ecol Evol 2020; Nakatani et al. Nat Commun 2021). Second, the authors studied duplication events in the cyclostome lineage, and concluded that the cyclostome genomes were shaped by two cyclostome-specific WGD events (called CR1 and CR2) in addition to 1R. The inferred duplication scenario is completely different from previous studies: i.e. chromosome-scale duplications (Smith et al. Nat Genet 2018) and genome triplication (Nakatani et al. Nat Commun 2021). Finally, the authors studied how developmental complexity and morphological diversity increased in cyclostomes and gnathostomes. They performed ATAC-seq from hagfish embryos and suggested that both cyclostomes and gnathostomes likely increased regulatory complexity independently after WGD events. Next, they studied morphological diversity and found that only the 2R event likely had a large impact on morphological evolution in vertebrates.

The main contribution of this paper is the discovery of two cyclostome-specific WGD events (CR1 and CR2). The authors also discussed potential roles of WGD on the morphological and developmental evolution of gnathostomes: It is possible that WGD events are not causally related to morphological complexity; alternatively, the nature of duplication events (2R is allotetraploidy but 1R/CR are autotetraploidy) might have been important.

General Comments:

Whole-genome duplication events during evolution of early vertebrate lineages (i.e. cyclostomes and gnathostomes) have been hotly disputed but remained unresolved for a long time. Many researchers anticipated that cyclostome genomes (i.e. lampreys and hagfishes) are the key. However, sequencing projects of cyclostome genomes have reported quite different duplication scenarios: The sea lamprey genome analysis suggested that the lamprey genome was shaped by accumulation of segmental or chromosome-scale duplications (refs 4 and 17); the Arctic lamprey genome analysis suggested that the proto-cyclostome genome was shaped by genome

triplication (ref 6); and now the inshore hagfish genome analysis proposes two cyclostome-specific WGD events (CR1 and CR2) in this manuscript.

It is not surprising that some discrepancies arise from these genome analyses performed by independent research groups, considering the extremely old timing of 1R/2R/CR1/CR2. Therefore, it is important for the current manuscript to clarify similarities and discrepancies in methods and results of relevant studies, as described in more detail in Major Comments.

As the authors discussed in the manuscript, the inshore hagfish genome is crucially important for understanding the biology and evolution of early vertebrates, and has a large impact on related fields. Thus, I hope the manuscript will be eventually published after addressing the review comments written below.

We are thankful for the reviewer's positive comment. We also acknowledge and apologise for the lack of clarity in our manuscript that led to sharp discrepancies. We have now addressed all issues raised by them and the other reviewers and hope that our results and conclusions picture a more consistent scenario.

Major Comments:

1. The authors studied the pre-1R ancestral genome, timing of 2R, and existence of CR1/CR2 events by comparative genome analyses. Similar analyses have been performed by Smith et al. (ref 17), Sacerdot et al. (ref 11), Simakov et al. (ref5), and Nakatani et al. (ref 6). In fact, the authors have already compared their analyses with these relevant previous studies in the manuscript (e.g. in Lines 249-253 and in Supplementary Table 34); however, sometimes citations and discussions seem to be insufficient in the main text (please see minor comments written below). In particular Smith et al. proposed that the lamprey genomes were shaped by segmental and chromosome-scale duplications, and Nakatani et al. proposed that a genome triplication occurred in the early cyclostome lineage. The authors' analysis (Fig 4f) seems to be compatible with these previous models, and therefore it is important to discuss if the hagfish genome analysis can reject the previously proposed models.

During the last steps of our previous writing process (which was rushed as referee #2 rightly points out), we already realised that our results are in fact more consistent with an hexaploidization origin of the ancestral cyclostome genome, as suggested by Nakatani et al. 2021 for lampreys (ref. 6). However, for the sake of a speedy submission, we did not include that possibility in the original submission. We have now changed the main text to acknowledge this scenario as the most probable one in agreement with ref. 6. A WGD (1R) occurred in an ancestral vertebrate, and after the divergence of cyclostomes and gnathostomes, a triplication event occurred in an ancestral cyclostome. Whether this triplication happened as a single event (e.g., through meiosis errors) or was the result of two events [first, a duplication event (CR1: allo- or autopolyploidy unknown), and then an allopolyploid hybridization between a duplicated ancestor and an unduplicated one (1R+CR1)] remains elusive. We have modified our article accordingly (all sections) to reflect this new scenario.

While this article was under review, a separate hagfish genome analysis was posted as a preprint in bioRxiv (<https://doi.org/10.1101/2023.04.17.537254>). This second analysis, performed on a different hagfish species (*Eptatretus atami*), by a different group, reaches similar conclusions. Furthermore, our gnathostome-specific karyotype phylogenetic analysis demonstrates that crown gnathostomes diverged from a 2R-derived ancestor (two whole genome duplications) and its genome was not the result of segmental duplications (whole karyotype phylogeny). This same conclusion has been reached by the most recent analyses, despite minor differences in the process used (see refs. 5, 6, 11 and *E. atami* bioRxiv preprint). Currently, we strongly believe that a scenario in which the ancestral gnathostome genome architecture is the result of large segmental duplications has been rejected given the evidence of the genome-wide nature of the duplications (all ancient linkage groups / chromosomes are present in 3 or more copies in all descendant vertebrates).

Our results are largely in agreement with those of ref. 11, ref. 5 and ref. 6, and also with the recently uploaded preprint about a different hagfish species' genome analysis (<https://doi.org/10.1101/2023.04.17.537254>), with very minor differences.

2. If the hagfish genome analysis fails to reject the possibility of the previous models, the authors can discuss the three alternative duplication scenarios, underlying assumptions, methods, biological significance, etc. in Discussion or in Supplementary Information.

As we explain above, our scenario largely agrees with Nakatani et al. (2021) and the results of a second hagfish genome analysis (bioRxiv, <https://doi.org/10.1101/2023.04.17.537254>). We believe that after all these analyses (ref. 6 with lampreys; our analysis with *E. burgeri*; and Marlétaz's with *E. atami*), which are the more recent ones and include the most complete cyclostome genomes, the most likely scenario is that of a triplication event in the cyclostome lineage (ref. 6). First, the segmental duplication scenario can be ruled out without question; second, a pan-vertebrate 2R event was inferred either using a limited number of gene trees (Kuraku et al., 2009) or using a parsimony criteria from an ancestral amniote genome reconstruction using only a single lamprey genome as outgroup (Sacerdot et al., 2018), so we strongly believe that the triplication event is the most likely scenario, as proposed by ref. 6, Marlétaz et al., 2023 (bioRxiv, <https://doi.org/10.1101/2023.04.17.537254>) and ourselves.

3. In the analysis for detecting fusions between 1R and 2R, the authors identified three or four paralogous chromosomes in chicken or gar (Line 217 in Main Text). In previous studies three paralogous chromosomes were regarded as evidence of chromosome duplications in refs 4 and 17. On the other hand, Sacerdot et al. (ref 11) increased the number of analyzed genomes to address this issue and reconstructed 54 post-2R chromosomes. Besides, the number of fusions between 1R and 2R is different (i.e. seven in ref 11 and eight in the authors analysis). Thus, it might be necessary to check consistency by comparing Fig 3D in ref 11 and Supplementary Figure 15.

Thanks for your questions and concerns. Regarding the first part of your comment, those cases where there are three remaining paralogous chromosomes in chicken or spotted gar are 6 out of 17 (AC7, AC8, AC9, AC15, AC16 and potentially AC17), but most ACs (11/17) correspond to four chromosomes in these gnathostomes species (Extended Fig. 5A, see below). This genome wide

pattern is better explained by WGD, assuming chromosomal losses after the duplication. Under the chromosomal duplication model in refs. 4 and 17, a one-to-four correspondence is equal to two independent chromosomal duplications after 1R; and a one to three correspondence is equal to one independent chromosomal duplication after 1R. Thus, according to the segmental duplication model there existed 28 ($11 \times 2 + 6 \times 1 = 28$) independent chromosomal duplications in the gnathostome lineage after 1R. 28 independent chromosomal duplication is too large a number and therefore highly unlikely, especially after considering the fact that aneuploidy is scarce in animals and tend to be detrimental due to dosage imbalance (Makino, T., and McLysaght, A. (2010). PNAS). To answer the second part of the reviewer's comment, we have used three representative gnathostome genomes after careful consideration. First, we reasoned that in a macrosyteny comparative approach is key to select species based on karyotype conservation, rather than increasing the number of species to use. Therefore, we used spotted gar, chicken and elephant shark because:

- (1) Elephant shark, spotted gar and chicken represents Chondrichthyes, Actinopterygii and Sarcopterygii lineages, respectively, all major lineages of gnathostomes;
- (2) From the spotted gar genome paper (Ingo Braasch et al., 2016, ref. 62) and elephant shark genome paper (Venkatesh et al., 2014. *Nature*, 505: 174–179. <https://doi.org/10.1038/nature12826>, now cited in the main text as ref. 41), the three genomes share profound chromosome level conservation, clear reflection of their slow evolving nature. The latest little skate shark genome (see figure below) is consistent with this conclusion, see Marlétaz et al., 2023, bioRxiv, <https://doi.org/10.1101/2023.04.17.537254>;

Figure 2b (Marlétaz et al., 2023, bioRxiv), The syntenic orthology relationship between skate, gar and chicken, relying on genes with a significant CLG assignment in regard to amphioxus. Skate chromosomes are coloured by segmental identity and links are coloured by CLG

- (3) Sacerdot et al. (ref 11) included 61 species to help reconstruct post-2R chromosomes, but we found at least some of these species are inappropriate for such analysis. For example, four invertebrate genomes are used, including *Caenorhabditis elegans*, *Drosophila melanogaster*, *Ciona intestinalis* and *Ciona savignyi*. Synteny level conservation of these genomes with gnathostomes have not been reported. Urochordata genomes are known to be fast evolving (Berná and Alvarez-Valin, 2014. *Genome Biol. Evol.* 6(7):1724–1738. doi: 10.1093/gbe/evu122). Teleosts, evolving after third round of WGD, were also used in ref. 11, including zebrafish, stickleback and medaka. Their strong lineage specific evolutionary process may not necessarily provide new information into the reconstruction of ancestral gnathostome karyotype. All in all, the three representative gnathostome species, which have shown excellent synteny level conservation, are more suitable to reconstruct the WGD history of gnathostomes.

Despite all this, the differences in the number of post-2R ancestral chromosomes between our study and ref. 11 is minimal. In our study, this is 50 chromosomes (Supp. Information lines 1572-1573, see below) but 54 in ref. 11. The difference come from the discrepancy on the modelling of post-1R fusions: (i) we detect one additional post-1R fusion event (AC12-AC13) than ref. 11; and (ii) one post-1R fusion event (AC4-AC5-AC12) in our model corresponds to three post-1R chromosomes but only to two post-1R chromosomes in ref. 11 (Suppl. Table 37, see below). So, our reconstructed post-1R chromosomes are reduced by 2 chromosomes (from 27 to 25) compared to ref. 11, and thus by 4 chromosomes after 2R WGD (from 54 to 50). A comparison of our post-1R fusions and those of previous studies have been summarised in Suppl. Table 37. Our detected post-1R fusion events are all reliable because they are inferred based on phylogenetic evidence, which are assumed but not provided from previous studies (see figure below). For example, Sacerdot et al., 2018 proposed what a post-1R fusion scenario would look like and what products it would generate (see below, panel A). This scenario needs to assume two bifurcating trees involving chromosomes A/C/B/D and chromosomes B/D/E/F respectively (see below, panel A'). However, evidence is lacking from their original report. So do other studies (see below, panel B, C and B', C'). Instead, we formally built chromosomal level phylogenies and used them to infer post-1R fusions (see below, panel D).

A) Adapted from Sacerdot et al., 2018, Fig. 4b (partial) :

A') what is assumed in A) :

B) Adapted from Simakov et al., 2020, Fig. 1a (partial) :

Chromosome linkage group	1a	1f	2a	2f
CL9C	GG41P - LOC1* 2079c 8.330-8.367 0.315	GG4 - LOC1* 479c - 3.952 3.128	GG41P - LOC1* X73g 8.482-8.491 0.009	GG42P - LOC2* 479c 8.128-8.148 0.026
CL10	GG4V - LOC2* 479c 8.085-8.102 0.008-0.044	-	GG4V - LOC2* 37631* 0.483-0.492 0.009-0.006	GG42V - LOC2* 479c 8.133-8.183 0.050

B') what is assumed in B) :

C) Adapted from Nakatani et al., 2021, Fig. 4a (partial) :

C') what is assumed in C) :

D) Adapted from our manuscript, Fig. S22 (partial) :

4. In the relaxed molecular clock analysis, the authors inferred the age of 2R and other WGD events based on the duplication nodes of the gene trees. However, if 2R is allotetraploidy as suggested in refs 5 and 6, a "duplication" node in a gene tree might actually be species divergence of two post-1R/pre-2R ancestors that eventually underwent allotetraploidy. In addition, if 1R/CR1/CR2 are autotetraploidy, the estimated time may point to the timing of diploidization after WGD. Therefore, there might be additional uncertainty when we compare the inferred time of WGD with e.g. 'ostracoderm' fossil record

The reviewer is correct; if 2R is an allotetraploid event, our estimate for the timing of 2R will instead represent the time of divergence of the two hybridising lineages. We now only recognise one stem-cyclostome WGD event (triplication). It has been argued that, because of delayed rediploidization post-autopolyploidy, dating attempts provide only a minimum constraint in the

timing of WGD (Redmond et al. 2023. Nat. Comm. 14: 2879). However, the shift from tetrasomic to disomic inheritance is predicated on the accumulation of mutations (Campbell et al. 2019. G3. 9: 2017-2028) facilitating divergence of ohnologues from their shared ancestral state at the point of WGD. Therefore, our dating of autopolyploidy events represents the WGD events themselves, not the timing of rediploidization. Regardless, this debate does not impact on our conclusions since the phenotypic outcome of the CR event, as assayed in our disparity analyses, is negligible compared to 2R. The auto/allopolyploid nature of the 2R event remains unclear and so we retain our dating of the events, but develop the implications of 2R being an allopolyploidy event (which is not inferred from our results) in the discussion.

Minor Comments:

1. Page 3, Line 81: "debate centres on whether 2R predated (refs 10 and 11)"

I think ref 60 also argued that 2R predated gnathostome-cyclostome divergence.

Smith, J. J. et al. Sequencing of the sea lamprey (*Petromyzon marinus*) genome provides insights into vertebrate evolution. *Nat Genet* 45, 415–421 (2013).

Thanks for the comment, but we have to point out that's not completely correct. Although Smith et al. (2013) (ref. 60) favoured that 2R predated the gnathostome-cyclostome split, they did not provide direct evidence. In this regard, the authors acknowledged this in the main text of their article: "Although the less parsimonious scenario involving one or two independent and ancient whole-genome duplication events in gnathostome and lamprey lineages cannot be completely ruled out". They based their preferred scenario on parsimony alone. Moreover, ref. 60 reported results of an analysis of a lamprey somatic genome (from liver), which is incomplete compared to the germ line genome due to programmed DNA elimination. This was later updated with a germline genome by the same authors (Smith et al., 2018: ref. 17), whose analysis did not support the existence of 2R at all but segmental duplications as mentioned above by the referee. In conclusion, we believe that adding ref. 60 here in the main text does not add any further support to 2R predating the gnathostome-cyclostome split.

2. Page 3, Line 85: "support 2R occurring before (ref 11) or after (ref 5)"

I guess that the authors forgot to cite ref 60 here.

Smith, J. J. et al. Sequencing of the sea lamprey (*Petromyzon marinus*) genome provides insights into vertebrate evolution. *Nat Genet* 45, 415–421 (2013).

As explained above, we did not intend to add ref. 60 here, as it did not provide any evidence supporting 2R before the gnathostome-cyclostome split beyond parsimony and the same authors supported a different scenario in ref. 17.

3. Pages 2-3, Lines 87-89: "Analysis of the Arctic lamprey (*Lethenteron camtschaticum*) genome has suggested that 2R occurred in the gnathostome lineage and that two additional WGD events might have occurred in the lamprey lineage (refs 6, 18)"

I could not find any sentences in refs 6 and 18 that support the authors' sentence. It might be possible that the authors misunderstand those papers. For example, the authors' sentence seems to be slightly different from the following sentence in ref 18: "it is possible that the first two rounds

of whole-genome duplication may have occurred independently in the lamprey and gnathostome lineage.”

In regards to ref. 18, it argues in favour of three total duplication events in the lamprey lineage. First, due to the presence of six Hox clusters, they suggest that an additional duplication event might have taken place in an ancestral lamprey after 1R and 2R, as stated in these two sentences in page 4 of that reference: “The presence of at least six Hox clusters in the Japanese lamprey and the sea lamprey suggests that the lamprey lineage has experienced an additional round of genome duplication after 1R and 2R” and “Given our present finding that the ancestor of the Japanese lamprey and sea lamprey lineages experienced an additional genome duplication, the timing of 1R and 2R needs to be reexamined taking into account that lamprey genomes contain a large number of paralogs resulting from the third round of a genome duplication event”. Later on, in the same section, they mention, as the referee cites, that “two rounds of whole-genome duplication may have occurred independently”, but they suggest that the lamprey genome was shaped by three rounds of WGD.

In regards to ref. 6, the referee is right, as they don’t argue in favour of two duplications after 1R, but a triplication event (hexaploidization). We have corrected this along the manuscript.

All in all, we agree that our sentence is confusing and not completely exact, so we have changed it as follows:

Lines 89-105: *“Analysis of the Arctic lamprey (*Lethenteron camtschaticum*) genome has suggested that 2R occurred in the gnathostome lineage while independent WGD events might have occurred in the lamprey lineage (refs 6, 18)”*

4. Page 4, Lines 91-92: “has impeded assessment of the conservation of these events during early vertebrate evolution.”

The meaning of “assessment of the conservation of these events” is not clear to me. Is it possible to clarify what the lack of the hagfish genome has impeded? I guess the authors actually meant “during early cyclostome evolution”.

We agree that our meaning is not clearly stated. The lack of a hagfish genome has precluded researchers from confidently assessing how many of the whole genome duplication events happened in each vertebrate lineage and when they occurred. For example, without a hagfish genome, the events suggested in the analyses using different lampreys cannot be placed before or after the divergence of lampreys and hagfishes with complete certainty. Also, due to the different, and sometimes contradictory, conclusions reached by different groups from the analysis of different lamprey genomes regarding 1R and 2R, we think that “during early vertebrate evolution” makes sense in this context, as the events occurring before the gnathostome and cyclostome divergence, as well as the early events taking place separately in each lineage soon after, are all included in that expression.

To clarify, we have changed our sentence as follows:

Lines 106-108: "However, the lack of a hagfish genome assembly, the only major vertebrate group without a reference genome, has challenged attempts to constrain the number and phylogenetic timing of ploidy events in early vertebrate evolution"

5. Page 5, Line 132: " $P < 2.2 \times 10^{-16}$, two-sided Wilcoxon sum-rank test"

Please correct this typo: 10-16. I had never heard of "sum-rank test". Isn't it rank sum test?

Thanks for pointing it out. It is a two-sided Wilcoxon rank sum test and we have also corrected the text accordingly.

6. Page 5, Line 134: "hagfish genome and than"

Typo?

Thank you for spotting this typo. We have removed "and".

7. Page 6, Line 161: "++81, ++86 and ++98"

Where can I find the definition of "++"?

We have added its definition the first time they appear, also suggested by Reviewer #3:

Lines 199-201: "*highly retained novel gene families (also known as novel core genes, i.e, genes that are not lost in descendant lineages, and by convention indicated by ++)*"

8. Page 7, Lines 199-200; "This implies that there were likely more than two WGDs in early vertebrate evolution (refs 6, 18)"

Please clarify that the authors counted hexaploidization proposed in ref 6 as two separate events here.

We have now changed this sentence to read as follows:

Lines 255-257: "*This implies that the events suggested from the different analyses of the Arctic lamprey genome, two extra WGDs (ref. 18) or a triplication (ref. 6), might have occurred in early cyclostome evolution, before the lamprey and hagfish divergence (refs. 6, 19)*"

In any case, we'd like to make clear that the ancestral hexaploid cyclostome genome originated through at least two events: a pan-vertebrate duplication (1R) and a triplication event in the cyclostome lineage, as originally proposed in ref. 6.

9. Page 8, Line 205: "with either 10-13 (refs 4, 15)"

The following paper should be included.

Putnam, N. H. et al. Sea anemone genome reveals ancestral eumetazoan gene repertoire and genomic organization. *Science* 317, 86–94 (2007).

Thanks for the comment, but we respectfully disagree. Putnam et al. 2007 refers to putative ancestral linkage groups in the last common ancestor of eumetazoans, while in this sentence we are referring to the karyotype of the last common ancestor of vertebrates. It is therefore inappropriate to cite Putnam et al. 2007 here.

10. Page 8, Line 205: "or 17 (refs 5,11,14) ancestral, pre-duplicative chromosomes"

27

Ref 6 with 18 ancestral chromosomes may be added here.

Thank you very much for the suggestion. We have now changed the sentence to read as follows, and added as well ref. 38 (Simakov et al., 2022, which was ref. 39 in the previous version):

Lines 262-264: *“Earlier attempts at reconstructing the ancestral vertebrate karyotype have yielded widely disparate outcomes, indicating either 10-13 (Refs. 4,15) or 17-18 (Refs. 5,6,11,14,38) ancestral, pre-duplicative chromosomes.”*

11. Page 8, Lines 207-209: *“The lack of a hagfish genome assembly has precluded confirmation or rejection of the predictions based on lamprey genomes (ref 37)”*

I am grateful if the authors could indicate the exact sentences in ref 37 that justify this citation, since I don't have time to confirm before the deadline for submitting this review.

Ref. 37 (now 8) mentions in their concluding remarks that including more agnathan genomes might improve analyses of vertebrate karyotype evolution. We have changed our sentence to be consistent with the reference, as the lack of a hagfish genome was not directly suggested as being an impediment to solve the problem, but rather to help elucidate these events. We have changed our sentence as follows:

Line 265-268: *“The inclusion of a hagfish genome in analyses pertaining to the reconstruction of the ancestral karyotype could potentially provide validation or refutation of predictions grounded in lamprey genomes (ref. 8)”*

12. Page 8, Lines 215-216: *“largely consistent with previous studies (refs 5,11,14), but with minor differences (Supplementary Table 34, Supplementary Information).”*

Supplementary Table 34 compares the reconstructions proposed in refs 5, 11 and 6 but not 14?

We apologise for this mistake and thank the reviewer for spotting it out. We have now changed ref. 14 for ref. 6.

13. Page 8, Line 218: *“confirming the 2R event (ref 39)”*

Similar observation was previously said to be consistent with accumulation of chromosome-scale duplications as discussed in ref 4. Besides, I am not sure if ref 39 states that the presence of four or three paralogous chromosomes in chicken or gar confirms the 2R event.

This is a mistake, thank you for spotting this. We have changed ref. 39 for refs. 3, 5 and 14 here. In ref. 5 for instance: *“First, the majority of CLGs (ten out of 17) are found in four descendent copies in bony vertebrate genomes (Fig. 3); the remainder are found in three copies. This pattern supports the 2R hypothesis if we allow for secondary chromosome loss via ancient aneuploidy of initially quadruply redundant copies.”*

We find 11 ACs in four descendent copies, one more than ref. 5; and the rest in 3 copies. This genome-wide pattern strongly supports whole genome duplication, and not segmental duplications. As per our response to Major Comments #1 and #2, we believe that the accumulated evidence coming from the most recent analyses using both lamprey and hagfish chromosome-scale genomes makes the segmental duplication scenario largely unlikely. We think that still considering this possibility despite the lack of direct evidence and parsimony adds confusion to

28

the debate, which is converging into an agreement by different researchers, all using different genomes and different methods: ref. 5, ref. 6, Marletaz et al., 2023 (bioRxiv) and ourselves here.

14. Page 8, Line 235: "are compatible with 1R"

Are compatible with shared 1R?

Thank you for spotting this. Indeed, we mean with shared 1R, and have changed the text accordingly.

15. Page 9, Line 237: "8.2% of cyclostome genes are compatible with a 2R history"

It would be helpful if the definition of "cyclostome genes" is described also in the main text (in addition to the description in the figure legend).

In this case we mean including both hagfish and lamprey. We have changed the text accordingly as follows:

Lines 463-464: "8.2% of cyclostome –including both lamprey and hagfish— gene trees are compatible with a 2R history".

Legend of Fig. 4a: "Phylogenetic support of gnathostome and cyclostome genes for 1R and 2R. Elephant shark, hagfish, lamprey or both cyclostome genes (both hagfish and lamprey genes included) were analysed in the context of spotted gar and chicken gene phylogenies by AC (using amphioxus genes) and orthologous sea cucumber genes (outgroup)."

16. Page 9, Lines 241-249: "When assessing how hagfish and lamprey chromosomes descended from the 17ACs ... suggesting that the lamprey (and, thus, cyclostomes) diverged after the 1R but before all eight post-1R/pre-2R fusions detected in gnathostomes."

If the authors decide to not use the hagfish genome for this analysis, what is the difference (or improvement) from similar analyses described in previous studies (refs 5 and 6)?

Thank you for the question. In our manuscript, the 1R-sharing hypothesis was supported by two independent lines of evidence, i.e., the post-1R fusion and the gene level phylogenetic tree analysis. Compared to previous studies, although we only used lamprey data for post-1R fusion analyses, we used new methodology. Moreover, for phylogenetic tree-based analysis, the hagfish and sea lamprey genome led to same conclusion.

Regarding the post-1R fusion based analysis, although we only used lamprey data as previous studies, our methodology has made progress on confirming post-1R fusion events with empirical evidence for the first time. As we responded to Major Comment 3, Sacerdot et al., 2018, Simakov et al., 2020 and Nakatani et al., 2021 have proposed what a post-1R fusion scenario would look like and what products it would generate in their original work (Panel A-C). All these previous works have assumed the phylogenetic relationships of different chromosomes (Panel A'-C'). However, empirical evidence was absent from the three studies. By contrast, we formally built chromosomal level phylogenies, and then used them to infer post-1R fusions (Panel D).

Analysis of the hagfish genome strengthened our second line of evidence. With both hagfish and sea lamprey genes, our phylogenetic tree-based analysis always supports the hypothesis that

29

cyclostomes shared the first round of WGD with gnathostomes (see main Fig. 4a in our manuscript) . This hypothesis could not be firmly supported based solely on the post-1R/pre-2R fusion related analysis presented in Sacerdot et al., 2018, Simakov et al., 2020 and Nakatani et al., 2021. In other words, absence of post-1R fusions only support that jawed and jawless vertebrates do not share the second round of WGD. However, it could mean that they even did not share the first round of WGD.

All in all, we combine two independent lines of evidence to support the 1R-shared hypothesis.

17. Page 9, Line 261: "at least three WGD events (refs 6, 18)"

Three WGD events were suggested in ref 18, but it is not clear how the authors counted three WGD events in ref 6.

We apologise for our mistake. We have now removed ref. 6 here and added a second part in the sentence to indicate the hexaploidization scenario. The sentence now reads as follows:

Lines 492-492: "*It has been suggested that the lamprey genome has been shaped by either three duplicative events (Mehta et al., 2013) or an hexaploidization (Nakatani et al., 2021)*".

18. Page 9, Lines 264-265: "we find that multiple chromosomes are direct descendants of each AC in both cyclostome groups."

The authors wrote that the hagfish chromosomes are highly rearranged. Which hagfish chromosomes (i.e. which Hi-C clusters) are direct descendants of each AC? What is the meaning of "direct" here?

Thank you for the comment. The "direct" word is misleading here, the sentence planned to mean that multiple chromosomes and/or large chromosomal sections after rearrangements in the hagfish have a clear corresponding relationship with each AC in both cyclostome groups. We have changed the text to read as follows:

Lines 501-502: "*we find that multiple chromosomes and large chromosomal sections are descendent copies of each AC in both cyclostome groups.*"

19. Page 10, Lines 286-287: "suggesting that, at least, a second WGD occurred in cyclostomes"
This observation seems to be consistent with accumulation of chromosome-scale duplications proposed in refs 4 and 17.

We have now changed "at least, a second WGD occurred" to "at least, a second WGD might have occurred". At this point in the main text, we have not reached the results that suggest a triplication in cyclostomes. In this regard, please, see our reply to Major Comment #1. In this sentence, we mentioned that the large majority of ACs have derived into three or more chromosomes in both cyclostomes. Given the scale, this is better explained by ploidy events (whole genome) followed by chromosome losses. Later, our OR and clustering of retention profiles show that this affects most chromosomes, so we do not think this is consistent with or supports chromosome-scale duplications, especially given (i) the short time between the last common ancestor of vertebrates and that of cyclostomes and (ii) the number of chromosome-scale duplications required in

30

comparison with the number of chromosome losses (which are a relatively common outcome of ploidy events).

We'd like to emphasise once more that after the most recent analyses by ref. 6, Marlétaz et al., 2023 with *E. atami*, and ourselves here, we think that the debate between chromosome-scale duplications and whole genome events should end.

20. Page 10, Lines 290-291: "suggesting that cyclostome genomes have been shaped by three shared WGD events."

It seems that other models proposed in refs 4, 6 and 17 are also possible. After thinking about this argument, I think it might be necessary to add a clause like "assuming that chromosomes can be duplicated only by WGD."

As explained above, our manuscript now supports that cyclostome genomes were shaped by an hexaploidization. The paragraph containing this sentence has been changed accordingly (see lines 558-562).

21. Page 11, Lines 309-310: "Overall, intra- and interspecific gene retention profile analyses indicate that two independent WGD events took place in the cyclostome stem-lineage;"

The result of the authors' analysis seems to be consistent with previously proposed models (refs 4,6 and 17). In order to avoid confusions, I suggest adding "under the assumption that cyclostome chromosomes were duplicated only by WGD."

As explained in our responses to above points, definitely not with refs. 4 and 17, but it is with ref. 6, a scenario we now favour. We have changed the sentence accordingly as it follows:

Lines 592-593: "*Overall, intra- and interspecific gene retention profile analyses indicate that a triplication event took place in the cyclostome stem-lineage*"

22. Page 13, Line 370: "all three of WGD events"
Which three? 2R, CR1 and CR2?

We have now changed the sentence accordingly to the triplication model:

Lines 686-687: "*Thus, while 2R and CR are of comparable antiquity*"

23. Page 13, Line 372: "2R occurred very early within the gnathostome stem-lineage"

Since refs 5 and 6 have suggested that 2R was allotetraploidy (i.e. hybridization of related species followed by genome doubling to recover the homologous chromosomes), it is not obvious that the inferred date corresponds to the genome duplication event.

We develop this point in the discussion section of the revised manuscript.

24. Page 13, Lines 379-380: "First, it reveals a robust and accurate history of WGD events in early vertebrates, demonstrating that cyclostomes diverged from gnathostomes after the 1R, but before 2R."

What is the meaning of robust and accurate here? Is it related to Page 54, Lines 1127-1128 in Supplementary Information?

We have removed this hyperbole though we note that this is the first attempt to estimate the timing of 1R and 2R without simply pegging the events to the fossil record.

25. Page 13, Lines 381-382: "This is consistent with early studies on the matter (refs 12, 14), ending debate over the timing of 2R"

Maybe ref 13 is missing? Also, refs 5 and 6 can be mentioned around here?

Thank you for your suggestion. We have slightly changed the sentence and added refs. 5, 6 and 13 here.

Lines 694-759: *"This is consistent with early (refs. 12–14) and recent studies on the matter that included the lamprey (refs. 5,6). We think that debate over the timing of 2R (ref. 8) is now closed."*

26. Page 13, Lines 382-283: "evidencing two additional WGD events, CR1 and CR2, in stem-cyclostomes"

I think that the previously proposed models (refs 4, 6 and 17) should also be mentioned in Discussion, if they were not rejected by the hagfish genome analysis.

We now argue in favour of the model proposed by ref. 6, confirming its occurrence before the divergence of lamprey and hagfish lineages. We have now modified the manuscript accordingly, including Discussion, to reflect this. However, as we explained above, the genome-wide nature of the pattern observed by us and other recent studies confidently rejects the chromosome-scale duplication hypothesis, so we are not including this possibility in Discussion.

27. Page 14, Lines 397-399: "Another possibility is that the 2R event is different in nature from the 1R and CR events...., while 1R was likely an autopolyploidy event, 2R was an allopolyploidy" It might be necessary to discuss if the CR events were auto-polyploidy before the argument that the 2R event is different from 1R and CR.

Thanks for pointing this out. As above, we now discuss this point at this stage of discussion.

28. Page 20, Lines 574-575:

Please add the common name of *X. tropicalis*, *D.* should be *Danio*, *mili* should be *mili*, and please delete the period after *Petromyzon*.

Thanks for spotting these out. Added and corrected.

29. Page 23, Line 655: "x10⁻⁵"

Is it correct?

Thank you for pointing this out. Corrected it as 10⁻⁵.

30. Page 27, Lines 785-787: "All stem gnathostomes were adjudged to have undergone ... molecular clock analysis"

32

The inferred timing of 2R is the date of divergence between the two progenitor species, which may not be close to the timing of allotetraploidy. Can we still safely assume that all stem gnathostomes post-dated 2R?

Thanks for raising this point. No, not as safely, but the ostracoderms are so much younger than this inferred age that it is extremely unlikely that they are pre-2R. Nevertheless, this uncertainty remains and adds to our main conclusion, that the hypothesis that WGD confers evolvability is unfounded, in this case at least.

31. Page 39, Figure 4a

The figure explains the case for 4-way paralogous trees, but not for 3-way paralogous trees. If chicken chromosome 27 and gar LG15 are deleted from the figure, how do we classify the purple and yellow arrows in Fig 4a? Purple and yellow arrows will be 1R compatible and 2R uninformative, and in this case the purple and yellow arrows are not counted in "Phylogenetic Support for shared 2R." Is it correct?

That is correct, when chicken chromosome 27 and gar LG15 are deleted, purple and yellow arrows in the original figure are not used to infer 2R related support. We have also extended section 4.8 of Supplementary Information to make this clearer.

Suppl. Information, Lines 1708-1714: *"The possible position of tested genes from each model follows the example presented in Fig. 4a of the main text. For each tree, we classified it into shared 1R-compatible or -incompatible first. If it was classified as shared 1R-compatible, we further classified it into shared 2R-compatible or -incompatible. Note that unlike the cases where there is a one-to-four correspondence in Fig. 4a, sometimes one AC gene corresponds to three chicken or gar genes (Extended Data Figure 5a), in which case we could not distinguish between shared 2R-compatible or incompatible. These cases would be only used to infer shared 1R compatibility, but not 2R."*

Supplementary Information

32. Page 33, Line 665: " $e \leq 10^{-6}$ "

Typo?

Corrected: " $e\text{-value} \leq 10^{-6}$ "

33. Page 53, Line 1106: "two rounds of WGDs"

Two rounds of WGD?

Corrected here and also in several other instances in Suppl. Information.

34. Page 54, Lines 1143-1144: "manually split these homologous gene into two parts given the 20%, 30%, 40%, 50%, 60%, 70% and 80% quantile rank position"

The authors wrote their method calculates an upper bound of the overlapping ratio (Page 55, Line 1150). If we really need an upper bound or a conservative estimate, it might be easier to understand if fission simulations are performed at all gene boundaries.

33

Thanks for raising this question. We have now performed fission simulations at all possible gene boundaries. Upper limit of each random split has been re-estimated. The results are described below. The new results, after splitting on all possible boundaries, show a similar overlapping ratio distribution (see below, panel c' and d'). Only 3/94 (3.2%) cases of random split on chicken and gar chromosomes have an OR exceeding 0.15 in these new results. It means a specificity of 96.8%. So, 0.15 still serves as a good threshold to define paralogous chromosome pairs. In summary, splitting on all possible gene boundaries does not alter other parts of our analysis.

We have replaced the original Fig. 4c and d bottom panels with these new results:

Old Fig. 4c and d:

New Fig. 4c and d:

Raw data of random splitting in Supplementary Table 40 has also been updated. The methodology related to this part of analysis has also been updated accordingly in the corresponding section of Supplementary Information:

Supp. Information, Line 1800-1825:

“4.10.2) Threshold of overlapping ratio to discriminate homologous chromosomes. We generated the random expectation of ORs by artificially splitting a chromosome into two parts and calculating the ratios. In other words, for a vertebrate chromosome corresponding to one AC, we ordered the homologous genes by their actual positions on the chromosome and manually split these homologous genes into two parts on all possible gene boundaries. Then, if each part harbours at least 20 retained genes, we calculated their ORs. Therefore, for each chromosome and the corresponding AC, there would be multiple ORs from random splits, the maximum value of which was recorded as the final ratio for this chromosome. Two rationales were invoked in this simulation: 1) we could not randomly sample two chromosomes since they may be duplicates derived from one AC; 2) we chose the maximum rather than median value in order to be conservative, i.e., to generate an upper-bound estimation of overlapping ratios of two random chromosomal segments. Finally, given each chromosome and the corresponding AC, there are a total of 49, 45, 48 and 47 simulated split results in chicken, gar, hagfish and sea lamprey genome, respectively (Supplementary Table 43).

By comparing the distribution of overlapping ratios for artificially split chromosome pairs and bona fide paralogous chromosome pairs in gnathostomes, we reasoned that $OR=0.15$ should serve as a good threshold. Only 3 of 94 (3.2%) random splits from either chicken or gar genome have an overlapping ratio > 0.15 (Supplementary Table 43). So, the specificity of our methodology to identify paralogous chromosome pairs has a lower bound of 96.8%.”

35. Page 59, Lines 1235-1245

Is it possible to calculate the theoretical distribution of paralogous chromosome fusions?

Thanks for your suggestion. Monitoring it mathematically should be possible, but extremely challenging.

36. Page 61, Line 1278: "four gnathostome species"

What is the reason for choosing these four species?

There is publicly available data for these four species. Three out of these four were the ones used in the original publication (ref. 4) finding an increase in regulatory complexity in gnathostomes. Adding chicken here served us to corroborate this in gnathostomes.

37. Page 65, Line 1359: "Branchiostoma_lanceolatum"

Typo? Branchiostoma lanceolatum

Thank you for pointing this out, we have now corrected it.

38. Page 91, Line 1966 Supplementary Files

It would be helpful if the URL for downloading these supplementary files is written between Line 1966 and Line 1967.

Added: https://figshare.com/projects/Hagfish_Genome_Project/163186.

39. Page 108, Supplementary Figure 9:

HoxC6, C3, C1, D5 and D2 are missing in the Crown Gnathostome Ancestor?

Thank you very much for spotting this. We have now corrected this Figure.

40. Page 117, Line 2256: "circus"

Circos?

Corrected.

41. Supplementary Table 31, "Note, chicken chrZ is mainly orthologous to gar LG4.2 and chicken chr4.2 is mainly othologous to gar LG2.2."

Is it correct? The table looks different. Also, please check the spelling of orthologous.

Thanks for pointing it out. We have now corrected these.

Reviewer #2 (Remarks to the Author):

Reviewer: Aoife McLysaght (I'm happy to reveal my name to the authors ... I think they'll guess anyway :-))

Dear Aoife, we're really thankful for your being open about your identity and for helping us to improve our manuscript. Your criticism has been taken humbly and we have tried now to address all your concerns. In particular, we'd like to apologise for our careless use of citations, which now have been carefully corrected.

This manuscript describes the sequencing and genome analysis of the inshore hagfish, *E. burgeri*.

This is the first Hagfish genome to be published, both in terms of the data availability (the genome data has generously been available through [ensembl.org](https://www.ensembl.org) for several years) and in terms of the manuscript release (this appeared on BioRxiv about a week before another hagfish genome paper was published there, though of a different hagfish species). Thus this is an intrinsically important paper as the hagfish lies in an important position in the vertebrate tree, forming a clade with the lampreys. Additionally, as pointed out in this manuscript the GC content of hagfish is not as extreme as lampreys, which should make this genome very useful for comparative genomics of vertebrates. I am confident that this paper will attract a large amount of attention and will be highly cited.

The summary of my view is that this paper should definitely be published in your journal, but first the manuscript requires some important improvements.

We deeply appreciate your positive comments. We have now updated our interpretation and discussion of our results according to your suggestions (see point by point answers).

In some aspects this manuscript (as distinct from the analyses) appears to be a bit rushed. One consequence of this is that some of the existing literature is not correctly cited or is cited inconsistently (in some places of the manuscript, but not others even though some papers are cited), or not cited at all. The authors need to pay close attention to acknowledging the work of others. This is just one symptom of a not terribly well written paper - the authors can definitely fix this with some care and attention. There are some parts where the manuscript doesn't flow well, where the figure legends are a bit lacking, and where the explanations too sparse. I have pointed out a good number of these below.

The reviewer is correct in that we rushed our writing, aiming at a timely submission. We apologise for our sloppy writing and the incorrect use of citations this entailed. We hope we have now addressed all issues and concerns so kindly raised by the reviewer.

Nakatani et al (ref 6) had several findings that are mis-cited here: they showed that only the 1R WGD was shared with cyclostomes, and this finding was definitive due to the non-sharing of several chromosomal fusions that occurred after 1R but before 2R; they also showed that there was additional polyploidy ancestral to the cyclostomes and inferred that this was a hexaploidy/triplication (notably the data presented by Yu et al are consistent with hexaploidy, more on that later) but the manuscript cites this paper in reference to 'additional WGD events' in

lamprey, but this is triplication, not duplication. Nakatani was the first to propose hexaploidy in cyclostomes; Nakatani et al were also the first to infer that the proto-vertebrate genome contained 18 chromosomes, an idea which has since been accepted by Simakov and co-authors, but does not originate with Simakov, as implied by the citation here. I feel honour-bound to point out that I am corresponding author on Nakatani et al, and I would never abuse the role of reviewer to request citations unnecessarily, but in this case the paper is very relevant and it is important in my view to cite it accurately.

We definitely appreciate the pioneering and inspiring works of peers in the field of early vertebrate evolution and deeply apologise for the mis-citations of their work. Nakatani et al. 2021 has quickly become a seminal work in early vertebrate genome evolution, and inspiring to us.

We also acknowledge now that our data is more consistent with the hexaploidy scenario and modified our interpretation of the results and the discussion accordingly, as well as properly cited Nakatani et al. 2021 (ref. 6). We'd like to point out, however, that although it is true that Nakatani proposed this event ancestral to cyclostomes, this needed to be confirmed with the inclusion of a hagfish species. Our manuscript now corroborates this to be the case, so we believe that our hagfish genome indeed reinforces the hexaploidy scenario proposed by Nakatani and provides irrefutable evidence that it was ancestral to cyclostomes.

We have also added a citation to Nakatani's article when mentioning and discussing the possibility of a 18-chromosome proto-vertebrate karyotype, and all misrepresentations and mis-citations of its results are now corrected in the revised manuscript. We have also added this scenario in our Figure 1, panel c.

Gene complement analysis (line 153 onwards) - this is interesting but it would benefit from a comparison with other results, for example those from the whale shark paper
Tan, M. et al. The whale shark genome reveals patterns of vertebrate gene family evolution. *Elife* 10, e65394 (2021).

Thank you for the suggestion. Tan et al. uses a different taxonomic sampling for their analysis, particularly they use more chondrichthyan but very few invertebrate genomes (only two). Even so, they generally find similar results, although with different numbers. Particularly, they find that the larger peaks of gene family gains occur also in the lineage towards the MRCA of Olfactores as well as Vertebrata and Gnathostomes. This can be due to the use of few invertebrate genomes (only *Ciona robusta* and *Branchiostoma floridae*).

We have now added a sentence to acknowledge this previous study (underlined):

Lines 202-204: "These are notably larger than those observed in other major evolutionary episodes in metazoan evolution (Paps and Holland, 2018; Guijarro-Clarke et al., 2020), but generally similar to a recent study using more chondrichthyan but only two invertebrate genomes (Tan et al., 2021)"

Inference of the proto-vertebrate chromosomes (PVCs)/ancestral chromosomes (ACs) and analysis of 1R and 2R:

The details of the PVCs are opaque and the text is muddled. The text talks about 17 inferred proto-gnathostome chromosomes, whereas it is clear from supp figure 5 that they are actually referring to proto-vertebrate chromosomes, (and instead there are 24/26 proto-gnathostome chromosomes in their reconstruction ... trying to count in the figure). Additionally, they vary between 16 (e.g. Fig 4) and 17 Acs and there is only a small note about this in the supplementary information (line 955) that I could find. The authors clearly went to a great deal of effort to reconstruct these ACs/PVCs, but it isn't clear to me what they gain over using the available published reconstructions or how their method might be superior. I note that the supp info compares their reconstruction to those of Simakov, Nakatani and Sacerdot (supp table 34) but these are not all cited on line 215 (main text). The comparison is very sparse - just naming equivalent chromosomes in the various reconstructions, without any detailed comparisons, so it is hard to know what is added here with another new construction, and one that has a different number of chromosomes to the others.

We apologise for the lack of clarity. This was a mistake and we inferred 17 ancestral chromosomes in vertebrates, not in the LCA of gnathostomes. We have corrected this mistake:

Lines 277-279: *"With this, we inferred a proto-vertebrate karyotype of 17 ancestral chromosomes (ACs) (Supplementary Information; Supplementary Fig. 15; Supplementary Table 32)."*

We agree with the reviewer that it is not clear why we are using 16 ACs in some analyses instead of 17. Originally, we were able to reconstruct the gene content of only AC1-16; however, we have now repeated our analysis and reliably mapped 20 amphioxus genes to AC17. We have repeated analyses not demanding big sample sizes, including the karyotype phylogeny depicted in Extended Data Fig. 5a

We have now reorganised and rewritten the sections corresponding to our ancestral karyotype inference, both in the main text and in Supplementary Information, also providing better details of our comparison to previous models (which are now done by direct comparisons rather than using the correspondence between proto-chromosomes provided in previous studies) [see sections 4.5] *Inference of vertebrate ancestral chromosomes* in Supplementary Information].

Although we agree that recent karyotype reconstructions by Simakov and Nakatani provide excellent models of ancestral conditions, our analysis allowed us to provide an explicit phylogenetic framework of such reconstructions, allowing us to inspect the exact duplicative trajectory of extant gnathostome chromosomes (Extended Data Fig. 5a). Our study also establishes that although the ancestral chordate karyotype most certainly consisted of 18 chromosomes, whether the ancestral vertebrate karyotype had 17 or 18 chromosomes remains an open question given the uncertainty of the AC3 versus Pvc8-Pvc9 (CLGQ-CLGI) conundrum, now discussed in the main text (see section "Ancestral vertebrate karyotype"), in which we favour a 17-chromosome scenario, but do not reject a model with 18 chromosomes:

Lines 423-424: “Therefore, although we infer the number of ancestral vertebrate chromosomes to be likely 17, a scenario with 18 chromosomes (Nakatani et al., 2021; Simakov et al., 2022) cannot be ruled out.”

Given the proto-gnathosome/proto-vertebrate confusion I then find the rest of the paragraph (line 217 onwards) hard to follow. I am struggling to find the details of the pre-1R fusion in the supplementary material and I am thus unsure of the strength of evidence to infer this.

We have now rewritten and reorganised the part of the main text regarding our ancestral karyotype reconstruction including more details about the pre-1R fusion that we think gave rise to AC3 (see Lines 406-424 of the main text in the resubmitted manuscript). In the supplementary material this is within sections 4.6.3 and 4.6.4.

With regard to the post-1R pre-2R fusions, these have been previously described, but this is only properly acknowledged in the supplement. This paragraph in the main text (line 222 and around) does not acknowledge this. This must be corrected. Similarly, the following paragraphs that use the non-sharing of the fusions to infer the non-sharing of 2R with cyclostomes previously appeared in ref 6 (Nakatani) at the very least.

We apologise for not properly citing earlier studies regarding the post-1R pre-2R fusion analysis. We have now added appropriate references in the main text when talking about identification of post-1R fusions and mentioned this kind of evidence was used before, including Sacerdot et al., 2018, Simakov et al., 2020 and Nakatani et al., 2021. This is now discussed in the second paragraph of section “Gnathostomes and cyclostomes share 1R but not 2R” of the main section (Lines 463-495).

In the section on cyclostome-specific WGD (line 260 onwards) the data presented are consistent with genome TRIplication (hexaploidy) as first proposed by Nakatani et al, but these authors do not discuss that. In Fig 4f we see that the maximal multiplicity of proto-vertebrate chromosomes to cyclostome chromosomes is 6, which is consistent with hexaploidy. Could they please analyse their data in the light of that published hypothesis? The Hox cluster data are also consistent with hexaploidy. Is there actually any reasonable evidence to support CR1+2 over hexaploidy? If the authors have gene tree topology evidence or other more detailed synteny evidence then they should present it. As it stands, the data provided look to be more supportive of hexaploidy.

Although our clustering analysis of retention profiles of hagfish and lamprey descendent copies of the ACs led us to first favour a scenario with two duplicative events in the cyclostome lineage, we realised by the end of our writing that this was not equivalent to a phylogenetic signal that supported such scenario. We agree with the reviewer that our results are better explained with an hexaploid cyclostome ancestor, and have now modified the manuscript thoroughly to reflect our new vision, including proper citations to Nakatani et al., 2021. This is particularly reflected in the new title of the corresponding section, now entitled “Cyclostome-specific whole genome triplication” (Line 497).

Dating the polyploidy: the approach is not invalid, but it is not conservative. It is not clear if the approach of using the posterior results from the species tree based dating is entirely appropriate.

I am not convinced it is wrong to do this, but I do think it is the opposite of being conservative about the ability to date the WGD events because these posteriors are likely more constrained in time than the raw, purely fossil-based priors. I would be interested to see 1. Analysis of this type using the priors from the species tree analysis (how does this affect the date range for WGDs?), and 2. Runs under the prior (i.e. the posterior of the original species tree analysis) presented to show that using the posteriors of the main analyses are sufficiently diffuse so as not to inform the WGD dates. In addition to this – you should probably consider that if CR2 is not well supported then the CR1 and CR2 dating is inappropriate.

We appreciate, in retrospect, that we did not provide sufficient justification for our approach. Our revised text and methods remedy this shortcoming. In summary, our original submission followed a sequential Bayesian approach in which it is statistically consistent to use the posteriors of the species dating analysis as priors in the gene dating analysis so long as the data are not duplicated between the analyses [Alvarez-Carretero, S., et al. (2022). *Nature* 602: 263–267]. To assay the impact of calibrating our gene trees in this way, we reran the dating analysis of the gene tree alignment using only fossil calibrations. Comparison of the ages show that they are generally compatible except for nodes that were constrained loosely in the original fossil calibrations. We run analyses without sequence data as a matter of course, to determine whether the effective priors are biologically reasonable, however, the specified priors in the gene dating analysis are simply uniform distributions based on the 95% HPD of the species tree posterior node age estimates. We have added these results to the supplementary information and we refer to them in the revised text.

In the Discussion (line 378 onwards) some of the claims are overstated in terms of novelty: It was previously demonstrated that cyclostomes diverged after 1R and before 2R; additional polyploidy in cyclostomes has previously been reported.

We have now toned down our Discussion in terms of novelty, but highlighted that the hagfish genome has served to corroborate what previous studies have suggested using the lamprey.

With regard to the uncoupling of polyploidy and morphological evolution, note also the studies on paddlefish and sturgeon - sturgeon is one of those with hardly any morphological change yet has experienced polyploidy.

Many thanks for highlighting this point; we have developed the discussion to include this point.

Minor comments:

line 59: add 'often assumed to be' before 'causally'

Added.

line 61: it would be a quite eccentric view these days to doubt the veracity of 2R

We absolutely agree with the reviewer and have removed "the veracity" from the sentence.

line 255: Nakatani was the first to infer 18 proto-vertebrate chromosomes.

First, we apologise for not having cited Nakatani in this sentence before. We have now rewritten and reorganised this section thoroughly, and added the corresponding citations when an inference of 18 chromosomes is mentioned:

Lines 406-408: "*Our inference of an ancestral karyotype with 17 ACs matches a previous study (Sacerdot et al., 2018), has minor differences with other 17-chromosome inferences (Putnam et al., 2008; Simakov et al., 2020), and depicts one less chromosome than more recent studies (Nakatani et al., 2021; Simakov et al., 2022)*"

Lines 423-424: "*Therefore, although we infer the number of ancestral vertebrate chromosomes to be likely 17, a scenario with 18 chromosomes (Nakatani et al., 2021; Simakov et al., 2022) cannot be ruled out.*"

line 324: appears to be an incomplete sentence or bad syntax. Higher than what?

We have corrected this:

Lines 606-608: "*We found a significantly higher number of ACRs per gene than in the cephalochordate amphioxus, similar to what has been observed in gnathostomes (Marlétaz et al., 2018) (Fig. 5a)*"

note on nomenclature: WGD refers to 'duplication' but not every polyploidy is tetraploidy. (duplication) so sometimes WGD is not an appropriate acronym.

The use of WGD generally includes other polyploids in the bibliography, at least in plants. For an example, see Alix et al. 2017, *Annals of Botany* 120: 183–194 doi:10.1093/aob/mcx079 ["Polyploidy events [...] refer to either single or multiple rounds of WGD (i.e. duplication or triplication)"]. We now similarly define the acronym in the abstract and in the first sentence of the main text as "Polyploidy or whole-genome duplication (WGD)". Nonetheless, we refer specifically

to the triplication event in cyclostomes and the hexaploid nature of its ancestor when necessary for the sake of clarity.

line 660 : cite the study where the Hox genes were sourced

Done.

Fig 4a is confusing. I wonder if there are missing purple and orange arrows? My interpretation is that there should be another purple arrow on the branch directly below the first one, and the same for all the branches below the orange arrow - shouldn't they all have orange arrows? It is also unclear what is meant in the legend when it is written "possible positions were test genes can group" - what is 'grouping' in this context?

We agree that the panel was confusing, and we apologise for it. We have now modified the panel to be easier to understand and to include all the possibilities as the reviewer mentions. Also, we have modified the legend to make it easier to understand. By 'grouping' we meant where the test genes (those from elephant shark as control, and lamprey and hagfish genes) would branch in the tree. We have modified the legend accordingly.

Legend of Fig. 4a: "*Phylogenetic support of gnathostome and cyclostome genes for 1R and 2R. Elephant shark, hagfish, lamprey or both cyclostomes genes (both hagfish and lamprey genes included) were analysed as test genes in the context of spotted gar and chicken gene phylogenies by each AC (using amphioxus genes) and orthologous sea cucumber genes (outgroup).*"

In fig 4 there are only 16 ACs presented but this is very confusing because there is no mention of there being only 16 anywhere else in the main text (including the methods of the main text) - this only appears, not well explained (that I can find) in the supplement.

We agree with the reviewer that the reasons to exclude AC17 from this analysis were not clear. In the previous version of our manuscript, we had not been able to anchor any gene to AC17 in a reliable manner. After repeating the analysis during the revision of the manuscript, we have now reconstructed its gene content, although we have been able to map only 20 genes to it. We now better explain our reconstruction of the gene content of our ACs, fundamental for this analysis, in the main text (Lines 280-402), in the methods (Lines 1257-1271) and in supplementary information (Section 4.5.2).

We also better explain why AC17 was excluded from the OR analysis:

Main text, lines 283-402: "*In total, we mapped 5,065 Belcher's lancelet genes to AC1-17 (ranging from 115 to 534 genes in AC1-AC16; AC17 consisting of only 20 genes and being thus excluded from several subsequent analyses; Supplementary Table 34).*"

Main text, lines 508-509: "*gene-poor AC17 was excluded from this analysis, which required at least 20 genes retained in each descendent chromosome*"

Legend Fig. 4: "*Note that AC17 was excluded from the analyses depicted in c-f because of the low number of genes we recovered (20 genes)*"

43

Supplementary Information, section 4.9.2, Lines 1787-1789: *"To enhance robustness, we required at least 20 retained genes on descendent chromosomes from a given AC before calculation of the overlapping ratio; since it harboured only a total of 20 genes, AC17 was excluded from the analysis"*

Fig5e - explain 'red', 'subf 'spec' in the legend

We have now added what these abbreviations stand for in the legend. They are potential redundancy, potential subfunctionalization and potential specialization, respectively.Reviewer #3 (Remarks to the Author):

This article presents the first genome analysis of a hagfish, representing one of the two major lineages of extant agnathan, cyclostome vertebrates and one of last major groups of vertebrates to be genomically explored. By its study subject itself, this work will be of highest interest and impact to the vertebrate genome evolution community.

We really appreciate the positive consideration of our study.

Major parts of the present study use the hagfish genome assembly to investigate and clarify major patterns of genome evolution in early vertebrate lineages, especially with regard to the timing and patterns of whole genome duplications (WGD) within cyclostomes as well as in the jawed vertebrates, the gnathostomes. By the addition of the hagfish genome, the current study confirms cyclostome monophyly and that only the first round of WGD (1R) was shared among all extant vertebrates. According to the present study, the cyclostomes then underwent two additional rounds of WGD (CR1, CR2) before the divergence of hagfish and lampreys, explaining for example their high number of Hox clusters. The authors further find that the 2R event was specific to the gnathostome lineage in agreement with previous studies (Simakov et al. 2020; Nakatani et al. 2021). The study then goes on to investigate patterns of post-WGD evolution in terms of gene gain and retention, gene expression, and regulatory evolution. Most interestingly, phylogenetic analyses integrating data from the fossil record provide evidence that this 2R WGD predates the massive morphological diversification across the lineage leading to extant gnathostomes, suggesting that 2R may have spurred this extensive exploration of morphological space.

Overall, I think that the study will be of highest interest to the broad readership of Nature Ecology and Evolution. However, the manuscript needs to be rewritten for a more general audience, especially for more clarity. In several places, I needed to go back and forth between the main text and the supplements to understand the main conclusions to be gained from the analyses. The authors should strive to let the main text stand for itself.

We really appreciate the positive comment of the reviewer about our study, while at the same time we apologise for the lack of clarity of our main text. We think that after addressing all reviewer's concerns, our manuscript has improved considerably. Our responses to each of the concerns raised by the reviewer can be found below.

General points:

1. Genome assembly:

Multiple versions of the hagfish genome have been generated. Publicly available has been an intermediate genome assembly version in Ensembl for quite a while now (version 3.2). The final assembly version used for the analyses in this manuscript (v4.0) does not seem to be publicly available yet and accession numbers for this final version are not available as far as I can tell.

Thank you for pointing it out. The *Eptatretus burgeri* (inshore hagfish) v4.0 genome is now available in NCBI now under accession number GCA_900186335.3 and we have added this at the end of the manuscript under the data availability statement.

2. Hagfish and cyclostome biology:

Comparatively small parts of the article are devoted to the genomic exploration of hagfish biology itself which may be presented in another study (e.g., genes involved in the formation of the iconic “slime eel” slime, etc.). At some instances, the analyses show that the gene complement of cyclostomes is quite reduced compared to that of other vertebrate lineages (e.g., l. 172 “suggesting that a strong asymmetric reduction of gene complements accompanied the early evolution of the group”). I think it would have been interesting to discuss these results in more detail and connect them to the general understanding that cyclostomes and especially hagfishes represent rather morphologically derived and reduced lineages.

As the reviewer rightly guesses, we will carefully discuss some of these cases (such as the slime) in detail in future studies. The general understanding that cyclostomes are rather derived, reduced or simplified is inherently faulty, in the sense that species, organisms or lineages, considered as a whole, are not per se ancestral or derived, but character states are. So we prefer not to participate in extending this erroneous view. Characters should be tested case by case (e.g. in our present paper: opsins and related genes, circadian rhythm genes and immune system genes), and we think that our genome provides the ideal framework and resources for related future analyses.

3. Genomic comparisons:

For the comparative genomic analyses, a lot of other vertebrate and non-vertebrate key taxa are included. I appreciate the inclusion of Supplementary Table 1 to list all the resources. However, information should be added whether the included versions of these species are chromosome-level genome assemblies or not.

Thank you for your suggestion. We have added a column in Supplementary Table 1 now indicating the assembly level of each genome assembly. We have also added a new row for *Petromyzon marinus*, as we had made use of the new chromosome-level assembly (see answer below).

This is particularly relevant for the synteny-based analyses. For example, the elephant shark genome assembly used does not seem to be the more recent, chromosome-level version (Nakatani et al. 2021). An alternative assembly for Chondrichthyes could have been the little skate genome. Likewise, the sea lamprey genome assembly also seems to have been updated since the versions used for the analyses here, and the Arctic lamprey (Nakatani et al. 2021) could have been included as an additional cyclostome genome.

To be clear, I do not necessarily ask the authors to redo or expand their analyses, but some more clarity and discussion on the limitations of the assemblies used in the present study would be helpful.

We agree with the reviewer that using chromosome-scale assemblies for as many species as possible would be the most optimal strategy. We performed the synteny analysis before the

chromosome-scale assemblies of the elephant shark and the Arctic lamprey were released. In the case of the elephant shark genome, we used it only to recover the ancestral karyotypes of chicken and spotted gar, which, after our modifications, showed almost perfect one-to-one chromosome level synteny (Supplementary Table 31). As reported previously [Ref. 40: Ingo Braasch et al. (2016); Ref. 41: Venkatesh et al. (2014)], elephant shark, spotted gar and chicken genomes all share extensive chromosomal synteny. Moreover, the recent analysis of the little skate genome [Marlétaz et al. (2023) 616(7957):495-503. doi: 10.1038/s41586-023-05868-1], mentioned by the reviewer, also shows a large macrosynteny conservation with spotted gar and chicken (see image below).

Figure 2b from (Ferdinand Marlétaz et al., 2023). The syntenic orthology relationship between skate, gar and chicken, relying on genes with a significant CLG assignment in regard to amphioxus. Skate chromosomes are coloured by segmental identity and links are coloured by CLG.

All in all, we believe that repeating our macrosynteny analysis with either the little skate genome or the updated, chromosome-level assembly of the elephant shark, although preferable, would not change our results in any significant way.

About the updated chromosome-scale germline genome of the sea lamprey *Petromyzon marinus*, we were aware of it while carrying out our study, but its use was under embargo, so to carry out our analyses before its publication, we modified the 2018 version of the *P. marinus* genome making use of the Pacific lamprey meiotic map published in the same article (Smith et al., 2018) for most analyses. With that, we corrected several misassemblies. This was indicated in the main text in lines 280-281 of our original submission. However, the chromosome-level assembly of *P. marinus* was published just before our submission [Timoshevskaya et al. (2023) *Cell Rep.* 42(3):112263. doi: 10.1016/j.celrep.2023.112263]. The 2023 genome assembly for the sea lamprey is marginally better than the assembly from Smith et al., 2018 *Nature Genetics* (N50 size 13.0 Mb v.s. 12.3 Mb). Nonetheless, we quickly used it, to confirm misassemblies of the 2018 version of the genome before the OR analysis, and also to confirm the lack of signals of any of

the eight post-1R fusions. We agree with the reviewer that this was not clear, so we have modified this section of the main text, adding the following lines (underlined), and also included a new Supplementary Figure 24:

Lines 544-547: "We then applied the OR metric to the sea lamprey (after correcting misassemblies assisted by a meiotic map of the Pacific lamprey *Entosphenus tridentatus* (Smith et al., 2018) and confirmed by the recent chromosome-level genome assembly (Timoshevskaya et al., 2023)); see Supplementary Information: Supplementary Table 50: Supplementary Fig. 24"

And we have also rewritten the corresponding sections of the Supplementary Information: (i) we improved what is now section 4.8 (old 4.11), entitled *Further super-scaffolding of the 2018-version of the sea lamprey germline genome assembly*; (ii) we added one new row in Supplementary Table 1 for the newest version of the germline genome of *P. marinus*, to indicate that we used it for our macrosynteny analysis; and (iii) we also extended section 4.9 (*Investigating whether lamprey lineage shared gnathostome pre-1R and post-1R fusions*) to better explain how we conclude that there are no reliable signals of any of the post-1R fusions and that we used the latest *P. marinus* genome assembly to corroborate our results (including a new Supplementary Figure 27).

Last, regarding the chromosome-level assembly of the Arctic lamprey, unfortunately the protein coding gene annotation was not publicly provided and it's also not available from NCBI (https://www.ncbi.nlm.nih.gov/assembly/GCA_018977245.1), precluding further analyses. In any case, we can safely assume that our conclusions would not change, due to conserved synteny between Sea lamprey and Arctic lamprey (Smith et al., 2018).

4. Vertebrate WGD nomenclature:

The cyclostome-specific genome duplications here are termed CR1 and CR2.

1R is the WGD shared among all crown vertebrates, 2R is the WGD in the lineage leading to gnathostomes. Within gnathostomes, later WGDs have been called for example 3R for the teleost fish WGD, in an enumerating manner starting from the vertebrate ancestor.

CR1 and CR2 hence are the 2nd and 3rd WGD in the cyclostome lineage and to me, it would make more sense to call them CR2 and CR3, respectively. Likewise, it would be good to qualify 1R as, for example, V1R (vertebrate 1R) and G2R (gnathostome 2R). I think this study provides an excellent opportunity to rework the nomenclature of vertebrate WGDs.

We have now changed our main conclusion regarding polyploidy in the cyclostome lineage and believe that our data is more supportive of a triplication event. We now call this event CR. We also prefer to keep using the classic nomenclature of 1R and 2R for the stem-vertebrate and stem-gnathostome WGD events, since that is the original nomenclature and the original dating of these events.

5. Hox cluster evolution:

The current study clarifies the orthology of the six hox clusters among cyclostomes. However, I could not find a model of how these six clusters were generated through the three cyclostomes WGDs. Following the 1R-CR1-CR2 model, we could expect to see up to 8 hox clusters in cyclostomes, so 2 clusters were lost somewhere along the way to the cyclostome ancestor. Using

48

your map of cyclostome chromosomal relationships to the ancestral chromosomes (ACs), it should be possible to build a model of hox cluster evolution through the three rounds of WGD leading to cyclostomes.

As mentioned above, we are now supporting a triplication event rather than two WGD events in the cyclostome lineage.

6. Post 2R morphological evolution:

One of the most exciting parts of the current study was the integration of fossil data which led the authors to conclude that basically all non-cyclostome vertebrate lineages (extinct and extant) are derived from the 2R WGD (Fig. 5h). This has important implications for the significance of the 2R event for the morphological diversification of vertebrates. However, I found this part particularly difficult to follow and to understand how its major finding was derived. Please discuss your approach in more detail and make it more clear how you derived this major conclusion. As written, the relevant section in the main text (starting l. 350) does not clearly convey how you came to this far-reaching conclusion. Even after reading the supplementary text multiple times, I am still not fully clear how it was concluded that basically all non-cyclostome vertebrates are 2R-derived. Please rework this part so that it becomes digestible for the non-expert.

Many thanks for raising this point. We have developed the main text to better explain and justify our approach.

7. Nature of WGD events:

a) Simakov et al. 2020 concluded that 1R was an auto- and 2R an allotetraploidization event. Nakatani et al. 2021 later came to similar conclusions. The current study, however, does not make an inference for establishing the nature of the 1R and 2R events. I thus wonder if the current analyses are at least consistent with and/or supporting Simakov's model? I find this particularly important since the allotetraploidization of 2R from previous studies is cited here as possible explanation for the impact of 2R on morphological evolution. I therefore would like to see a more in-depth analysis and discussion whether the authors' chromosome evolution data here also support an allotetraploidization for 2R (and an 1R autotetraploidization).

Thanks for raising this important question. We have now performed quantitative analysis and concluded, as previous studies, that 1R was an autotetraploidization and 2R an allotetraploidization.

We have added this in the main text:

Lines 455-461: *"Furthermore, we found a significant gene retention asymmetry after 2R, with a median of 1:2.28 genes per ohnologous (duplicates that originate through WGD, after Susumu Ohno(Ohno, 1970)) chromosome pairs, but not after 1R (median 1.16; $P=3.4 \times 10^{-7}$, Wilcoxon rank sum test; Extended Data Figure 5b). This pattern is consistent with previous studies suggesting that 1R was an autotetraploidization event, and 2R an allotetraploidization (Nakatani et al., 2021; Simakov et al., 2020) (but see Parey et al. (2022) on asymmetric gene retention after teleost 3R autotetraploidy)."*

We have added the asymmetry ratios in Extended Data Fig. 5b, and also added a section to explain these results in Supplementary Information (section 4.6.2) *Nature of 1R and 2R polyploidy events*).

b) Nakatani et al. 2021 conclude that following 1R, the extant cyclostome genome was generated by hexaploidization which appears to be at odds with the CR1-CR2 model derived here. I would like to see additional discussion why the hexaploidization model has been rejected in the current study.

We now support the hexaploidization model in our manuscript.

8. Gene regulatory evolution:

I was missing some clarity on the definition of distal regions among the different species analyzed for accessible chromatin regions. Have they been scaled in some way to genome size? Maybe I missed this information, but it was hard to find.

We should have described this point in a more clear way. In our analyses, we defined distal accessible chromatin regions (ACRs) as those not overlapping with exons or regions between 5kb upstream and 1 kb downstream of all TSSs of genes. In the analysis of Figure 5b, we used this definition for data of different species without scaling the genomic distances between ACRs and the closest TSSs based on the genome size. This is because we wanted to classify ACRs based on whether they could interact with the corresponding TSSs from distant places, possibly via 3D chromatin conformation.

Meanwhile, as for the analysis in Figure 5c, the difference in the genome size (to be more specific, the sizes of intergenic regions) needs to be considered because here we examined how the ACRs are distributed relative to TSSs of genes in each species genome. Therefore, we showed in our manuscript the distribution of the genomic distances of ACRs from the closest TSSs, not only without, but also with scaling based on the sizes of intergenic regions (Figure 5b and Extended Data Fig. 8c, respectively).

To avoid the confusion, we've amended the relevant sentences as follows:

Legend of Figure 5, Lines 1773-1774: *"For the result with scaling based on the average length of intergenic regions of each species genome, see Extended Data Figure 8c."*

Supplementary Information, Lines 2488-2491: *"For comparing the genomic distributions of ACRs relative to the TSSs of genes across species, the raw distances between ACRs and the closest TSSs are used for Figure 5 and Extended Data Figure 8d; distances were normalized by the average length of intergenic regions for Extended Data Figure 8c"*

9. Hagfish genome:

a) Karyotype:

I found it difficult to understand the description of the expected hagfish chromosome number to which the assembly is matched. Please say it more clearly in the text. I understand that the germline genome is $n=26$ so that we could expect the germline assembly here to have 26

50

scaffolds representing these chromosomes? The somatic genome should then have $n=18$ after elimination of 8 chromosomes. The description in the supplements was particularly confusing. Please revise karyotype descriptions and expectations for the assembled genome.

Exactly, the germline genome is $n=26$, but we cannot expect to get 26 scaffolds representing them because the 8 that are lost in somatic tissues consist of highly repetitive elements, and given that we did not use long-read sequencing technology, there was no expectation to assemble 26 clusters, but at least 18. Previous studies have suggested that the germline-specific chromosomes barely contain genes, and accordingly, 98.3% of the annotated genes are included in clusters 1 to 18. We have slightly modified this part in the main text, and made it clear in the Supplementary Information (section 1.1.6) *Investigation on hagfish Hi-C cluster 19 has been extensively modified*)

b) Sex chromosomes:

Please also provide a clearer explanation why an XY sex determination system is suggested from the genome assembly. Also, do you mean an XY system with highly sexually dimorphic chromosomes? While read-depth analyses could potentially identify such a system, what if the sex determination region is small and read-depth is only prevalent in a small portion of the chromosome (instead of showing a strong effect across the entire chromosome/cluster)? I think this part of the genome analysis is too preliminary and should either be expanded (e.g., by a sliding window analysis across clusters to identify potential regions with read-depth differences) or removed from the current study.

We have removed this section. We agree with the reviewer that the data is not conclusive. This will be addressed in a future study.

Additional Specific Points:

- 1. 109/110: 16,500 predicted genes seem a rather small number compared to gnathostomes. It would be good to compare here to the predicted total gene numbers in Sea and Arctic lampreys. Is this low number due to more gene "hiding" in hard to assemble microchromosomes? See also 2. Above.

The genome of the Arctic lamprey has been reported to have 19,455 genes annotated (Nakatani et al., 2021); the annotation of the latest chromosome-scale assembly of the sea lamprey contains 17,580 protein-coding genes; for comparisons, that of the elephant shark is 18,747 (Nakatani et al., 2021). We think our annotation is within the normal range, although in the lower tier. This can be due to the different tools used, since our annotation process was performed by the Ensembl team, which employs a very stringent and conservative pipeline, instead of other more common tools such as MAKER (used in most other genomes). This was indicated in Supplementary Information:

Supplementary, Lines 271-274: *"As such, the annotation relied heavily on the transcriptomic data over homology-based structures wherever possible. The protein-coding gene count of 16,513 is likely to be an underestimate of the true number of coding genes and more representative of highly conserved or highly expressed genes."*

- l. 124, better say: "... Although having an overall GC content similar to that of the lamprey..."

Changed as suggested.

- l. 131, say: "The hagfish genome..."

Changed as suggested.

- l. 134: "...larger size of hagfish genome than those of..." (delete 'and')

Changed as suggested.

- l.135/6: I'd avoid this statement to say that hagfish may be a better model than lamprey. Later in the text, it is said that lamprey is more suited for other types of analyses (l.245-46). Rather say something along the lines that hagfish genome provides essential, complementary information to genomes of lampreys.

We agree with the reviewer, and have changed our statement accordingly:

Lines 163-165: *"Altogether, the hagfish genome provides essential, complementary information to genomes of lampreys, especially in analyses such as gene tree reconstruction and comparative genomics."*

- l. 144/145: Briefly say what type of data are underlying the Bayesian phylogenetic analysis, even if it is explained in more detail in the supplements or elsewhere. Figure legend 2 also doesn't provide this information.

We have now added this information:

Lines 173-184: *"We used Bayesian inference to reconstruct the phylogeny of vertebrates from an alignment of 190 single copy genes in all taxa analysed (84,017 sites), strongly supporting a monophyletic Cyclostomata (Fig. 2, Extended Data Fig. 2a)."*

- l. 158: lowest amount of gene losses among what?

We have toned down this sentence, and now reads as follows:

Line 196-199: *"also characterised by a very low amount of gene losses (-341 and -382, respectively) when compared with other deuterostome and chordate nodes (Fig. 2; Extended Data Fig. 3a; Supplementary Information)".*

- l. 159/60, for clarity, say for example: "... highly retained novel gene families (also known as novel core genes, i.e., genes that are not lost in descendant lineages, by convention indicated with ++)."...

Thank you for the suggestion. We have modified it as suggested.

- I. 168, better say: "... gnathostomes and cyclostomes convergently evolved independent adaptive immune systems..."

We have changed it as suggested.

- I. 174: "Inferred rates of gene duplication..." Say more explicitly here that you do not distinguish types of duplication (at least this is my understanding).

The reviewer is correct, and we have added that information to make it clear.

- I. 176, say: "...1R, 2R, and teleost 3R WGD events"

Changed as suggested.

- I. 196/97: "the crown-cyclostome already possessed six Hox clusters, distinct from the crown-gnathostome ancestor" You could make it more explicit that this observation is another evidence for cyclostome monophyly.

Thank you for the suggestion. We have added a short sentence as suggested:

Line 251: "*This observation provides another evidence of cyclostome monophyly.*"

- I.199/20: This last sentence is confusing, as it is so vague – what is the point you want to make here? I think it could also just be deleted.

We have now changed this sentence to read as follows:

Lines 256-258: "*This implies that the events suggested from the different analyses of the Arctic lamprey genome, two extra WGDs (ref. 18) or a triplication (ref. 6), might have occurred in early cyclostome evolution, before the lamprey and hagfish divergence (refs. 6, 19)*"

- I. 203: with "pre-duplicative" do you mean "pre-WGD"?

Yes, and we have now changed it to avoid confusion.

- I. 213: provide some justification for the use of the sea cucumber genome as reference here instead of some other possible non-vertebrate genomes (e.g. Nakatani et al. 2021 and Simakov et al. 2020 used scallop). To be clear, I am not questioning its use but would like to know more why this genome has been chosen here over other options. It does not seem to be a chromosome-level genome assembly.

We started our comparative genomic analysis in 2018, a time when no chromosome-level assembly was available for any amphioxus species. The next closer lineages to vertebrates or chordates are within Ambulacraria, and the echinoderm sea cucumber *Apostichopus japonicus* (Zhang et al., 2017, ref. 42 in our revised version of the manuscript) was the only one with an available chromosome-level assembly with linkage group information. We have now reorganised and rewritten the whole section of the ancestral karyotype inference, both in the main text and in the Supplementary Information (section 4.5) *Inference of vertebrate ancestral chromosomes*,

including a macrosynteny comparison between *A. japonicus* and *B. floridae* genomes (subsection 4.5.3) *Comparing sea cucumber and Florida lancelet genomes as outgroup taxa for macrosynteny conservation analysis*), and added new Supplementary Table 35 with the homology relationships between *B. floridae* chromosomes and *A. japonicus* linkage groups. All in all, we conclude that the use of *A. japonicus* is a suitable alternative to *B. floridae* as an outgroup to vertebrates, and the results of our karyotype inference would have been the same.

- I. 217, for clarity say: "...in the gnathostomes chicken and gar..."

Changed as suggested.

- I. 223-24: Why is the lancelet gene number brought up, this is confusing. Either explain or delete the info.

We have modified this part of the main text and believe it is now clear why we use amphioxus genes to represent our ACs.

- I. 226: Is AC = amphioxus? Please explain.

This is related to the previous comment. Since we have modified the part of the ACs gene reconstruction, we believe that it is now clear in the manuscript that we proxy the genes of these ACs from amphioxus genes.

- I. 233-34: Please explain the switch to elephant shark as gnathostome representative here. Also see point 1. above about the elephant shark genome version, which does not seem to be chromosome-level.

We used elephant shark genes in this analysis only as a positive control for genes that would support both 1R and 2R: "including elephant shark orthologs as a control for the 2R signal". Since in this case a gene-level analysis rather than a synteny analysis was performed, the assembly level of the *C. milii* genome is irrelevant for this analysis.

- I. 240-41, say: "... of the inferred four pre-1R and..."

Changed as suggested.

- I. 261-263: I understand you mean to say that 2 Hox clusters of a total of potentially 8 Hox clusters after 1R->CR1->CR3 were lost along the way to the LCA of crown cyclostomes. See my comments on Hox clusters above (point 5.).

We have now changed our main conclusions and agree that our results support a triplication event in the cyclostome stem. The beginning of this section has been changed accordingly and thus the potential loss of 2 Hox clusters is no longer relevant.

- I. 293: What would be a comparable OR value for a different vertebrate lineage that is derived from 3 rounds of WGD, e.g. in zebrafish?

Thanks for raising this intriguing and important question. We have now calculated the OR values for zebrafish, *Danio rerio*, a teleost fish. Consistent with our predictions, the median OR is 0.276 for paralogous chromosome pairs in the zebrafish genome, which is much lower than the median OR value for chicken or spotted gar genomes (median OR 0.49 and 0.50, respectively) and similar to that of sea lamprey and hagfish (median OR 0.29 and 0.30, respectively).

We added these zebrafish results to the corresponding figure, now Supplementary Figure 29, new panel D).

These results have also been added to our Supplementary Information:

Supplementary Information, Lines 1903-1904: "Thus, as expected, after a third round of WGD, the mean OR of paralogous chromosomes would further drop. Presumably, triplication would lead to a similar drop."

Finally, we conducted a direct test of our aforementioned conclusion using the zebrafish genome, which underwent the teleost-specific third round of WGD. Our findings are aligned with our model, as evidenced by the zebrafish paralogous chromosome pairs displaying a median OR of 0.28 (Supplementary Figure 29). This value is much lower compared to the median OR values of the chicken and spotted gar genomes, which are 0.49 and 0.54, respectively. In contrast, it is much similar to the observed median OR in cyclostome species (0.30, Supplementary Table 47), consistent with a larger polyploid event than a duplication, such as the suggested triplication event. "

- I. 323, better say: "...using one embryos of *E. burgeri* at two different stages..."

We have changed the sentence to read as follows:

"using a total of two embryos of *E. burgeri*, each at a different stage"

- I. 324, please explicitly say what your comparison is, higher than amphioxus?

We have corrected this:

Lines 606-608: "We found a significantly higher number of ACRs per gene than in the cephalochordate amphioxus, similar to what has been observed in gnathostomes (Marlétaz et al., 2018) (Fig. 5a)"

- I. 351-352, please elaborate what you mean with deterministic and permissive.

Determinism reflects an inevitability, that a cause, like WGD, will lead directly to an effect, like phenotypic innovation or diversification. The permissive alternative reflects effects that could not have happened without a specific precondition, like WGD, but such effects were not inevitable.

- I. 383-84: Please explain your reasoning why the key vertebrate innovation would have originated in a post-1R tetraploid stem vertebrate, as opposed to a pre-1R stem vertebrate. What is the evidence that 1R came before these innovations?

Precisely, our sentence stated pre-2R, as we cannot reliably establish if they appeared before or after 1R, just along the stem vertebrate lineage.

- I. 570-71: In which ways were gene models manually curated?

Thanks for your question. It is an essential step in the analysis regarding the phylogenetic support around 1R/2R of cyclostome genes. This was explained in Supplementary Information, now section 4.6.1 (see lines 1438-1539 of the revised document). Without these curation, the gene models for some genes have problems, for instance due to putative misannotations (genes splitted into two gene models, non-real exons or exons missing), which could lead to unreliable multiple sequence alignments, confounding phylogenetic analyses. We provided all manually curated genes in Supplementary File 4.

We are showing below an example of an artificial split model. In this example, three sea lamprey genes, PMZ0042473, PMZ0042472, and PMZ0042471, are arranged in tandem in the sea lamprey supra-scaffold 18. They are all homologous to one lancelet gene XP_019644382, which is on AC3. In the multiple sequence alignment of protein sequences below, we noticed that each sea lamprey gene was aligned exclusively to different regions in the multiple sequence alignment (see below). Therefore, the three sea lamprey genes should in fact belong to one intact gene which has been erroneously split in the current annotation. In our manual curation, the separately aligned parts of these three sea lamprey genes are concatenated as one protein sequence.

In another example shown below, we found there is a stretch of sequence potentially missing from the sea lamprey protein PMZ_0030219 (see below, panel A, dashed box). So, we mapped the protein sequence of PMZ_0030219 to the genome in UCSC genome browser (see below, panel B) and founded the automatically generated AUGUSTUS gene models are able to complement that lost sequence in PMZ_0030219 (see below, panel B, dashed box). Then we manually inserted the missing sequence from the AUGUSTUS gene model into the sequence of PMZ_0030219. The corrected sequence now has its gap filled (see below, panel A, dashed box).

- I. 583-84: Overall, the RSCU analysis is not well explained. Why is it relevant and what is its purpose?

Sorry for the confusion. We revised the Online Methods accordingly.

Lines 1077-1080: "RSCU (*relative synonymous codon usage*) calculates the relative synonymous codon usage on degenerative sites of third codon positions, which is independent from the amino acid usage. Therefore, RSCU is used as a robust measurement of GC bias in codons. Correspondence analysis of RSCU values was performed with *codonW* according to Smith et al. 2013".

The purpose of comparing the GC bias of different species is to corroborate previous results that the sea lamprey genome has a strong GC bias in the third position of codons compared to the genome-wide level (75% vs. 46%, respectively) and to analyze in which situation the hagfish genome is, for the first time in a genome wide manner. Since RSCU is free of amino acid related selection force, it is a robust measurement of GC bias in codons.

- I. 591: It would be good to also provide statistics for the vertebrata dataset – which will be highly biased towards gnathostomes.

Thank you for your suggestion. As the reviewer commented, the Vertebrata dataset is highly biased towards gnathostomes (see image below). This was expected as the Vertebrata dataset used in BUSCO consists mainly of orthologs from bony fishes (see Hara et al., 2015. *BMC Genomics* 16, 977. doi:10.1186/s12864-015-2007-1), therefore we do not believe that adding this information, that is expected beforehand to be biased, provides significant information to our manuscript. We prefer not to add a supplementary figure with this information.

We would like to point out that we have updated our BUSCO analyses (with the Metazoa dataset) using the latest chromosome-level assembly for the sea lamprey (Timoshevskaya et al., 2023) and the latest chicken genome (version 7.0). We have modified the corresponding section in Online Methods, added citations to the exact genomes used and updated the corresponding panels (Figure 1e and Extended Data Figure 1c).

- l. 674-99: It is unclear which figures this part refers to, please include figure numbers.

We added a sentence in the corresponding method section:

Lines 1254-1256: *"the dates of species divergence is presented in Fig. 2; and dating of WGD events in Fig. 2 and Supplementary Fig. 34"*.

- l. 711, say: "By definition..."

This sentence has been removed.

- l. 754-66: Please make it more explicit that you are only measuring on/off states here (at least this is my understanding). It would be helpful to refer to the relevant figure in the supplements that show the definitions (Suppl. Fig. 30).

I am not convinced that you can infer 'subfunctionalization' with data from ingroups only and without evidence for the ancestral, pre-duplication state inferred from an outgroup. As presented, some expression domains could still be novel. Thus, without outgroup information, I'd rather call this 'tissue-specific expression' or 'divergent expression'.

A pre-duplication state is lacking in our analysis due to the unavailability of outgroup data: there are only five overlapping tissues with available RNA-seq data from amphioxus. So, it is practically hard to distinguish among redundancy, subfunctionalization and specialization with only these five tissues. Therefore, we approximated the three types of expressional evolution based on the different level of divergent expression patterns for hagfish ohnologs.

This figure is now Supplementary Figure 31. We have added a scheme of mild specialization cases, as suggested in a comment below, and also defined it in the Online Methods (subsection *Fate of*

ohnologs after WGD) and Supplementary Information (subsection 5.1.3) *Fate of ohnologs after WGD*). Following the reviewer's suggestion, we have added the definition of on/off states in the legend of the figure. How we define 'expressed' or 'not expressed' was already stated in our Online Methods section (subsection *Fate of ohnologs after WGD*)

The terms 'tissue-specific expression' or 'divergent expression', as suggested by the reviewer, are reasonable terms. However, to follow the context of analysis of previous studies in the field (Marlétaz et al., 2018), we prefer to tone down the terms as 'potential subfunctionalization', 'potential specialization' and 'potential redundancy', and we have modified the manuscript accordingly.

In addition, a call to Supplementary Figure 31 has been added in the main text, Online Methods and Supplementary Information.

- I. 779: ... the tree is available..."

Corrected.

- Fig. 1. I. 986: lungfish -> hagfish

Corrected.

- Fig. 2: are there also data for gains and losses in Chondrichthyes? What data is this phylogeny based on?

We did not infer these data in Chondrichthyes, and although it would be possible to infer, we did not analyse younger nodes than those of crown vertebrates, and then gnathostomes and cyclostomes. Lineages within each of the two groups of vertebrates were used to infer Novel versus Novel Core gene families. However, after a comment from Reviewer #2, we now have added a citation to Tan et al., 2021, *Elife* 10, e65394, who presents a more detailed analysis around Chondrichthyes.

The data this phylogeny is based on is explained in the Online Methods, in sections *Species tree inference*. Very briefly, first, the topology of the tree is obtained from the analysis of a multisequence alignment of 190 single copy genes (84,017 sites), which is available within the Supplementary File 13, and using PhyloBayes.

- I. 1052: should be "ohnologous". The term ohnolog does not seem to be defined in the main text. Please do so at first appearance.

The typo has been corrected, and a definition added:

Lines 456-457: "*ohnologous (duplicates that originate through WGD, after Susumu Ohno (Ohno, 1970)) chromosome pairs*".

- Supplements, I. 145: What do you mean with the sex determination system is "mostly unknown"?

We have now removed this section.

- Supplements, I. 622-631: It would be interesting to know which signalling pathways were differently retained between gnathostomes and cyclostomes?

Novel Core families appear in the last common ancestor of vertebrates, and therefore are retained in both gnathostomes and cyclostomes. We think that investigating those signalling pathways that do not constitute novel core homology groups are beyond the scope of our manuscript. But, it is a very interesting question to address in a future study using the resources provided here.

- Supplements, I. 629, it sounds strange to say "In contrast to vertebrates, gnathostomes retained genes...". Please rephrase.

We have slightly modified this paragraph and rephrased this sentence.

- Supplements, I.1385-85. The last sentence is unclear, please rephrase.

We have updated the sentence to present a clearer point.

Lines 2460-2461: "*ACRs falling into the cis-regulatory region of one gene were considered to be related to its transcriptional regulation.*"

- Supplements, 5. Regulatory Evolution: This section needs to be reorganized and streamlined as it is difficult to follow how the different subsections connect to each other. For example, is 5.1 the overall discussion of the results?

We apologise that our descriptions included in Supplementary Information Section 5 were not sufficiently organised. To ensure clarity, we revised and polished this section.

- Suppl. Fig. 30: Please refer to this figure when presenting definitions of ohnolog fates in the text. I'd also include a scheme to define "mild specialization" as shown in Extended Fig. 8. I couldn't find a definition of this case.

This is now Supplementary Figure 31, and we have added the requested scheme.

- Suppl. Fig. 31: This figure appears to be essential to understand the dating of WGDs but the figure legend is too minimal to understand the figure. Please explain in detail here and/or in the relevant text section; see also my main point on WGD dating above (6.).

We have further elaborated this figure legend.

Decision Letter, first revision:

10th October 2023

Dear Juan,

Thank you for submitting your revised manuscript "Hagfish genome elucidates vertebrate whole genome duplication events and their evolutionary consequences" (NATECOLEVOL-23040791A). It has now been seen again by the original reviewers and their comments are below. The reviewers find that the paper has improved in revision, and therefore we'll be happy in principle to publish it in Nature Ecology & Evolution, pending minor revisions to satisfy the reviewers' final requests and to comply with our editorial and formatting guidelines. I should stress that we will need to see meaningful discussion related to the comments of the first two reviewers.

Sincerely,

[REDACTED]

Reviewer #1 (Remarks to the Author):

First, I'd like to thank the authors for their detailed response to my previous review comments.

In the revised manuscript, the authors kept their analyses mostly unchanged, but they seem to have reversed the conclusion (the previous manuscript proposed two WGDs in the cyclostome lineage, but the revised manuscript supports genome triplication). This makes me concerned about the robustness of their analyses. In the revised manuscript, the authors do not critically compare the previous two-WGD scenario with the triplication scenario (instead they simply deleted discussions for supporting the two-WGD scenario. e.g. the following text was deleted: "Actually, the observed median OR in cyclostome species is 0.30, fitting well with the simulated scenario where cyclostomes were subject to three rounds of WGDs"). Additional analysis/discussion of this point in the main text would strengthen the manuscript, since the previous manuscript and the revised manuscript seem to indicate that the authors' OR-based analysis supports both duplication scenarios. Also, additional discussion of the limitations of the OR-based analysis would be helpful in understanding the robustness and accuracy of the authors' analysis. (The authors wrote a detailed discussion of genome triplication in their response to the reviewers, but not in the main text.)

62Overall, the revised manuscript discusses some interesting analyses that demonstrate the importance of the hagfish genome sequence for evolutionary research:

(1) Although it seems that the authors' OR analysis is not specifically designed to accept/reject the two-WGD scenario, the hagfish genome analysis is consistent with the previously proposed model of cyclostome genome triplication; (2) The hagfish genome analysis confirms that the hagfish and lamprey lineages diverged after the genome triplication event, suggesting that the hagfish genome is important for understanding how the genome triplication event affected the evolution of cyclostomes; (3) The morphological disparity analysis shows that the 2R event was important for the evolution of morphological diversity in gnathostomes.

Thus, I hope that the manuscript will eventually be published in Nature Ecology and Evolution.

Specific Comments on revised texts:

Lines 257-258: "Therefore, although we infer the number of ancestral vertebrate chromosomes to be likely 17, a scenario with 18 chromosomes^{6,38} cannot be ruled out."

It was not clear to me why the authors' "less parsimonious"(Line 253) reconstruction ($n=17$) is more likely than the previous reconstructions ($n=18$) in Refs 6 and 38.

Lines 211-275: "Ancestral vertebrate karyotype"

In the first paragraph of this section, the authors wrote "The inclusion of a hagfish genome in analyses pertaining to the reconstruction of the ancestral karyotype ...", but the hagfish genome was not used at all in this section? This is confusing to me.

Lines 300-301: "but constraints 2R to the gnathostome lineage as recently suggested in similar analyses^{5,6}"

Typo: missing period.

Lines 336-338: "We do not find more than 6 descendent copies from any AC, supporting a whole genome triplication in the cyclostome lineage as previously proposed with the analysis of the lamprey genome 6."

Is it possible that some ohnologous chromosomes are missing in the analysis with $OR > 0.15$, as we can see in Supplementary Fig. 29 (OR values between duplicated chromosomes in zebrafish) in the revised manuscript? A lower threshold for the OR value might detect more ohnologous chromosomes (e.g. six ohnologous chromosomes for more ACs or eight ohnologous chromosomes for some ACs)?

Line 1213 in Supplementary Information: "4.9) Investigating whether lamprey lineage shared gnathostome pre-1R and post-1R fusions."

In the revised manuscript, the hagfish genome was partially used in this analysis, so the section title should be "cyclostome lineage"?

Line 1227 in Supplementary Information:

typo: "becasue"

Finally, some line numbers mentioned in the authors' rebuttal file are not found in the revised

manuscript (.pdf/.docx) or in the supplementary information: e.g. "In our study, this is 50 chromosomes (Supp. Information lines 1572-1573, see below) but 54 in ref. 11."

Reviewer #2 (Remarks to the Author):

Reviewer: Aoife McLysaght

I appreciate the authors' sincere efforts to address the reviewer comments and I am satisfied that the manuscript is much improved.

My only remaining comment is that I would appreciate a bit more text to elaborate on the implications of the inference of only 17, not 18 pre-1R chromosomes. The authors here insist (page 9) that 17 is more parsimonious because otherwise a similar fusion occurs twice post-1R in gnathostomes (but not cyclostomes). However, they do not fully acknowledge that their model requires two post-1R fissions that reconstitute the pre-fusion chromosomes in cyclostomes. I understand that the authors are committed to their 17 chromosome reconstruction, and I am not asking them to alter that, however, I think the discussion of parsimony is inadequate and incomplete without considering the parsimony of the two fission events that reverses a (fairly) recent fusion post-1R. My personal view is that the symmetric fission x2 is not very likely. The authors can give their view of this in the text.

Minor comment: on page 12, line 477, for completeness state that the interspecific homologous chromosomes are more similar in the instance where rediploidisation precedes speciation.

Fig 4 title has a hanging 'and'.

Reviewer #3 (Remarks to the Author):

In the revised version, the authors have addressed all my previous concerns appropriately - and in my assessment, also the issues raised by the other reviewers. The writing of the new version has much improved as it much better allows to understand the extensive and complex set of analyses and conclusions.

This study will be a highly important contribution to our understanding of the early evolution of the vertebrate lineage in the face of multiple WGDs.

I have one last item for the final version of the article, getting back to a point I raised before:

l.446-447 "Thus, key vertebrate innovations (e.g., elaborate tripartite brain, neural crest cell derived tissues among other novelties) originated in a tetraploid stem-vertebrate, before 2R."

In response to my previous question, the authors wrote that "we cannot reliably establish if they [the innovations] appeared before or after 1R, just along the stem vertebrate lineage."

64I think that the word "tetraploid" needs to be deleted from the sentence in l.446-447, as it implies that 1R occurred before the innovations. Or, to make this even more clear, rephrase the sentence to something along the lines of:

"Thus, key vertebrate innovations (e.g., elaborate tripartite brain, neural crest cell derived tissues among other novelties) originated in a stem-vertebrate before 2R. However, at this point we cannot reliably establish if these innovations pre- or post-date the 1R event."

Our ref: NATECOLEVOL-23040791A

26th October 2023

Dear Dr. Pascual-Anaya,

Thank you for your patience as we've prepared the guidelines for final submission of your Nature Ecology & Evolution manuscript, "Hagfish genome elucidates vertebrate whole genome duplication events and their evolutionary consequences" (NATECOLEVOL-23040791A). Please carefully follow the step-by-step instructions provided in the attached file, and add a response in each row of the table to indicate the changes that you have made. Please also check and comment on any additional marked-up edits we have proposed within the text. Ensuring that each point is addressed will help to ensure that your revised manuscript can be swiftly handed over to our production team.

****We would like to start working on your revised paper, with all of the requested files and forms, as soon as possible (preferably within two weeks). Please get in contact with us immediately if you anticipate it taking more than two weeks to submit these revised files.****

In recognition of the time and expertise our reviewers provide to Nature Ecology & Evolution's editorial process, we would like to formally acknowledge their contribution to the external peer review of your manuscript entitled "Hagfish genome elucidates vertebrate whole genome duplication events and their evolutionary consequences". For those reviewers who give their assent, we will be publishing their names alongside the published article.

Nature Ecology & Evolution offers a Transparent Peer Review option for new original research

65manuscripts submitted after December 1st, 2019. As part of this initiative, we encourage our authors to support increased transparency into the peer review process by agreeing to have the reviewer comments, author rebuttal letters, and editorial decision letters published as a Supplementary item. When you submit your final files please clearly state in your cover letter whether or not you would like to participate in this initiative. Please note that failure to state your preference will result in delays in accepting your manuscript for publication.

Cover suggestions

We welcome submissions of artwork for consideration for our cover. For more information, please see our [guide for cover artwork](https://www.nature.com/documents/Nature_covers_author_guide.pdf).

Nature Ecology & Evolution has now transitioned to a unified Rights Collection system which will allow our Author Services team to quickly and easily collect the rights and permissions required to publish your work. Approximately 10 days after your paper is formally accepted, you will receive an email in providing you with a link to complete the grant of rights. If your paper is eligible for Open Access, our Author Services team will also be in touch regarding any additional information that may be required to arrange payment for your article.

Please note that *Nature Ecology & Evolution* is a Transformative Journal (TJ). Authors may publish their research with us through the traditional subscription access route or make their paper immediately open access through payment of an article-processing charge (APC). Authors will not be required to make a final decision about access to their article until it has been accepted. [Find out more about Transformative Journals](https://www.springernature.com/gp/open-research/transformative-journals)

Authors may need to take specific actions to achieve [compliance with funder and institutional open access mandates](https://www.springernature.com/gp/open-research/funding/policy-compliance-faqs). If your research is supported by a funder that requires immediate open access (e.g. according to [Plan S principles](https://www.springernature.com/gp/open-research/plan-s-compliance)) then you should select the gold OA route, and we will direct you to the compliant route where possible. For authors selecting the subscription publication route, the journal's standard licensing terms will need to be accepted, including [self-archiving-and-license-to-publish](https://www.nature.com/nature-portfolio/editorial-policies/self-archiving-and-license-to-publish). Those licensing terms will supersede any other terms that the author or any third party may assert apply to any version of the manuscript.

For information regarding our different publishing models please see our [page](https://www.springernature.com/gp/open-research/transformational-journals). If you have any questions about costs, Open Access requirements, or our legal forms, please contact ASJournals@springernature.com.

[REDACTED]

Best regards,

[REDACTED]

Reviewer #1:

Remarks to the Author:

First, I'd like to thank the authors for their detailed response to my previous review comments.

In the revised manuscript, the authors kept their analyses mostly unchanged, but they seem to have reversed the conclusion (the previous manuscript proposed two WGDs in the cyclostome lineage, but the revised manuscript supports genome triplication). This makes me concerned about the robustness of their analyses. In the revised manuscript, the authors do not critically compare the previous two-WGD scenario with the triplication scenario (instead they simply deleted discussions for supporting the two-WGD scenario. e.g. the following text was deleted: "Actually, the observed median OR in cyclostome species is 0.30, fitting well with the simulated scenario where cyclostomes were subject to three rounds of WGDs"). Additional analysis/discussion of this point in the main text would strengthen the manuscript, since the previous manuscript and the revised manuscript seem to indicate that the authors' OR-based analysis supports both duplication scenarios. Also, additional discussion of the limitations of the OR-based analysis would be helpful in understanding the robustness and accuracy of the authors' analysis. (The authors wrote a detailed discussion of genome triplication in their response to the reviewers, but not in the main text.)

Overall, the revised manuscript discusses some interesting analyses that demonstrate the importance of the hagfish genome sequence for evolutionary research:

- (1) Although it seems that the authors' OR analysis is not specifically designed to accept/reject the two-WGD scenario, the hagfish genome analysis is consistent with the previously proposed model of cyclostome genome triplication;
- (2) The hagfish genome analysis confirms that the hagfish and lamprey lineages diverged after the genome triplication event, suggesting that the hagfish genome is important for understanding how the genome triplication event affected the evolution of cyclostomes;
- (3) The morphological disparity analysis shows that the 2R event was important for the evolution of morphological diversity in gnathostomes.

67Thus, I hope that the manuscript will eventually be published in Nature Ecology and Evolution.

Specific Comments on revised texts:

Lines 257-258: "Therefore, although we infer the number of ancestral vertebrate chromosomes to be likely 17, a scenario with 18 chromosomes^{6,38} cannot be ruled out."

It was not clear to me why the authors' "less parsimonious"(Line 253) reconstruction (n=17) is more likely than the previous reconstructions (n=18) in Refs 6 and 38.

Lines 211-275: "Ancestral vertebrate karyotype"

In the first paragraph of this section, the authors wrote "The inclusion of a hagfish genome in analyses pertaining to the reconstruction of the ancestral karyotype ...", but the hagfish genome was not used at all in this section? This is confusing to me.

Lines 300-301: "but constraints 2R to the gnathostome lineage as recently suggested in similar analyses^{5,6}"

Typo: missing period.

Lines 336-338: "We do not find more than 6 descendent copies from any AC, supporting a whole genome triplication in the cyclostome lineage as previously proposed with the analysis of the lamprey genome 6."

Is it possible that some ohnologous chromosomes are missing in the analysis with OR>0.15, as we can see in Supplementary Fig. 29 (OR values between duplicated chromosomes in zebrafish) in the revised manuscript? A lower threshold for the OR value might detect more ohnologous chromosomes (e.g. six ohnologous chromosomes for more ACs or eight ohnologous chromosomes for some ACs)?

Line 1213 in Supplementary Information: "4.9) Investigating whether lamprey lineage shared gnathostome pre-1R and post-1R fusions."

In the revised manuscript, the hagfish genome was partially used in this analysis, so the section title should be "cyclostome lineage"?

Line 1227 in Supplementary Information:

typo: "becasue"

Finally, some line numbers mentioned in the authors' rebuttal file are not found in the revised manuscript (.pdf/.docx) or in the supplementary information: e.g. "In our study, this is 50 chromosomes (Supp. Information lines 1572-1573, see below) but 54 in ref. 11."

Reviewer #2:

Remarks to the Author:

Reviewer: Aoife McLysaght

I appreciate the authors' sincere efforts to address the reviewer comments and I am satisfied that the

68manuscript is much improved.

My only remaining comment is that I would appreciate a bit more text to elaborate on the implications of the inference of only 17, not 18 pre-1R chromosomes. The authors here insist (page 9) that 17 is more parsimonious because otherwise a similar fusion occurs twice post-1R in gnathostomes (but not cyclostomes). However, they do not fully acknowledge that their model requires two post-1R fissions that reconstitute the pre-fusion chromosomes in cyclostomes. I understand that the authors are committed to their 17 chromosome reconstruction, and I am not asking them to alter that, however, I think the discussion of parsimony is inadequate and incomplete without considering the parsimony of the two fission events that reverses a (fairly) recent fusion post-1R. My personal view is that the symmetric fission x2 is not very likely. The authors can give their view of this in the text.

Minor comment: on page 12, line 477, for completeness state that the interspecific homologous chromosomes are more similar in the instance where rediploidisation precedes speciation.

Fig 4 title has a hanging 'and'.

Reviewer #3:

Remarks to the Author:

In the revised version, the authors have addressed all my previous concerns appropriately - and in my assessment, also the issues raised by the other reviewers. The writing of the new version has much improved as it much better allows to understand the extensive and complex set of analyses and conclusions.

This study will be a highly important contribution to our understanding of the early evolution of the vertebrate lineage in the face of multiple WGDs.

I have one last item for the final version of the article, getting back to a point I raised before:

l.446-447 "Thus, key vertebrate innovations (e.g., elaborate tripartite brain, neural crest cell derived tissues among other novelties) originated in a tetraploid stem-vertebrate, before 2R."

In response to my previous question, the authors wrote that "we cannot reliably establish if they [the innovations] appeared before or after 1R, just along the stem vertebrate lineage."

I think that the word "tetraploid" needs to be deleted from the sentence in l.446-447, as it implies that 1R occurred before the innovations. Or, to make this even more clear, rephrase the sentence to something along the lines of:

"Thus, key vertebrate innovations (e.g., elaborate tripartite brain, neural crest cell derived tissues among other novelties) originated in a stem-vertebrate before 2R. However, at this point we cannot reliably establish if these innovations pre- or post-date the 1R event."

69Author Rebuttal, first revision:Note: Line numbers are indicated according to the .docx document with changes not tracked.

Reviewer #1 (Remarks to the Author):

First, I'd like to thank the authors for their detailed response to my previous review comments.

In the revised manuscript, the authors kept their analyses mostly unchanged, but they seem to have reversed the conclusion (the previous manuscript proposed two WGDs in the cyclostome lineage, but the revised manuscript supports genome triplication). This makes me concerned about the robustness of their analyses. In the revised manuscript, the authors do not critically compare the previous two-WGD scenario with the triplication scenario (instead they simply deleted discussions for supporting the two-WGD scenario. e.g. the following text was deleted: "Actually, the observed median OR in cyclostome species is 0.30, fitting well with the simulated scenario where cyclostomes were subject to three rounds of WGDs"). Additional analysis/discussion of this point in the main text would strengthen the manuscript, since the previous manuscript and the revised manuscript seem to indicate that the authors' OR-based analysis supports both duplication scenarios. Also, additional discussion of the limitations of the OR-based analysis would be helpful in understanding the robustness and accuracy of the authors' analysis. (The authors wrote a detailed discussion of genome triplication in their response to the reviewers, but not in the main text.)

We appreciate the Reviewer's thoughtful consideration and would like to clarify that our intention was not to reverse our conclusion but rather to adjust it based on a more plausible scenario supported by our analyses. The OR metric remains a robust tool for distinguishing between orthologous and ohnologous chromosome relationships, and we have retained this aspect of our analysis without significant alterations. Our modifications stem from a rigorous examination of the data and a reinterpretation of our findings in light of the reviewers' constructive feedback.

The observed median OR in cyclostome species consistently points to a maximum of six ohnologous chromosomal regions in both hagfish and lamprey. While our initial interpretation suggested two rounds of WGD followed by losses, a more parsimonious explanation, supported by the OR analysis, is genome triplication. We acknowledge that our previous proposal was influenced by the clustering results of the retention profiles, mistakenly considering those as phylogenetic evidence.

In response to the Reviewer's suggestion, we recognize the importance of providing a critical comparison between the two-WGD and triplication scenarios. We have now included an additional, short statement in the main text proposing that although our analysis cannot reject the alternative scenario of two cyclostome-specific WGDs followed by chromosome losses, this is less parsimonious than a triplication event. These lines read as follows:

Lines 352-355: *"While our data does not definitively rule out the possibility of two cyclostome specific WGD events followed by extensive chromosome losses, this scenario is less plausible"*

than a single triplication event, particularly given the absence of instances with eight copies of any chromosomal region in the lamprey or the hagfish."

We thank the Reviewer because this addition will contribute to a more comprehensive understanding of our findings, strengthening the overall manuscript.

Overall, the revised manuscript discusses some interesting analyses that demonstrate the importance of the hagfish genome sequence for evolutionary research:

(1) Although it seems that the authors' OR analysis is not specifically designed to accept/reject the two-WGD scenario, the hagfish genome analysis is consistent with the previously proposed model of cyclostome genome triplication; (2) The hagfish genome analysis confirms that the hagfish and lamprey lineages diverged after the genome triplication event, suggesting that the hagfish genome is important for understanding how the genome triplication event affected the evolution of cyclostomes; (3) The morphological disparity analysis shows that the 2R event was important for the evolution of morphological diversity in gnathostomes.

Thus, I hope that the manuscript will eventually be published in Nature Ecology and Evolution.

We thank the reviewer for their positive comments. We would also like to express our gratitude for the Reviewer's thorough revision of our manuscript; their insightful comments have been instrumental in refining our work.

Specific Comments on revised texts:

Lines 257-258: "Therefore, although we infer the number of ancestral vertebrate chromosomes to be likely 17, a scenario with 18 chromosomes^{6,38} cannot be ruled out."

It was not clear to me why the authors' "less parsimonious"(Line 253) reconstruction ($n=17$) is more likely than the previous reconstructions ($n=18$) in Refs 6 and 38.

We thank the Reviewer for raising this important point and apologise for the inadequacy of our explanation. This point has also been raised by Reviewer #2 so we have addressed it in detail. First, we apologise for the lack of clarity in our writing. Indeed, as the Reviewers correctly point out, a parsimony explanation is inadequate, because although an 18-chromosome model requires two pairwise post-1R fusions in the gnathostome lineage, our model requires both AC3-derived, post-1R chromosomes to symmetrically split in cyclostomes. However, there is a high number of chromosomal rearrangement events that account for cyclostome karyotypes compared to those of gnathostomes; therefore, we believe that a pairwise fission of AC3 in cyclostomes is more plausible than a pairwise fusion in the gnathostomes, although the other scenario is also possible.

We agree with the reviewer that suggesting our model to be more parsimonious is inadequate, so we have now modified our text to better explain why we are inclined to support an ancestral vertebrate karyotype consisting of 17 chromosomes as more plausible, while still acknowledging that our model cannot reject the alternative scenario. We have also updated Supplementary

72

2Figure 23 to better represent both alternative scenarios. The modified text reads as follows:

Lines 252-268: *"The difference with previous 18-chromosome models is that while we consider that the vertebrate AC3 is a single chromosome resulting from a pre-1R fusion event of two ancestral chordate chromosomes (Supplementary Fig. 23a), others (Nakatani et al., 2021; Simakov et al., 2022, 2020) consider that these two chromosomes remained separate through 1R (Nakatani's Pvc8 and 9, or Simakov's CLGQ and CLGI, respectively). While Pvc8/CLGQ and Pvc9/CLGI are consistently co-located in gnathostome chromosomes, they remain separate in invertebrate karyotypes (Nakatani et al., 2021; Simakov et al., 2022, 2020). We did not find any signals of linkage between Pvc8/CLGQ and Pvc9/CLGI in the lamprey and the hagfish genomes (Supplementary Figs. 25, 26). Therefore, there exists two alternative scenarios: (i) the 18-chromosome model implies that two independent pairwise fusions occurred after 1R in a stem gnathostome, mimicking a single pre-1R fusion event (Supplementary Fig. 23a); and (ii) our 17-chromosome model requires symmetric fissions of two AC3-derived post-1R chromosomes occurring in an ancestral cyclostome (Supplementary Fig. 23b). Although in silico simulations show that a scenario of pairwise post-1R fusions would not be extremely rare (30% of cases expected by chance; Supplementary Table 40; Methods), we believe the pairwise fissions to be more plausible given the higher level of reorganisation found in cyclostome karyotypes (see next section). Altogether, while we propose a scenario involving 17 ancestral vertebrate chromosomes, a scenario with 18 chromosomes (Nakatani et al., 2021; Simakov et al., 2022) is also possible."*

Lines 211-275: "Ancestral vertebrate karyotype"

In the first paragraph of this section, the authors wrote "The inclusion of a hagfish genome in analyses pertaining to the reconstruction of the ancestral karyotype ...", but the hagfish genome was not used at all in this section? This is confusing to me.

We thank the Reviewer for pointing this out. We have now removed this sentence.

Lines 300-301: "but constraints 2R to the gnathostome lineage as recently suggested in similar analyses^{5,6}"

Typo: missing period.

Thank you for spotting this typo; it has now been corrected.

Lines 336-338: "We do not find more than 6 descendent copies from any AC, supporting a whole genome triplication in the cyclostome lineage as previously proposed with the analysis of the lamprey genome 6."

Is it possible that some ohnologous chromosomes are missing in the analysis with $OR > 0.15$, as we can see in Supplementary Fig. 29 (OR values between duplicated chromosomes in zebrafish) in the revised manuscript? A lower threshold for the OR value might detect more ohnologous chromosomes (e.g. six ohnologous chromosomes for more ACs or eight ohnologous chromosomes for some ACs)?

The distribution of OR values in the zebrafish genome is based on putative, but not bona fide, ohnologous pairs. Consequently, a few OR values lower than 0.15 are conceivable. However, we acknowledge that factors such as (1) rampant gene losses, (2) complex genome duplication, and (3) chromosomal fusion/fission events may contribute to lower OR values for bona fide ohnologous chromosome pairs.

Regarding the second question, using a lower threshold for the OR value did not result in the detection of more than six ohnologous chromosomes in our reanalysis. Detailed information on all OR values for each AC is provided in Suppl. File 9. Nonetheless, it is noteworthy that we could identify one additional AC with six ohnologous chromosomes in both the hagfish and the lamprey (see below Revision Figure 1, depicting the AC6 case, adapted from Suppl. File 9, page 6) when the OR cut-off is relaxed. For instance, when the cut-off is relaxed to 0.13, six hagfish chromosomes become mutually paralogous (see below Revision Figure 1, the left panel). Further relaxing the cut-off to 0.09 reveals six sea lamprey chromosomes as mutually paralogous (see below, the right panel). However, it is important to acknowledge that lower cutoff values may increase the risk of false positives. Therefore, we opted to maintain a relatively stringent cutoff of 0.15.

Revision Figure 1. The OR values between all putative ohnologous chromosomes corresponding to AC6. This figure follows the same convention than Extended Data Fig. 7A. Numbers in colour-coded cells (bottom left triangle) indicate the OR between two chromosomes. Numbers in white cells (top right triangle) indicate the number of shared retained genes between two chromosomes. Numbers on the diagonal line from top left to bottom right (thick-lined cells) indicate the total number of retained genes of a chromosome. Green arrows are added to indicate the OR values lower than 0.15.

Line 1213 in Supplementary Information: "4.9) Investigating whether lamprey lineage shared gnathostome pre-1R and post-1R fusions."

In the revised manuscript, the hagfish genome was partially used in this analysis, so the section title should be "cyclostome lineage"?

We have now corrected the section title.

Line 1227 in Supplementary Information:
typo: "becasue"

Corrected.

Finally, some line numbers mentioned in the authors' rebuttal file are not found in the revised manuscript (.pdf/.docx) or in the supplementary information: e.g. "In our study, this is 50 chromosomes (Supp. Information lines 1572-1573, see below) but 54 in ref. 11."

We apologise for the inconvenience. We were aware before submission that our tracked Word file and the generated PDF file did not align in terms of line numbers, which may have caused some confusion. In the specific example mentioned by the Reviewer, the correct lines in our previous submission were 1586-87 in the tracked .docx (with Track Changes set to Markup Options → Balloons → Show Revisions in Balloons).

Reviewer #2 (Remarks to the Author):

Reviewer: Aoife McLysaght

I appreciate the authors' sincere efforts to address the reviewer comments and I am satisfied that the manuscript is much improved.

My only remaining comment is that I would appreciate a bit more text to elaborate on the implications of the inference of only 17, not 18 pre-1R chromosomes. The authors here insist (page 9) that 17 is more parsimonious because otherwise a similar fusion occurs twice post-1R in gnathostomes (but not cyclostomes). However, they do not fully acknowledge that their model requires two post-1R fissions that reconstitute the pre-fusion chromosomes in cyclostomes. I understand that the authors are committed to their 17 chromosome reconstruction, and I am not asking them to alter that, however, I think the discussion of parsimony is inadequate and incomplete without considering the parsimony of the two fission events that reverses a (fairly) recent fusion post-1R. My personal view is that the symmetric fission x2 is not very likely. The authors can give their view of this in the text.

We apologise for the lack of clarity in our writing. Indeed, as the Reviewer correctly points out, our model requires both AC3-derived, post-1R chromosomes to split in cyclostomes. Therefore, a parsimony explanation is inadequate. However, we disagree with the Reviewer and think that the symmetric fission x2 is indeed more plausible than symmetric pairwise fusions in gnathostomes. There is a higher number of chromosomal rearrangement events that account for cyclostome karyotypes compared to those of gnathostomes; therefore, we believe that pairwise fissions of AC3 in cyclostomes is more plausible than a pairwise fusion in the gnathostomes.

We agree with the reviewer that suggesting our model to be more parsimonious is categorically inadequate, so we have now modified our text to better explain why we suggest that an ancestral vertebrate karyotype consisting of 17 chromosomes is more plausible, while still acknowledging that our model cannot reject the alternative scenario of 18 chromosomes, which is also possible. We have also updated Supplementary Figure 23 to better represent both alternative scenarios. This modified text reads as follows:

Lines 252-268: "The difference with previous 18-chromosome models is that while we consider that the vertebrate AC3 is a single chromosome resulting from a pre-1R fusion event of two ancestral chordate chromosomes (Supplementary Fig. 23a), others (Nakatani et al., 2021; Simakov et al., 2022, 2020) consider that these two chromosomes remained separate through 1R (Nakatani's Pvc8 and 9, or Simakov's CLGQ and CLGI, respectively). While Pvc8/CLGQ and Pvc9/CLGI are consistently co-located in gnathostome chromosomes, they remain separate in invertebrate karyotypes (Nakatani et al., 2021; Simakov et al., 2022, 2020). We did not find any signals of linkage between Pvc8/CLGQ and Pvc9/CLGI in the lamprey and the hagfish genomes (Supplementary Figs. 25, 26). Therefore, there exists two alternative scenarios: (i) the 18-chromosome model implies that two independent pairwise fusions occurred after 1R in a

stem gnathostome, mimicking a single pre-1R fusion event (Supplementary Fig. 23a); and (ii) our 17-chromosome model requires symmetric fissions of two AC3-derived post-1R chromosomes occurring in an ancestral cyclostome (Supplementary Fig. 23b). Although in silico simulations show that a scenario of pairwise post-1R fusions would not be extremely rare (30% of cases expected by chance; Supplementary Table 40; Methods), we believe the pairwise fissions to be more plausible given the higher level of reorganisation found in cyclostome karyotypes (see next section). Altogether, while we propose a scenario involving 17 ancestral vertebrate chromosomes, a scenario with 18 chromosomes (Nakatani et al., 2021; Simakov et al., 2022) is also possible."

Minor comment: on page 12, line 477, for completeness state that the interspecific homologous chromosomes are more similar in the instance where rediploidisation precedes speciation.

This is a very important clarification raised by the Reviewer, and a very interesting topic, one that we would like to address in the future for early vertebrates with improved hagfish and lamprey genome assemblies. We have now included a mention to this in our manuscript and added a citation to Redmond et al, 2023 here (now reference 46) as an example.

Fig 4 title has a hanging 'and'.

Corrected.

Reviewer #3 (Remarks to the Author):

In the revised version, the authors have addressed all my previous concerns appropriately - and in my assessment, also the issues raised by the other reviewers. The writing of the new version has much improved as it much better allows to understand the extensive and complex set of analyses and conclusions.

This study will be a highly important contribution to our understanding of the early evolution of the vertebrate lineage in the face of multiple WGDs.

We thank the reviewer for their enthusiasm and in-depth analysis of our manuscript, which helped us to significantly improve the overall quality of it.

I have one last item for the final version of the article, getting back to a point I raised before:

I.446-447 "Thus, key vertebrate innovations (e.g., elaborate tripartite brain, neural crest cell derived tissues among other novelties) originated in a tetraploid stem-vertebrate, before 2R."

In response to my previous question, the authors wrote that "we cannot reliably establish if they [the innovations] appeared before or after 1R, just along the stem vertebrate lineage."

I think that the word "tetraploid" needs to be deleted from the sentence in I.446-447, as it implies that 1R occurred before the innovations. Or, to make this even more clear, rephrase the sentence to something along the lines of:

"Thus, key vertebrate innovations (e.g., elaborate tripartite brain, neural crest cell derived tissues among other novelties) originated in a stem-vertebrate before 2R. However, at this point we cannot reliably establish if these innovations pre- or post-date the 1R event."

We agree with the reviewer. However, we noticed that "a stem-vertebrate before 2R" is redundant, since 2R occurred in the gnathostome stem, within the vertebrate crown, and thus anything occurring in the vertebrate stem has to have been prior to 2R, so we have also removed "before 2R". This also avoids the idea that these vertebrate innovations originated in a single organism and at the same time. This sentence now reads as follows:

Lines XXXX-XXXX: "*Thus, key vertebrate innovations (e.g., elaborate tripartite brain, neural crest cell-derived tissues among other novelties (Shimeld and Holland, 2000)) originated in a ~~tetraploid stem-vertebrate, before 2R. However, at this point we cannot reliably establish if these innovations pre- or post-date the 1R event.~~*"

Final Decision Letter:

4th December 2023

Dear Juan,

We are pleased to inform you that your Article entitled "Hagfish genome elucidates vertebrate whole genome duplication events and their evolutionary consequences", has now been accepted for publication in Nature Ecology & Evolution.

Over the next few weeks, your paper will be copyedited to ensure that it conforms to Nature Ecology and Evolution style. Once your paper is typeset, you will receive an email with a link to choose the appropriate publishing options for your paper and our Author Services team will be in touch regarding any additional information that may be required

Due to the importance of these deadlines, we ask you please us know now whether you will be difficult to contact over the next month. If this is the case, we ask you provide us with the contact information (email, phone and fax) of someone who will be able to check the proofs on your behalf, and who will be available to address any last-minute problems . Once your paper has been scheduled for online publication, the Nature press office will be in touch to confirm the details.

Acceptance of your manuscript is conditional on all authors' agreement with our publication policies (see www.nature.com/authors/policies/index.html). In particular your manuscript must not be published elsewhere and there must be no announcement of the work to any media outlet until the publication date (the day on which it is uploaded onto our web site).

Please note that *Nature Ecology & Evolution* is a Transformative Journal (TJ). Authors may publish their research with us through the traditional subscription access route or make their paper immediately open access through payment of an article-processing charge (APC). Authors will not be required to make a final decision about access to their article until it has been accepted. [Find out more about Transformative Journals](https://www.springernature.com/gp/open-research/transformative-journals)

Authors may need to take specific actions to achieve [compliance with funder and institutional open access mandates](https://www.springernature.com/gp/open-research/funding/policy-compliance-faqs). If your research is supported by a funder that requires immediate open access (e.g. according to [Plan S principles](https://www.springernature.com/gp/open-research/plan-s-compliance)) then you should select the gold OA route, and we will direct you to the compliant route where

79possible. For authors selecting the subscription publication route, the journal's standard licensing terms will need to be accepted, including <https://www.nature.com/nature-portfolio/editorial-policies/self-archiving-and-license-to-publish>. Those licensing terms will supersede any other terms that the author or any third party may assert apply to any version of the manuscript.

We welcome the submission of potential cover material (including a short caption of around 40 words) related to your manuscript; suggestions should be sent to Nature Ecology & Evolution as electronic files (the image should be 300 dpi at 210 x 297 mm in either TIFF or JPEG format). Please note that such pictures should be selected more for their aesthetic appeal than for their scientific content, and that colour images work better than black and white or grayscale images. Please do not try to design a cover with the Nature Ecology & Evolution logo etc., and please do not submit composites of images related to your work. I am sure you will understand that we cannot make any promise as to whether any of your suggestions might be selected for the cover of the journal.

You can generate the link yourself when you receive your article DOI by entering it here: <http://authors.springernature.com/share>.

Yours sincerely,

[REDACTED]

P.S. Click on the following link if you would like to recommend Nature Ecology & Evolution to your librarian <http://www.nature.com/subscriptions/recommend.html#forms>

80** Visit the Springer Nature Editorial and Publishing website at http://editorial-jobs.springernature.com?utm_source=ejp_NEcoE_email&utm_medium=ejp_NEcoE_email&utm_campaign=ejp_NEcoE for more information about our career opportunities. If you have any questions please click [here](mailto:editorial.publishing.jobs@springernature.com).**